# Evidence of elevation-dependent warming from the Chinese Tianshan Mountains

Lu Gao[1,2,3,4], Haijun Deng[1,2,3,4], Xiangyong Lei[3], Jianhui Wei[5], Yaning Chen[6], Zhongqin Li[7], Miaomiao Ma[8], Xingwei Chen[1,2,3,4], Ying Chen[1,2,3,4], Meibing Liu[1,2,3,4], Jianyun Gao[9]

[1]Institute of Geography, Fujian Normal University, Fuzhou 350007, China
[2]Fujian Provincial Engineering Research Center for Monitoring and Assessing Terrestrial Disasters, Fujian Normal University, Fuzhou 350007, China
[3]College of Geographical Sciences, Fujian Normal University, Fuzhou 350007, China
[4]State Key Laboratory of Subtropical Mountain Ecology (Funded by the Ministry of Science and Technology and the Fujian province), Fujian Normal University, Fuzhou 350007, China
[5]Institute of Meteorology and Climate Research (IMK-IFU), Karlsruhe Institute of Technology, Campus Alpine, Garmisch-Partenkirchen 82467, Germany
[6]State Key Laboratory of Desert and Oasis Ecology, Xinjiang Institute of Ecology and Geography, Chinese Academy of Sciences, Urumqi 830011, China
[7]State Key Laboratory of Cryospheric Sciences/Tianshan Glaciological Station, Northwest Institute of Eco-Environment and Resources, Chinese Academy of Sciences, Lanzhou 730000, China
[8]China Institute of Water Resources and Hydropower Research, Beijing 100038, China
[9]Fujian Key Laboratory of Severe Weather, Fuzhou 350001, China

*Correspondence to*: Lu Gao (l.gao@foxmail.com)

**Abstract.** The phenomenon in which the warming rate of air temperature is amplified with elevation is termed elevation-dependent warming (EDW). It has been clarified that EDW can accelerate the retreat of glaciers and melting of snow, which can have significant impacts on the regional ecological environment. Owing to the lack of high-density ground observations in high mountains, there is widespread controversy regarding the existence of EDW. Current evidence is mainly derived from typical high mountain regions such as the Swiss Alps, the Colorado Rocky Mountains, the Tropical Andes and the Tibetan Plateau/Himalayas. Rare evidence in other mountains has been reported, especially in arid regions. In this study, EDW features (regional warming amplification and altitude warming amplification) in the Chinese Tianshan Mountains (CTM) were detected using a unique high-resolution (1 km, 6-hourly) air temperature dataset (CTMD) from 1979 to 2016. The results showed that there were significant EDW signals at different altitudes on different time scales. The CTM showed significant regional warming amplification in spring, especially in March, and the warming trends were greater than those of continental China with respect to three temperatures (minimum temperature, mean temperature and maximum temperature). The significances of EDW above different altitude thresholds are distinct for three temperatures in twelve months. The warming rate of the minimum temperature in winter showed a significant elevation dependence ($p < 0.01$), especially above 3000 m. The greatest altitudinal gradient in the warming rate of the maximum temperature was found above 4000 m in April. For the mean temperature, the warming rates in June and August showed prominent altitude warming amplification but with different significances above 4500 m. Within the CTM, the Tolm Mountains, the eastern part of the Borokoonu Mountains,

the Bogda Mountains and the Balikun Mountains are representative regions that showed significant altitude warming amplification on different time scales. This new evidence could partly explain the accelerated melting of snow in the CTM, although the mechanisms remain to be explored.

## 1 Introduction

Elevation-dependent warming (EDW) indicates that the warming rate of air temperature is amplified with elevation, especially in high mountain regions. Two basic characteristics, regional warming amplification and altitude warming amplification are considered to be the "fundamental questions" of EDW by Rangwala and Miller (2012). Regional warming amplification means that the warming rate of air temperature in a certain mountain is greater than that in other regions outside of these mountain ranges. Altitude warming amplification means that the warming rate is greater in high-altitude

areas than in low-altitude areas in the same mountain. Rangwala and Miller (2012) also concluded that the EDW exists in some typical high mountain regions (e.g. Alps) because altitude warming amplification can be detected by both observation and climate models. However, regional warming amplification is still difficult to detect because of the limited observations at the global scale. Therefore, to some extent, EDW could be determined once regional warming amplification or altitude warming amplification could be detected.

Owing to the high sensitivity of glaciers and snow to climate change, mountains are regarded as outposts of global climate change (Sorg et al., 2012; Immerzeel et al., 2020). Previous studies have reported the potential widespread existence of the elevation-dependent warming (EDW) phenomenon, which is an ideal early indicator of climate warming in mountain systems under global climate change (Dong et al., 2015; Li et al., 2020). EDW can accelerate the changes in mountain ecosystems, cryosphere systems, water cycles and biodiversity, leading to irreversible and profound impacts on the regional

ecological environment and socioeconomic development (Pepin et al., 2015; Rangwala and Miller, 2012). Therefore, the detection and exploration of the spatial and temporal differentiation characteristics of EDW play a crucial role not only in the in-depth understanding of regional climate change and in improving the predictive ability of mountain climate, but also in maintaining the relative stability and ecological balance of these natural mountain ecosystems (You et al., 2020).

Current evidence for the EDW phenomenon mainly stems from multi-source data detection and regional climate models.

The main data resources include ground meteorological stations, radiosonde, reanalysis, and remote sensing data. For example, the warming rate has been found to be more intense in the high-altitude regions of Western Europe and Asia based on a global high-altitude observation dataset (Diaz and Bradley, 1997). The significant EDW phenomenon of the annual maximum and minimum temperatures in the Alps was detected based on ground observation sites (Beniston and Rebetez, 1996, Jungo and Beniston, 2001). A significant EDW phenomenon for the annual mean temperature in tropical alpine areas

was detected based on global radiosonde data (Seidel and Free, 2003). The warming trends for the maximum and minimum temperatures showed significant elevation dependence in the 2000–4000 m altitude range of the Rocky Mountains (Diaz and

Eischeid, 2007; Mcguire et al., 2012). The climate warming trends for the Qinghai-Tibet Plateau from 1961 to 1990 were proportional to altitude, especially in winter (Liu and Hou, 1998). In the high-altitude areas (above 4000 m) of the Qinghai-Tibet Plateau, the increment in the mean temperature over four seasons and on the annual scale is greater than that in the low-altitude areas (Du, 2001), and the temperature warming rate increases by 0.16 ℃ 10a$^{-1}$ for every 1 km increment in altitude (Wang et al., 2012). An observation dataset containing 139 weather stations showed warmer trends at 4000 m (0.5 ℃ 10a$^{-1}$) and 3000 m (0.357 ℃ 10a$^{-1}$) than at 2000 m (0.316 ℃ 10a$^{-1}$) with respect to the annual mean temperature in the Tibetan Plateau from 1961to 2012 (Yan and Liu, 2014). In general, from a regional perspective, the European Alps (e.g. Jungo and Beniston, 2001), Himalayan-Tibetan Plateau (e.g. Pepin et al., 2019), South American Andes (e.g. Vuille et al., 2015), and North American Rocky Mountains (e.g. Diaz and Eischeid, 2007) are hotspots for EDW studies (Wang et al., 2014; Thakuri et al., 2019; Guo et al., 2019). From the perspective of the significance of EDW, the seasonal scale is more significant than the annual scale because of significant changes in climate drivers such as snow/ice cover, clouds and others at different elevations (Pepin et al., 2015; Rangwala and Miller, 2012). Furthermore, the warming rate of the minimum temperature is greater than that of the maximum and mean temperatures on a global scale (Rangwala and Miller, 2012; Pepin et al., 2015).

Although many studies have detected EDW phenomena in different mountains globally, a widespread controversy still exists, and no consensus has been reached on the existence of EDW (You et al., 2020). This is mainly due to the scarcity of ground observation data, especially in mountains above 3000 m (Rangwala and Miller, 2012; Pepin et al., 2015). Even the detection of EDW is different within the same mountain that uses different observations (i.e., different numbers of sites or different site locations). For example, some studies have shown a significant prevalence of EDW since the second half of the 20th century over the Tibetan Plateau (Liu et al., 2009; Rangwala et al., 2009). However, an analysis claimed that the EDW over the Qinghai-Tibet Plateau is not significant based on observations from 71 ground stations and 56 reanalysis grid (NCEP and ERA-40 grid with a spatial resolution of 2.5°×2.5°) data (You et al., 2010). Similarly, EDW has been found to be insignificant at altitudes above 5000 m based on observations from 25 ground stations and 0.5° grid data combined with WRF model simulations (Gao et al., 2018b). Although satellite data compensate for the deficiencies of ground observation stations to a large extent, the associated relatively short time series, long revisiting cycle, and image interpretation and inversion errors limit the reliability of EDW signal detection. It can be concluded that a uniform high-resolution air temperature dataset is the basic premise for accurate EDW detection.

As the farthest mountain system from the ocean, and the largest mountain system in arid regions of the world, the Tianshan Mountain system is extremely important for assessing climatic changes and the ecological environment in north-western China and the entire nation because of its special geographical location and complex terrain (Chen et al., 2016, Li et al., 2020). As the "water tower" of Central Asia, the Tianshan Mountain system not only breeds many rivers, but also produces a unique desert oasis ecosystem (Sorg et al., 2012; Chen et al., 2016; Immerzeel et al., 2020). There are approximately 9035

glaciers with an area of ~9225 km$^2$ and of 1011 km$^3$ water resources (ice volume) in the Chinese Tianshan Mountains (CTM, Fig. 1) (Shi et al., 2009). However, recently, most glaciers in the CTM are in a state of accelerated degradation due to climate warming (Ding et al., 2006; Chen et al., 2016; Sorg et al., 2012). The warming rate of mean temperature in the entire CTM has reached 0.32–0.42 °C10a$^{-1}$ in the past 50 years, which is much higher than the national average (Gao et al., 2018a; Xu et al., 2018). However, the EDW in the CTM still lacks systematic detection. Current research on climate warming in the CTM does not provide sufficient solid evidence for the EDW phenomenon. Therefore, in this study, EDW features in the CTM were comprehensively and systematically detected based on a unique high-resolution (1 km, 6-hourly) air temperature dataset (hereafter referred to as CTMD) (Gao et al., 2018a). The present study reveals the EDW characteristics for different temperature indicators at different time scales.

## 2 Data and Methods

### 2.1 CTMD

The lack of sufficient ground observations is the biggest obstacle in the accurate detection of the EDW phenomenon. This is one of the original intentions of the development of CTMD. Previous studies have shown that the ECMWF's third-generation reanalysis product, ERA-Interim, has a relatively small large-scale error (±2.5 K) and can capture the annual and seasonal climatologies very well (Gao et al., 2012, 2014, 2017; Simmons et al., 2010). Hairiguli et al (2019) concluded that the ERA-Interim could capture the inter-annual variations of monthly mean temperature in the CTM from via comparing with 45 observation sites from 1984 to 2016. Bai et al. (2013) also found that ERA-Interim temperature data are better than NCEP/NCAR data based on a comparison with nine observation sites in the CTM from 2004 to 2006. The systematic bias of ERA-Interim is mainly due to the height discrepancy between the ERA-Interim model height and observations (Gao et al., 2012, 2014, 2017). Thus, the bias could be significantly reduced for local climate trend investigations via an appropriate elevation correction procedure. A robust approach based on internal vertical lapse rates derived from different ERA-Interim pressure levels was developed to downscale the 0.25 ° grid ERA-Interim temperature to a 1 km grid derived from SRTM (Gao et al., 2018a). This scheme is fully independent of meteorological stations via Equation (1).

$$T_{1km} = T_{ERA\_025} + \Gamma \times \Delta h \tag{1}$$

$T_{ERA\_025}$ is the original 6-hourly ERA-Interim 2-m temperature at a 0.25 ° grid. $\Gamma$ describes the ERA-Interim internal lapse rates derived from the temperatures and geopotential heights at different pressure levels. For example, $\Gamma_{500\_700}$ indicates the lapse rate between the 500 hPa and 700 hPa pressure levels, which is calculated by the temperature differences divided by geopotential height differences between these two pressure levels. $\Delta h$ is the height difference between the ERA-Interim model height and the 1 km grid. Different $\Gamma$ values were used according to the altitude of the 1 km grid. In other words, if the 1 km grid is lower than 1500 m in altitude, $\Gamma_{850\_925}$ is applied because the geopotential height between the 850 and 925

hPa pressure levels ranges from 150 m to 1500 m. If the grid altitude is higher than 4000 m, $\Gamma_{500\_600}$ is applied to the downscaling model. The geopotential height at the 850 and 700 hPa pressure levels is the demarcation at altitudes of 1500 m and 3000 m, respectively. In total, four lapse rates ($\Gamma_{500\_600}$, $\Gamma_{600\_700}$, $\Gamma_{700\_850}$ and $\Gamma_{850\_925}$) were used for different altitude ranges according to the 1 km grids (Gao et al., 2018a). Therefore, the unique high-resolution (1 km, 6-hourly) air temperature dataset (CTMD) for the Chinese Tianshan Mountains from 1979 to 2016 has a spatial resolution of 1 km (total 356133 grids) with 6-hourly time step at 00, 06, 12, and 18 UTC. More information regarding the downscaling scheme and on the CTMD can be found in Gao et al. (2012, 2017, 2018a).

Although, the CTMD was validated by 24 meteorological stations on a daily scale, indicating a high reliability for the climatology trend investigations, its limitations must be fully demonstrated. Whether the lapse rate accurately reflects the temperature changes at all altitudes is worth discussing. For example, the lapse rates of ERA-Interim are greater than observations from September to December, whereas the lapse rate in the free atmosphere is steeper than that near the surface because of the different radiation mechanisms (Gao et al., 2018a). The lapse rate may be positive rather than negative because of the "Cold Lake" effect in winter, such as in the Turfan Basin, which may lead to a temperature inversion layer at night. In this situation, the downscaling model may be disabled during winter. Therefore, the opposite trend for minimum temperature during winter was captured by the CTMD compared to the slight positive warming trend from the 24 observation sites. Meanwhile, the trend of the diurnal temperature range (DTR) was not captured very well by the CTMD in spring and autumn (Gao et al., 2018a). We emphasise that the CTMD is only validated by 24 sites, which are mainly located in low elevation terrain. Evaluating the credibility of the CTMD in the high peaks is difficult because few observations exist. Nevertheless, we believe that the CTMD is still creditable because it can capture the distribution characteristics of temperatures as well as the general warming trends.

## 2.2 CMA05

According to the two basic characteristics for the diagnosis of mechanisms responsible for EDW, regional warming amplification and altitude warming amplification were detected. The former feature is compared with those of other regions, such as plateaus, mountains, and low-altitude areas (basins and plains), to detect whether the warming trend in mountains is higher. The latter focuses on the warming trend differences within the mountains (e.g., different altitude ranges), which is to determine whether warming amplification in high-altitude areas is more significant than that in low-altitude areas. To detect the regional warming amplification, CMA05 was evaluated and compared with the CTMD.

The CMA05 dataset was obtained from the China Meteorological Data Sharing Service System of the National Meteorological Information Centre (http://data.cma.cn/data/cdcdetail/dataCode/SURF_CLI_CHN_TEM_MON_GRID_0.5.html). It contains three monthly temperature indices (minimum, mean and maximum temperature) at the 0.5 ° latitude-longitude grid. The CMA05 is highly

reliable and has been widely applied in climatology studies since it was interpolated using the thin plate spline method based on high-density ground stations (approximately 2400 national meteorological observation stations) since 1961 (Sun et al., 2015; Wu et al., 2017). A common time period 1979–2016 was extracted for the current study. Previous studies found that the Qinghai-Tibetan Plateau (QTP) has a significant temperature warming trend over China (e.g. You et al., 2010). To reduce the influence of warming, the Qinghai-Tibetan Plateau was eliminated. Thus, not only the whole continental China (WCC), but the low-altitude areas (LCC), represented by excluding the Tianshan Mountains and the Qinghai-Tibetan Plateau from the whole continental China, were also used for comparison.

## 2.3 Snow cover and snow depth data

To further discuss the possible hypotheses and mechanisms with respect to EDW, snow cover and snow depth data in the CTM were collected. The snow cover fraction was calculated by dividing the snow cover area by the total area. The snow cover area was interpreted based on the MODIS/Terra Snow Cover 8-Day L3 product (MOD10A2, version 5) with a 500 m spatial resolution from the NASA Snow and Glacier Data Centre (https://nsidc.org/data/MOD10A2/versions/5). The annual maximum and minimum snow cover fractions (only two values per year) in the CTM from 2002 to 2013 were calculated. This dataset was processed and provided by Chen et al. (2016) and Deng et al. (2019).

The daily snow depth data at a spatial resolution of 25 km from 1979 to 2016 over the CTM were derived from the National Earth System Science Data Centre, National Science & Technology Infrastructure of China (Che 2015). The snow depth was calculated based on the original daily passive microwave brightness temperature data (EASE-Grid) produced from SMMR (1979-1987), SSM/I (1987-2007) and SSMI/S (2008-2019) from the National Snow and Ice Data Centre (NSIDC). The monthly snow depth calculated from daily depth was applied for the follow up analysis. Detailed information on the data production can be found in studies of Che et al. (2008), Dai et al. (2015) and Dai et al. (2017).

## 2.4 Analytical methods

In this study, the 6-hourly (00, 06, 12 and 18 UTC) data of the CTMD were aggregated to the minimum temperature (Tmin), maximum temperature (Tmax), and mean temperature (Tmean) on daily, monthly, seasonal, and annual time scales. A standard linear regression was applied to calculate the warming rate in each grid from 1979 to 2016 for the CTMD and CMA05 datasets. The corresponding equation is given as follows:

$$y = \alpha x + \beta \tag{2}$$

where y is the temperatures (Tmin, Tmax and Tmean) on different time scales, x is the time series from 1979 to 2016, and the fitting coefficient (slope) $\alpha$ indicates the warming rate. Thus, in this study, EDW refers to the rate of warming over a multi-annual scale.

To detect the altitude warming amplification within the CTM, the entire altitude range was divided into 14 groups with 500 m intervals (Table 1). The numbers of grids in each group are listed in Table 1. Standard linear regression was also used to assess different significance levels ($p < 0.1$, $p < 0.05$, and $p < 0.01$) of EDW for different altitude groups. In this analysis, y is the warming rate (calculated by the equation 2) from 1979 to 2016 for each altitude group. The average warming rate of each group was used for the regression because of the different number of grids in each altitude group. Here, x denotes the 14 altitude groups (natural positive integers 1 to 14). Thus, the fitting coefficient (slope) represents the magnitude of the significance of EDW. The coefficients of determination ($R^2$) and confidence tests (p-values) illustrate the goodness of fit.

## 3 Results

### 3.1 Regional warming amplification of the CTM

The temperature trends on monthly, seasonal, and annual time scales with respect to Tmin, Tmax and Tmean were calculated using Eq. (2) based on the aggregated Tmin, Tmax and Tmean from 6-hourly (00, 06, 12 and 18 UTC) CTMD data for each grid. Table 2 shows the ratio of the sum of grids at different significance levels ($p < 0.1$, $p < 0.05$, and $p < 0.01$) to the total grids (356133) with respect to monthly temperatures. All grids reached the significant levels for Tmax in March, followed by 99.35% of all grids for the Tmean. The number of grids that reached the significance level was the lowest in December, especially for Tmean and Tmax. For Tmin, more than half of the grids in only two months (March and June) reached the significance levels. For Tmean, five months (March, April, June, July and August) exceeded 50% grids at the significance levels. For Tmax, only February and March had more than half of all grids at the significance level. Although the temperature trend at some grids in a certain month did not reach a statistically significant level, it can still reflect climate warming on a regional scale to a certain extent. Thus, the subsequent analysis depends on the temperature trend of all grids.

The annual and seasonal temperature trends in the CTM were weaker than those over WCC with respect to the mean temperature (Tmean), maximum temperature (Tmax), and minimum temperature (Tmin), except during spring (Table 3). The warming rates in the Tmax and Tmin of spring both exceeded 0.6 ℃ 10a$^{-1}$, which is much higher than that of WCC and LCC which is represented by excluding the Tianshan Mountains and the Qinghai-Tibetan Plateau from the whole continental China. The summer Tmin and Tmean trends of CTM were also higher than those of LCC. The annual Tmin showed the greatest warming trend with a rate of 0.347 ℃ 10a$^{-1}$, followed by Tmax and Tmean with warming rate of 0.323 and 0.245 ℃ 10a$^{-1}$, respectively, in the CTM (Table 3). While summer had a much higher trend than autumn for Tmean and Tmin, it showed a comparable rate for Tmax (Table 3). Winter had the lowest rates compared with other seasons for the three temperature trends, with even a decreasing trend (−0.085 ℃ 10a$^{-1}$) observed for Tmean. In general, Tmin and Tmax showed comparable rates in spring. A more significant increase in Tmin compared with Tmax was observed in summer and autumn. However, the trends of CTM were consistent with those of WCC and LCC, except for the winter Tmean (Table 3).

The warming rates vary from month to month, which is more significant than that from season to season. All temperature trends were negative in January and December in the CTM, which was different from that in the WCC and LCC (Table 4). The rate of decrease was more significant in January than in December. Notably, Tmax decreased slightly in May, whereas Tmin warmed significantly at a rate of 0.624 °C 10a$^{-1}$ in the CTM. The largest warming rates were observed for both the CTM and land surface of China in March for all temperature types. However, the CTM had a higher magnitude of warming. The warming trend was 1.339 °C 10a$^{-1}$, which was almost double that over the whole of China (Table 4). Both rates exceeded 0.8 °C 10a$^{-1}$ for Tmean and Tmin in the CTM in March. April showed the second largest Tmax and Tmean warming trends in the CTM, which were also higher than those over continental China. For Tmin, May and June had rates greater than 0.6 °C 10a$^{-1}$. The significant warming trends from March to May resulted in higher trends in CTM than in continental China, especially in March (Table 4). In general, a more significant increment in Tmin was observed from March to June compared to that in other months. March and April showed remarkable warming trends for Tmax and Tmean (Table 4). In the entire CTM, Tmin increased faster than Tmax and Tmean. In general, regional warming amplification was significant in March and June at all temperatures. The trend for Tmax also increased faster in the CTM in February and April than over the entire land surface of China. The warming rates of Tmean and Tmin in the CTM were faster than those in the WCC and LCC in April and May, respectively. All temperature trends at different time scales in the WCC were higher than LCC, which implies that the warming rate of the Qinghai-Tibet Plateau contributes significantly to regional warming. It is worth noting that the warming trends are larger in both Tmin and Tmax than in Tmean in some months for CTM, WCC and LCC (Table 3 and 4). The possible reason is very complicated. It is not only related to the local physical mechanisms that may change the diurnal cycle, but also related to the data resource which is dependent on the time of day (00, 06, 12, and 18 UTC). You et al (2020) also found this phenomenon over the Tibetan Plateau based on multiple data products.

## 3.2 Warming amplification with altitude within the CTM

The performances of different temperature types (Tmin, Tmean and Tmax) were diverse for different months. The monthly and seasonal temperature trends were calculated for each grid based on the averaged 6-hourly data. To detect the altitude warming amplification features in the mountain areas, the CTM was divided into 14 groups with a 500 m altitude interval (Table 1). Notably, the temperature trends in the different elevation groups were significant distinct than those of the entire CTM. Table 5 to 7 summarized the warming amplification with altitude for monthly Tmin, Tmean and Tmax over different altitudes from 1979 to 2016, respectively. The slope of fitting line between monthly temperature warming trends and altitude groups illustrated the significance of EDW above different altitude thresholds. For Tmin, the EDW could be found at full altitude (204-7100 m) in winter (January, February, and December). However, the most significant EDW appeared at altitudes above 3000 m in January and December (Table 5). The EDW was significant for April Tmin at altitudes below 3500 m. The EDW begins to appear above 2500 m altitude, but it is most significant above 4500 m in July (Table 5). The 4500 m is the threshold for the most significant EDW for Tmean in April, June, August, and September (Table 6). However, the EDW appeared at altitude above 2000 m in June. From January to March, the EDW exists over all elevation groups. The

most significant EDW was found at the elevation higher than 3000 m in January (Table 6). The $T_{max}$ in January, March and April showed the EDW over the whole elevations, but at different altitudes for the most significant EDW (Table 7). Above 3000 m altitude, the most significant EDW was found in January while above 2000 m in March. The EDW become more and more significant from 2000 m to 4000 m in April. From August to October, the 4500 m was the threshold for the most significant EDW (Table 7). In general, the altitudes at which the EDW phenomenon appeared are different for three temperatures in twelve months. The EDW differences (slope values) demonstrated the magnitudes of warming amplification at different altitudes. For example, for $T_{min}$, the most significant EDW could be detected at the altitude 3000 m in winter. For $T_{max}$, the EDW could be detected above 2000 m and 2500 m in April and August, respectively. However, the most significant warming amplification with altitude was found above 4000/4500 m in these two months (Table 7).

As the further explication of Table 5 to 7, Fig. 2 to Fig. 4 intuitively provided the temperature trends at different altitude groups as well as the significance of EDW above different altitude thresholds. Fig. 2 showed the $T_{min}$ warming trends in January, February, April, and December from 1979 to 2016. The fitting between temperature trends and altitude groups above 3000 m illustrated the warming amplification with altitude. As the number of grids in each elevation group is different, the boxplots show the interquartile range (25% to 75%) and median values. To maintain consistent trend calculation for the entire study, the average value was used for linear regression. Meanwhile, linear regression was applied based on the average values, which indicated the altitude dependence of the warming trend (i.e. the significance of EDW). In general, the EDW characteristics were significant for $T_{min}$ in January, February, April, and December. All lines of best fit are at the 0.01 significance level ($p < 0.01$). The temperature trends were positive at altitudes higher than 5000 m, with median values greater than 0 ℃ 10a$^{-1}$ above 4000 m in January (Fig. 2a). The median values of most elevation groups were above the reference line in February, although the corresponding line of fit had a lower slope (0.025) compared with that of January (Fig. 2b). The 75% quartile ranges of the trends for all elevation groups in April were higher than 0 ℃ 10a$^{-1}$ (Fig. 2c). All trends were positive for the regions above 4000 m in April. The prevalence of EDW was the most significant in December with the highest slope (0.096, $p < 0.01$). Although, most of lower altitude grids (< 4000 m) showed negative warming trends, the trends become positive at altitudes higher than 5000 m (Fig. 2d).

Fig. 3 showed the $T_{mean}$ warming trends in January, June, August, and September from 1979 to 2016. The altitude threshold was 4500 m. The warming rates in January were only slightly above 0 ℃ 10a$^{-1}$ at higher elevations. The significance of EDW was at the 0.05 significance level with a slope of 0.016 (Fig. 3a). The most significant EDW (slope = 0.045, $p < 0.01$) was found in June with all positive trends at all altitudes (Fig. 3b). The second significant EDW occurred above an elevation of 4500 m in August (slope = 0.037, Fig. 3c). The warming rates were above 0 ℃ 10a$^{-1}$ at higher elevations (> 4500 m). The fitting slope is 0.017 at the 0.05 significance level in September (Fig. 3d).

A threshold of 4000 m was applied for the significance of EDW investigation for Tmax in March, April, August, and September (Fig. 4). Although the slope (0.023) of the trend was not remarkable, all warming rates were greater than 0.8 ℃ 10a$^{-1}$ in March (Fig. 4a). A significant elevation-dependent cooling could be observed in the altitude range of 0–2500 m for Tmax in April. However, the most significant EDW also was detected above the altitude of 4000 m (slope = 0.09, Fig. 4b). Most of the warming rates were higher than 0.4 ℃ 10a$^{-1}$ in April. The EDW occurs at higher height of 4000 m in August and September (Fig. 4c and 4d). However, the EDW is more significant above 4500 m in September than that above 4000 m (Table 7). The temperature trends for all months and seasons are also provided in the supplementary material (Fig. S1-S13). Previous studies also found that EDW is significant at different altitudes. For example, Li et al (2020) found a significant EDW in the altitude of 2500–5000m from 1980-2012 in the high mountain Asia. You et al (2020) concluded a clear EDW above 2000 m in the Tibetan Plateau in 1961-1990.

**3.3 Spatial distribution pattern of the warming trend over the CTM**

The spatial distribution of warming trends for all months and seasons can be found in the supplementary material (Fig. S14–S30). In general, the warming trend of the mean temperature was not as dramatic as that of the minimum and maximum temperatures in the CTM. To better detect altitude warming amplification, four typical zones with high mountains (above 3000 m) were selected: Zone 1 (represented by the Tolm Mountains), Zone 2 (central Tianshan, including the eastern part of the Borokoonu Mountains), Zone 3 (represented by the Bogda Mountains), and Zone 4 (represented by the Balikun Mountains) (Fig. 5). The monthly minimum temperature trends of January in the higher altitude mountains were greater than those in the surrounding LCCs, especially in Zones 3 and 4 (Fig. 5a). The highest warming trend (exceeding 1.0 ℃ 10a$^{-1}$) was observed around the eastern Bogda Mountains (above 3000 m) in Zone 3. The lowlands to the north of the Bogda Mountains showed a cooling trend (Fig. 5a). Zone 4 also showed a remarkable EDW phenomenon (0.3–0.6 ℃ 10a$^{-1}$), wherein high mountains such as the Balikun were slightly warmer than the surrounding lowlands. Although the warming trend of Zone 1 was not as distinct as that of Zones 3 and 4, compared with the Ili Valley (cooling trend), the warming rate was still remarkable (~0.4 ℃ 10a$^{-1}$). In December, the warming trend was more significant in Zone 1 than in the other zones (Fig. 5b). The trend in the Tolm Mountains (exceeding 0.4 ℃ 10a$^{-1}$) was much higher than that in the Ili Valley (cooling trend), which is located in the northern part of Zone 1. The warming rate at high altitudes in Zone 3 was higher (0.2–0.4 ℃ 10a$^{-1}$) than that in the lowlands. There was no obvious warming amplification in the high-altitude mountains of Zone 4 to the low-altitude areas (Fig. 5b). However, it is worth noting that even in the same mountainous areas, such as in the Bogda Mountains in Zone 3, the warming rate in the east was notably higher than that in the northwest. It is clear that the minimum temperature warming trends accelerated as the elevation increased in January and December. The warming trends become positive at an altitude of approximately 3000 m (Fig. 5).

Zones 1 and 4 tended to exhibit the altitude warming amplification phenomenon for the monthly mean temperature in January (Fig. 6a). The temperature decreased (by approximately −0.2 to −0.4 ℃ 10a$^{-1}$) in the Ili Valley but increased

(approximately 0.05 to 0.15 °C 10a$^{-1}$) in the Tolm Mountains, especially in the high-altitude areas (Fig. 6a). Zone 4 warmed faster than regions outside the zone. However, the warming trend was not notable in the high-elevation areas compared with that in the lowlands within this zone (Fig. 6a). The temperatures showed cooling trends in Zones 2 and 3. Nevertheless, the trend was amplified with an elevation in January in Zone 2. The high-altitude areas were warmer than the low-altitude regions, especially in the Bogda Mountains of Zone 3 (Fig. 6a). The spatial distribution of the warming rate in February was similar to that in January. However, the trend in most areas of the CTM was positive (Fig. 6b). Zones 3 and 4 showed obvious EDW phenomena in February. The difference between the temperature warming rates in the high and low terrains of these two zones exceeded 0.2 °C 10a$^{-1}$. The trend in the high terrain of Zone 2 was greater than that in the valleys in the western part of the zone (eastern Ili Valley). However, the temperature in the south of the zone warmed faster than that in the high mountains in the northern part of Zone 2 (Fig. 6b). The southwestern Tolm Mountains in Zone 1 warmed up faster than the north-eastern mountains.

The maximum temperature in March in the entire CTM significantly increased with rates ranging from 0.9 to 2.0 °C 10a$^{-1}$ (Fig. 7a). The highest warming rate was observed in western Ili Valley. However, all typical zones showed strong altitude-warming amplification features. The areas above 4500 m in Zone 1 showed trends higher than 1.4 °C 10a$^{-1}$. The smoothed contour at 3000 m corresponds to a distinct boundary in Zone 2. The temperature warming rates were almost higher than ~1.5 °C 10a$^{-1}$ in the areas above 3000 m, whereas the rates were smaller in the low altitude areas (Fig. 7a). The trend was higher than ~1.1 °C 10a$^{-1}$ in March, whereas the trend became positive at approximately 2000 m in September (Fig. 7b). The difference between the warming rates in the high-altitude areas and low-altitude areas was the most remarkable in Zone 3. The temperature warming trend on the hilltop of the Bogda Mountains was much higher than that at the foot of the mountains (Fig. 7a). The temperature warming rate in Zone 4 ranged from 1.3 to 1.6 °C 10a$^{-1}$. The trend differences between the high-altitude areas and low-altitude areas in Zone 4 were not as remarkable as those in Zone 3 (Fig. 7a). However, the warming rate on the mountain peak was much higher than that in the neighbouring lowlands (Fig. 7a).

The spatial distribution of the maximum temperature in September showed distinctive east-west differentiation. The warming rates in Zones 3 and 4 were greater than those in Zones 1 and 2 (Fig. 7b). The EDW features were not notable in Zone 4. In contrast, the temperature in the high-altitude areas showed a slower warming trend (approximately 0.2–0.3 °C 10a$^{-1}$) than that in the low-altitude areas in Zone 3 (Fig. 7b). A slight EDW phenomenon was observed in the Tolm Mountains in Zone 1. However, Zone 2 showed remarkable EDW in September compared to that in the other zones. Similar to March, areas above 3000 m warmed faster than the lowlands, especially in the Ili Valley (Fig. 7b). In summary, Zone 2 was found to have a significant EDW area at the maximum temperature for March and September.

Figures 5 to 7 show the general features of EDW in four typical areas. Taking Zone 2 as an example, Fig. S31 showed the warming rate of Tmin in December was amplified with elevation in some certain transects (the grids in the red rectangles). However, there are also some elevation-dependent cooling phenomena in Zone 2 (the grids in the blue rectangle). It reveals

that altitude is not the only factor that affects temperature changes. The slope and aspect are other important factors responsible for the temperature changes due to the widespread valleys in the Zone 2. The local micro-terrains directly affect the absorption of solar radiation which would change the land surface processes such as latent heat, sensible heat and evapotranspiration. Thus, the EDW should be further detected on a finer spatial scale in some specific areas.

## 4 Discussion

Our analysis shows that the EDW phenomenon is very complicated for a large mountain system. It is difficult to arbitrarily judge the prevalence of EDW in mountain systems. Based on a comprehensive quantitative analysis, we believe that significant EDW signals exist in the CTM on local scales with respect to different temperature metrics. Although previous studies have mainly focused on the EDW of annual and seasonal temperatures, the monthly scale has not received sufficient attention. However, seasonal temperatures do not clearly reflect the EDW characteristics. In complex terrain, monthly temperature changes are more significant, especially during seasonal transitions. For example, rapid warming in March would accelerate the melting of ice and snow, affecting glaciers and regional water resources in the mountains.

The air temperature changes were mainly affected by two aspects: one is the vertical energy exchange between the ground and atmosphere, which leads to periodic changes on the daily and annual scales; and the other is the temperature advection caused by movement of cooling and heating masses, which leads to non-periodic changes. Numerous studies have shown that atmospheric circulation not only affects the latitude and zonality of climate via the zonal distribution of circulation but also expands the influence range of sea-land and topography via energy and water transportation (Dickinson, 1983; Harding et al., 2001). It is worth noting that the temperature trend is always positive at an altitude of 4500 m or higher in the CTM. However, the minimum temperature has a cooling trend in January and December below 4000 m (Fig. 2a and 2d). The significant altitude warming amplification phenomenon could be observed above 4500 m for the Tmean in June and August (Fig. 3 and Table 6). The significant EDW of Tmax could be observed above elevation of 4000 m in April. The air at high altitudes is similar to that in the free atmosphere and the dry adiabatic process is dominant. The absorption and reflection of solar radiation by the surface mainly determine the temperature change. In low-altitude areas, the impact of the underlying surface characteristics (e.g. terrain and land cover) is more significant. The CTM has a complex terrain with many mountain basins and canyons. Because of the "Cold Lake" effect in winter, the lapse rate is even positive. A temperature inversion layer often occurs in deep canyons at night. Meanwhile, in low-altitude areas, the more surface soil moisture results in an increase in the latent heat fluxes, which further causes more absorbed solar radiation and then temperature warming in winter (Rangwala et al., 2012). This mechanism is closely related to snowlines and treelines because the migration of snowlines and treelines changes the surface albedo (Pepin et al., 2015).

On a local scale, the ice and snow albedo, cloud cover, water vapour and radiation flux, and aerosols (including black carbon) are considered to be the main factors influencing EDW (Pepin et al., 2015; Rangwala and Miller, 2012). Guo et al (2021)

found that decreased snow depth caused by enhanced regional warming controls the pattern of EDW on the Tibetan Plateau. However, irrespective of clouds, water vapour, and aerosols, the core mechanism is that they affect the absorption of solar short-wave radiation by the land surface and long-wave radiation outward from the land surface (Shi et al., 2020; Zhang et al., 2018). The balance of surface energy changes leading to increasing/decreasing near-surface air temperature. In other words, the surface energy balance is the key mechanism that affects seasonal and interannual changes in EDW (Rangwala and Miller, 2012).

Surface albedo is a comprehensive indicator of many factors that affect surface energy balance. It is also the core factor and key variable that controls the surface energy budget (Dickinson, 1983; Harding et al., 2001; Wang et al., 2005). Numerous factors such as terrain, vegetation cover, ice and snow, soil moisture, soil physical properties, and meteorological conditions affect surface albedo (Zhang et al., 2018). For high mountain regions, vegetation cover and ice/snow cover are the two most important factors (Dickinson, 1983; Zhang et al., 2016; Zhang et al., 2018). For the entire CTM, small glaciers are more sensitive to the warming climate, and the estimated glacier mass loss could be $-2.3 \times 10^3$ kg m$^{-2}$ below 3000 m, especially in Zone 2 (Deng et al., 2019). The snow cover and its duration also show a decreasing trend (Sorg et al., 2012; Deng et al., 2019). Guo and Li (2015) found a decreasing trend in the ratio of snowfall to precipitation (S/P) in the CTM, especially in the four typical zones (Zones 1 to 4) in this study.

Deng et al. (2019) found that the snow cover fraction in the CTM decreased at a rate of 0.44% from 2002 to 2013. According to the snow cover fraction data from Chen et al. (2016) and Deng et al. (2019), the maximum snow cover fraction always occurred in January, whereas the minimum snow cover occurred in July. We tested the relationship between monthly Tmin, Tmean and Tmax with the maximum/minimum snow cover fraction for each month from 2002 to 2013. Table 8 and 9 showed the relationships between the temperatures and maximum/minimum snow cover fractions. However, only Tmin in February had a strong correlation ($R^2 = 0.302$, $p < 0.1$) with the maximum snow cover fraction (Fig. 8a). For the minimum snow cover fraction, Tmax in August ($R^2 = 0.256$, $p < 0.1$) and Tmean in May ($R^2 = 0.296$, $p < 0.1$) showed significant correlations (Fig. 8b, Table 9). The correlations between temperatures and the snow cover fractions in other months were not significant (Table 8 and 9).

The annual trend of snow depth over the CTM from 1979-2016 was -0.12 cm 10a$^{-1}$, which means that the melting of snow is accelerating. Except January (0.16 cm 10a$^{-1}$) and February (0.05 cm 10a$^{-1}$), the reduction in snow depth in other months ranged from -0.01 to 0.58 cm 10a$^{-1}$. The snow depth reduced the fastest in March with a rate of -0.58 cm 10a$^{-1}$, followed by April with a rate of -0.45 cm 10a$^{-1}$. Thus, spring had the highest decreasing trend of snow depth. However, the temperature warming trends were most significant in spring and March with respect to Tmin, Tmean and Tmax (Tables 3 and 4). The relationship between snow depth and temperature was further investigated in the CTM from 1979 to 2016 (Table 10). A significant correlation ($p < 0.01$) was found between Tmin and snow depth in March and June. For the couple of Tmean and snow depth, a correlation ($p < 0.01$) was also found in March, June and August, respectively. The significant correlation ($p <$

0.01) was observed only in December between Tmax and snow depth (Table 10). In cold months, for example, November and January, a relatively significant correlation ($p < 0.05$) was found between Tmean/Tmax and snow depth. Figure 9 shows the scatter plots of the comparison of Tmin and Tmean in March with the snow depth. A negative correlation was perspicuous and visible. In general, there was a negative correlation between temperature and snow cover/snow depth (Figs. 8 and 9), which implies that temperature warming promotes the accelerated melting of snow. Meanwhile, the accelerated

melting of snow may affect temperature warming. The detailed feedback mechanism between snow and temperature needs to be further verified and explored using advanced technology and models. In summary, although many hypothetical mechanisms of EDW have received widespread attention, most of them are limited to phenomenon description and qualitative analysis. The present study attempted to conduct preliminary explorations of the mechanism based on limited snow cover and snow depth data. There is a lack of quantitative investigations on the core processes, dominant factors, and

spatio-temporal differences of EDW.

## 5 Conclusions

Compared with the warming trend over the national land surface (WCC) and the low-altitude areas (LCC), the CTM is warming faster (0.633 and 0.640 °C 10a$^{-1}$ for Tmin and Tmax, respectively) in spring (Table 3). However, on a monthly scale, warming rates are more complicated. The warming trends of the three temperature indicators (Tmin, Tmax, and

Tmean) in March (0.835 and 1.339 °C 10a$^{-1}$ for Tmin and Tmax, respectively) and June (0.752 and 0.422 °C 10a$^{-1}$ for Tmin and Tmax, respectively) in the CTM were higher than those over the entire national land surface on an average (Table 4). In addition, Tmax in February, Tmax and Tmean in April, and Tmin in May were also higher than the national average. Therefore, EDW detection based on a monthly scale was more reasonable.

It cannot be simply concluded that the high-altitude areas are warming faster than the low-altitude areas. Quantitative

analysis is required to provide solid evidence for the EDW phenomenon. Using altitude grouping and a linear regression model, we quantitatively determined the significance of EDW along with the detailed performance of the warming trends with respect to Tmin, Tmean and Tmax at different altitudes. In general, the altitude thresholds for EDW phenomenon are different for three temperatures in twelve months. In the case of Tmin, January, February, April, and December showed significant EDW trends ($p < 0.01$). The most significant EDW phenomenon was found in December (Table 5, Fig. 2d). In

general, Tmin was associated with a strong EDW in winter. The warming rates of Tmin above 5000 m were always positive, which could lead to faster melting of snow. March, April, August, and September showed different elevation-based sensitivities with respect to Tmax (Table 7). The most significant EDW phenomenon can be found at altitudes above 4000 m in April and August as well as above 4500 m in September (Fig. 4). Almost all Tmax warming trends in March and April were positive in the CTM. The significant EDW phenomena were identified above 4500 m for Tmax in June and August.

January showed the EDW over the entire CTM with respect to Tmean. But the greatest slope was found above the altitude of
        3000 m (Table 6).

        The CTM is a large mountain system comprising many mountains. Therefore, EDW characteristics are diverse in different
        mountains. The EDW of Tmin in January was significant in the Bogda and Balikun Mountains, whereas it was significant in
        December in the Tolm Mountains (Fig. 5). For Tmax in March, all typical mountains exhibited EDW characteristics,
especially the central CTM and Bogda Mountains (Fig. 6). A significant EDW signal of Tmax was observed in September in
        central CTM (eastern part of the Borokoonu Mountains). The most significant EDW signal of Tmean was observed in the
        Tolm and Balikun Mountains in January. The Bogda and Balikun Mountains exhibited significant EDW features in February.

        After preliminary research, a significant negative correlation ($p<0.01$) between minimum/mean temperature and snow depth
        was observed in March and June (Fig. 9). However, the specific feedback mechanism between snow and temperature
remains unclear. Even in the same mountainous area, significantly different mechanisms of EDW were observed for
        different topographies, altitudes, and seasons. Future studies should focus on conducting in-depth quantitative research on
        the mechanism of EDW based on regional climate models and field surveys, especially in Zones 1 and 2 with accelerating
        snow melting.

## 6 Data availability

The dataset is released at https://doi.org/10.1594/PANGAEA.887700 in the Network Common Data Form (NetCDF) format.
        The coverage of the dataset is 41.1814-45.9945 °N, 77.3484-96.9989 °E. The spatial resolution is 1km and the total number
        of grid points is 818126 for the larger Chinese Tianshan Mountain region, which includes more surrounding areas. This
        study used 356133 grids. The time step was 6-hourly at 00, 06, 12, and 18 UTC. The dataset contains 288 NetCDF files and
        one user guidance file. The monthly temperature data set at the 0.5 °latitude-longitude grid (CMA05) over  continental China
was provided by the China Meteorological Data Sharing Service System of the National Meteorological Information Center
        (http://data.cma.cn/data/cdcdetail/dataCode/SURF_CLI_CHN_TEM_MON_GRID_0.5.html, last access: 05 January 2021).
        The MODIS/Terra Snow Cover 8-Day L3 product (MOD10A2, version 5) for snow cover fraction calculation was provided
        by the NASA Snow and Glacier Data Centre (https://nsidc.org/data/MOD10A2/versions/5, last access: 07 January 2021).
        The daily snow depth data were provided by the National Earth System Science Data Centre, National Science &
Technology Infrastructure of China (http://www.geodata.cn, 05 January 2021).

## Author contributions

L.G. designed the research and collected the data, H.D., X.L. and J.W. contributed to the data processing and analysis, L.G.
wrote the manuscript, and M.M., X.C., Y.N.C., Z.L., J.G., Y.C. and M.L. contributed to the discussion.

**Competing interests**

The authors declare that they have no conflict of interest.

**Additional information**

More analysis figures could be found in the Supplementary material.

**Acknowledgements**

This study was supported by the National Natural Science Foundation of China (41501106, 41877167 and 41807159), the

National Key Research and Development Program (2018YFE0206400 and 2018YFC1505805), the Scientific Projects from

Fujian Provincial Department of Science and Technology (2019R1002-3), the Scientific Project from Fujian Key Laboratory

of Severe Weather (2020KFKT01). Dr. Jianhui Wei was supported financially by the German Research Foundation through

funding of the AccHydro project (DFG-Grant KU 2090/11-1).

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

**Table 1. Grid number and percentage for each altitude group over the CTMD.**

|    | Altitude range (m) | Grid number | Percentage (%) |
|----|--------------------|-------------|----------------|
| 1  | <500               | 3139        | 0.881          |
| 2  | 500–1000           | 30810       | 8.651          |
| 3  | 1000-1500          | 83018       | 23.311         |
| 4  | 1500-2000          | 70229       | 19.720         |
| 5  | 2000-2500          | 46545       | 13.069         |
| 6  | 2500-3000          | 43400       | 12.186         |
| 7  | 3000-3500          | 39579       | 11.114         |
| 8  | 3500-4000          | 28256       | 7.934          |
| 9  | 4000-4500          | 8789        | 2.468          |
| 10 | 4500-5000          | 1666        | 0.468          |
| 11 | 5000-5500          | 496         | 0.139          |
| 12 | 5500-6000          | 150         | 0.042          |
| 13 | 6000-6500          | 52          | 0.015          |
| 14 | >6500              | 4           | 0.001          |

**Table 2. Ratio (%) of grids at different significance levels ($p < 0.1$, $p < 0.05$, and $p < 0.01$) to total grids (356133).**

| | Tmin | | | Tmean | | | Tmax | | |
|---|---|---|---|---|---|---|---|---|---|
| | $p<0.1$ | $p<0.05$ | $p<0.01$ | $p<0.1$ | $p<0.05$ | $p<0.01$ | $p<0.1$ | $p<0.05$ | $p<0.01$ |
| January | 1.57 | 1.47 | 0.24 | 2.65 | 1.00 | 0.00 | 4.38 | 2.10 | 0.00 |
| February | 5.18 | 4.48 | 0.01 | 0.55 | 0.00 | 0.00 | 23.50 | 27.28 | 5.87 |
| March | 17.26 | 21.26 | 13.50 | 4.30 | 19.61 | 75.44 | 0.54 | 7.57 | 91.88 |
| April | 3.09 | 0.66 | 0.00 | 8.93 | 14.26 | 45.97 | 8.13 | 19.05 | 19.18 |
| May | 11.78 | 15.99 | 19.19 | 12.47 | 16.31 | 0.44 | 3.74 | 3.76 | 0.13 |
| June | 16.25 | 24.04 | 40.04 | 11.71 | 24.76 | 55.91 | 6.63 | 6.95 | 36.06 |
| July | 9.71 | 15.10 | 22.05 | 4.20 | 4.46 | 43.31 | 4.09 | 9.99 | 24.74 |
| August | 8.21 | 12.96 | 14.40 | 9.86 | 8.06 | 38.45 | 13.70 | 10.67 | 16.47 |
| September | 6.82 | 10.67 | 2.38 | 4.30 | 18.36 | 25.11 | 9.12 | 10.57 | 15.63 |
| October | 6.01 | 5.58 | 0.18 | 12.47 | 13.05 | 0.00 | 5.44 | 5.66 | 0.31 |
| November | 6.00 | 4.98 | 1.02 | 8.64 | 5.43 | 0.00 | 7.55 | 6.57 | 0.00 |
| December | 0.30 | 0.08 | 0.00 | 0.00 | 0.00 | 0.00 | 0.00 | 0.00 | 0.00 |

**Table 3. Annual and seasonal temperature trends (°C 10a$^{-1}$) in the CTM (based on the CTMD) and the whole continental China (WCC) and low-altitude areas (LCC) (both based on the CMA05) from 1979 to 2016.**

| | CTM | | | WCC | | | LCC | | |
|---|---|---|---|---|---|---|---|---|---|
| | Tmin | Tmean | Tmax | Tmin | Tmean | Tmax | Tmin | Tmean | Tmax |
| Spring | 0.633 *** | 0.522 *** | 0.640 *** | 0.557 *** | 0.513 *** | 0.518 *** | 0.543 *** | 0.498 *** | 0.505 *** |
| Summer | 0.441 *** | 0.342 *** | 0.266 ** | 0.472 *** | 0.388 *** | 0.378 *** | 0.404 *** | 0.336 *** | 0.348 *** |
| Autumn | 0.302 | 0.200 * | 0.270 | 0.551 *** | 0.458 *** | 0.420 *** | 0.506 *** | 0.411 *** | 0.371 *** |
| Winter | 0.014 | -0.085 | 0.115 | 0.432 *** | 0.361 *** | 0.327 *** | 0.333 ** | 0.257 | 0.211 |
| Annual | 0.347 *** | 0.245 *** | 0.323 *** | 0.503 *** | 0.430 *** | 0.411 *** | 0.446 *** | 0.376 *** | 0.359 *** |

Note: the bold and underlined value indicates a greater warming trend in the CTM than WCC and LCC. * denotes the significance level $p<0.1$, ** denotes the significance level $p<0.05$, and *** denotes the significance level $p<0.01$.

**Table 4. Monthly temperature trends (°C 10a$^{-1}$) in the CTM (based on the CTMD) and the whole continental China (WCC) and low-elevation areas (LCC) by excluding the CTM and the QTP from the WCC (both based on the CMA05) from 1979 to 2016.**

| | CTM | | | WCC | | | LCC | | |
|---|---|---|---|---|---|---|---|---|---|
| | Tmin | Tmean | Tmax | Tmin | Tmean | Tmax | Tmin | Tmean | Tmax |
| January | -0.133 | -0.269 | -0.235 | 0.343[**] | 0.256 | 0.212 | 0.225 | 0.143 | 0.102 |
| February | 0.313 | 0.177 | **0.605**[**] | 0.558[***] | 0.523[***] | 0.549[**] | 0.486[**] | 0.456[*] | 0.475[*] |
| March | **0.835**[**] | **0.818**[***] | **1.339**[***] | 0.651[***] | 0.672[***] | 0.752[***] | 0.661[***] | 0.673[***] | 0.738[***] |
| April | 0.441 | **0.537**[***] | **0.664**[*] | 0.547[***] | 0.522[***] | 0.516[***] | 0.520[***] | 0.503[***] | 0.508[***] |
| May | **0.624**[**] | 0.211 | -0.082 | 0.475[***] | 0.345[***] | 0.284[***] | 0.447[***] | 0.317[***] | 0.270[***] |
| June | **0.752**[***] | **0.476**[***] | **0.422**[***] | 0.516[***] | 0.390[***] | 0.344[***] | 0.467[***] | 0.348[***] | 0.320[***] |
| July | 0.227 | 0.331[***] | 0.28 | 0.472[***] | 0.411[***] | 0.416[***] | 0.402[***] | 0.343[***] | 0.359[***] |
| August | 0.342 | 0.217[*] | 0.095 | 0.429[***] | 0.363[***] | 0.375[***] | 0.343[***] | 0.318[***] | 0.363[***] |
| September | 0.246 | 0.237 | 0.33 | 0.559[***] | 0.486[***] | 0.495[***] | 0.517[***] | 0.445[***] | 0.456[***] |
| October | 0.273 | 0.18 | 0.227 | 0.524[***] | 0.434[***] | 0.398[***] | 0.496[***] | 0.407[***] | 0.372[**] |
| November | 0.386 | 0.183 | 0.252 | 0.569[***] | 0.455[***] | 0.368[**] | 0.503[***] | 0.381[**] | 0.285 |
| December | -0.137 | -0.164 | -0.025 | 0.394[***] | 0.303[**] | 0.219 | 0.287[*] | 0.171 | 0.055 |

Note: the bold and underlined value indicates a greater warming trend in the CTM than WCC and LCC. [*] denotes the significance level $p<0.1$, [**] denotes the significance level $p<0.05$, and [***] denotes the significance level $p<0.01$.

**Table 5. Significance of elevation-dependent warming for monthly Tmin above different altitude thresholds based on the CTMD from 1979 to 2016.**

| | All altitude | > 2000 m | > 2500 m | > 3000 m | > 3500 m | > 4000 m | > 4500 m |
|---|---|---|---|---|---|---|---|
| January | 0.039*** | 0.040*** | 0.046*** | 0.049*** | 0.049*** | 0.040** | 0.023** |
| February | 0.033*** | 0.020*** | 0.021*** | 0.025*** | 0.028*** | 0.030*** | 0.029* |
| March | 0.023 | -0.014* | -0.019** | -0.018* | -0.013 | 0.0 | 0.018 |
| April | 0.021*** | 0.027*** | 0.031*** | 0.032*** | 0.026** | 0.014 | -0.002 |
| May | -0.056*** | -0.055*** | -0.054*** | -0.056*** | -0.055*** | -0.045** | -0.022 |
| June | -0.025*** | -0.050*** | -0.057*** | -0.060*** | -0.059*** | -0.047** | -0.025** |
| July | 0.0 | 0.008 | 0.015*** | 0.019*** | 0.022*** | 0.027*** | 0.034*** |
| August | -0.011 | -0.043*** | -0.047*** | -0.050*** | -0.045** | -0.027 | -0.003 |
| September | -0.006 | -0.050*** | -0.061*** | -0.065*** | -0.064** | -0.051** | -0.031*** |
| October | -0.073*** | -0.104*** | -0.109*** | -0.112*** | -0.104** | -0.074* | -0.026 |
| November | -0.032*** | -0.045*** | -0.050*** | -0.055*** | -0.057*** | -0.048** | -0.031** |
| December | 0.064*** | 0.082*** | 0.092*** | 0.096*** | 0.093*** | 0.074** | 0.044** |

Note: the positive value indicates the elevation-dependent warming trend while the negative value indicates the elevation-dependent cooling trend. * denotes the significance level $p<0.1$, ** denotes the significance level $p<0.05$, and *** denotes the significance level $p<0.01$.

**Table 6. Significance of elevation-dependent warming for monthly Tmean above different altitude thresholds based on the CTMD from 1979 to 2016.**

| | All altitude | > 2000 m | > 2500 m | > 3000 m | > 3500 m | > 4000 m | > 4500 m |
|---|---|---|---|---|---|---|---|
| January | 0.036[***] | 0.041[***] | 0.046[***] | 0.047[***] | 0.044[***] | 0.033[**] | 0.016[**] |
| February | 0.012[***] | 0.006[***] | 0.007[***] | 0.009[***] | 0.009[***] | 0.008[***] | 0.008[*] |
| March | 0.009[**] | 0.002[**] | 0.002[***] | 0.003[**] | 0.003[**] | 0.004[**] | 0.005[*] |
| April | -0.020[***] | -0.007 | -0.002 | 0.001 | 0.003 | 0.007 | 0.012[**] |
| May | -0.022[***] | -0.017[***] | -0.015[***] | -0.016[***] | -0.017[***] | -0.018[***] | -0.015[**] |
| June | 0.007 | 0.015[***] | 0.018[***] | 0.021[**] | 0.025[**] | 0.034[***] | 0.045[***] |
| July | -0.017[**] | -0.022[***] | -0.020[**] | -0.020[*] | -0.017 | -0.004 | 0.015 |
| August | 0.007 | 0.002 | 0.006 | 0.009 | 0.014 | 0.024[*] | 0.037[***] |
| September | 0.003 | -0.002 | -0.002 | -0.002 | 0.0 | 0.007 | 0.017[**] |
| October | -0.018[***] | -0.018[***] | -0.018[***] | -0.018[***] | -0.017[**] | -0.012[*] | -0.004 |
| November | -0.031[***] | -0.025[***] | -0.022[***] | -0.020[***] | -0.018[**] | -0.013[*] | -0.005 |
| December | -0.001 | 0.002 | 0.004[***] | 0.006[***] | 0.006[**] | 0.007[*] | 0.008[*] |

Note: the positive value indicates the elevation-dependent warming trend while the negative value indicates the elevation-dependent cooling trend. [*] denotes the significance level $p<0.1$, [**] denotes the significance level $p<0.05$, and [***] denotes the significance level $p<0.01$.

**Table 7. Significance of elevation-dependent warming for monthly Tmax above different altitude thresholds based on the CTMD from 1979 to 2016.**

| | All altitude | > 2000 m | > 2500 m | > 3000 m | > 3500 m | > 4000 m | > 4500 m |
|---|---|---|---|---|---|---|---|
| January | 0.037*** | 0.050*** | 0.057*** | 0.059*** | 0.054** | 0.036* | 0.013* |
| February | 0.008 | 0.010* | 0.017*** | 0.021*** | 0.023*** | 0.019* | 0.008 |
| March | 0.017*** | 0.027*** | 0.026*** | 0.025*** | 0.023*** | 0.023*** | 0.024*** |
| April | 0.002 | 0.055*** | 0.069*** | 0.077*** | 0.085*** | 0.090*** | 0.090*** |
| May | -0.045*** | -0.046*** | -0.047*** | -0.053*** | -0.062*** | -0.073*** | -0.079*** |
| June | -0.046*** | -0.069*** | -0.074*** | -0.081*** | -0.085*** | -0.075** | -0.048** |
| July | -0.019** | -0.002 | 0.006* | 0.010** | 0.011** | 0.013* | 0.016 |
| August | 0.010 | 0.007 | 0.014*** | 0.019*** | 0.022*** | 0.023*** | 0.023** |
| September | -0.004 | 0.005 | 0.007 | 0.008 | 0.012 | 0.022* | 0.038*** |
| October | 0.006 | 0.001 | 0.004 | 0.007* | 0.008* | 0.010 | 0.017** |
| November | -0.018*** | -0.021*** | -0.021*** | -0.023*** | -0.026*** | -0.028*** | -0.026*** |
| December | -0.018*** | -0.025*** | -0.022*** | -0.022*** | -0.025*** | -0.026** | -0.017** |

Note: the positive value indicates the elevation-dependent warming trend while the negative value indicates the elevation-dependent cooling trend. * denotes the significance level $p<0.1$, ** denotes the significance level $p<0.05$, and *** denotes the significance level $p<0.01$.

**Table 8. Relationship ($R^2$) of maximum snow cover fraction (%) and monthly Tmin, Tmean and Tmax from 2002 to 2013.**

| | Tmin | Tmean | Tmax |
|---|---|---|---|
| January | 0.086 | 0.024 | 0.117 |
| February | 0.302 * | 0.038 | 0.009 |
| March | 0.005 | 0.073 | 0.102 |
| April | 0.075 | 0.089 | 0.060 |
| May | 0.162 | 0.000 | 0.012 |
| June | 0.025 | 0.096 | 0.012 |
| July | 0.144 | 0.158 | 0.161 |
| August | 0.033 | 0.036 | 0.001 |
| September | 0.019 | 0.186 | 0.003 |
| October | 0.003 | 0.001 | 0.001 |
| November | 0.060 | 0.097 | 0.017 |
| December | 0.002 | 0.017 | 0.003 |

Note: * denotes the significance level $p < 0.1$.

**Table 9. Relationship ($R^2$) of minimum snow cover fraction (%) and monthly Tmin, Tmean and Tmax from 2002 to 2013.**

| | Tmin | Tmean | Tmax |
|---|---|---|---|
| January | 0.181 | 0.092 | 0.093 |
| February | 0.198 | 0.320 | 0.073 |
| March | 0.171 | 0.153 | 0.068 |
| April | 0.106 | 0.118 | 0.006 |
| May | 0.031 | 0.296 * | 0.043 |
| June | 0.085 | 0.244 | 0.020 |
| July | 0.246 | 0.006 | 0.019 |
| August | 0.000 | 0.156 | 0.256 |
| September | 0.004 | 0.081 | 0.043 * |
| October | 0.056 | 0.026 | 0.022 |
| November | 0.001 | 0.024 | 0.009 |
| December | 0.001 | 0.011 | 0.003 |

Note: * denotes the significance level $p < 0.1$.

**Table 10. Relationship ($R^2$) of snow depth (cm) and monthly Tmin, Tmean and Tmax from 1979 to 2016.**

| | Tmin | Tmean | Tmax |
|---|---|---|---|
| January | 0.021 | 0.098 * | 0.109 ** |
| February | 0.031 | 0.050 | 0.103 * |
| March | 0.399 *** | 0.400 *** | 0.033 |
| April | 0.003 | 0.076 | 0.008 |
| May | 0.086 * | 0.104 * | 0.012 |
| June | 0.194 *** | 0.230 *** | 0.095 * |
| July | 0.081 * | 0.108 * | 0.016 |
| August | 0.047 | 0.242 *** | 0.083 * |
| September | 0.001 | 0.072 | 0.150 ** |
| October | 0.010 | 0.020 | 0.103 * |
| November | 0.051 | 0.125 ** | 0.151 ** |
| December | 0.014 | 0.159 ** | 0.200 *** |

Note: * denotes the significance level $p < 0.1$, ** denotes the significance level $p < 0.05$, and *** denotes the significance level $p < 0.01$.

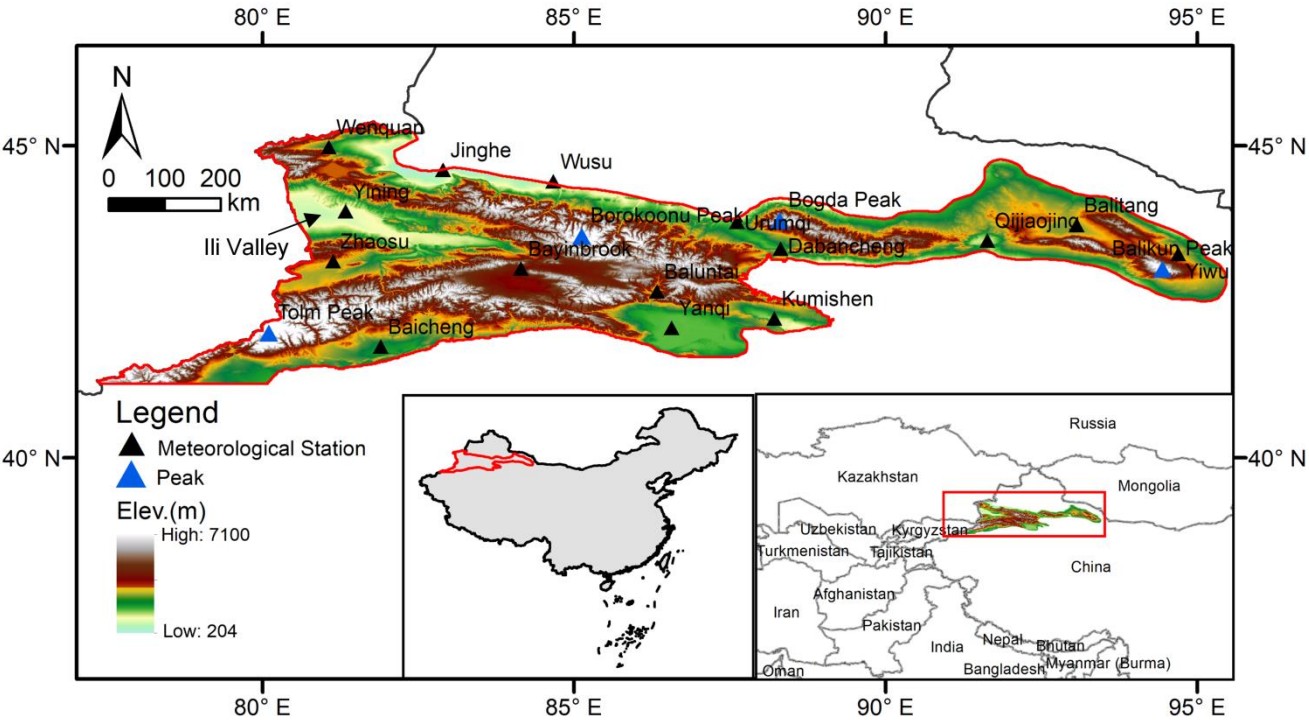

**Figure 1: Location of the Chinese Tianshan Mountains (CTM).The elevation ranges from 204 m to 7100 m a.s.l., with a DEM resolution of 1 km from SRTM. The grey sub-plot show the extent of the CMA05 at the 0.5 °×0.5 °grid.**

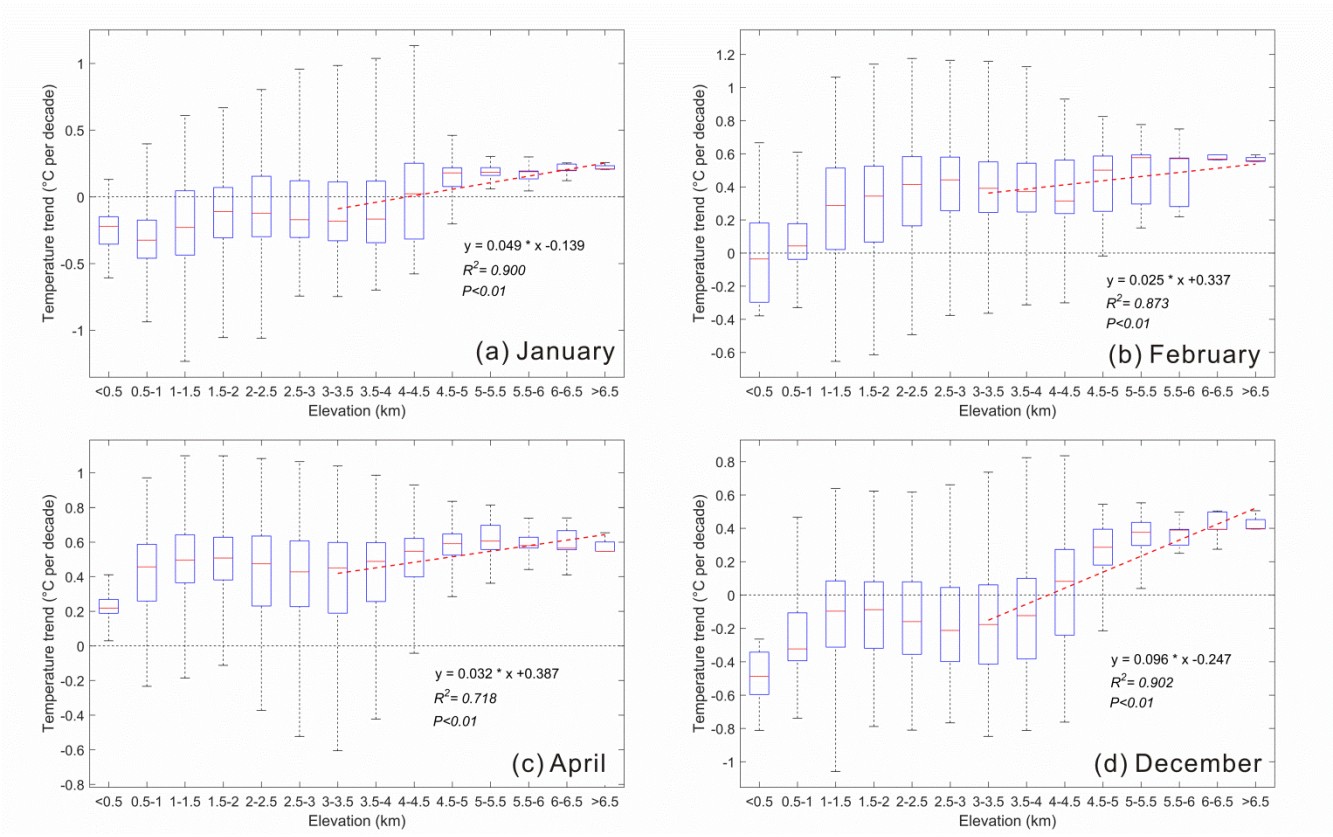

**Figure 2: Box plots of monthly minimum temperature trends at different altitudes from 1979 to 2016. (a) January, (b) February, (c) April, and (d) December. Thick horizontal lines in boxes show the median values. Boxes indicate the inner-quantile range (25% to 75%) and the whiskers show the full range of the values. The red dashed lines represent the significance of EDW by fitting the temperature trends and altitude groups above 3000 m.**

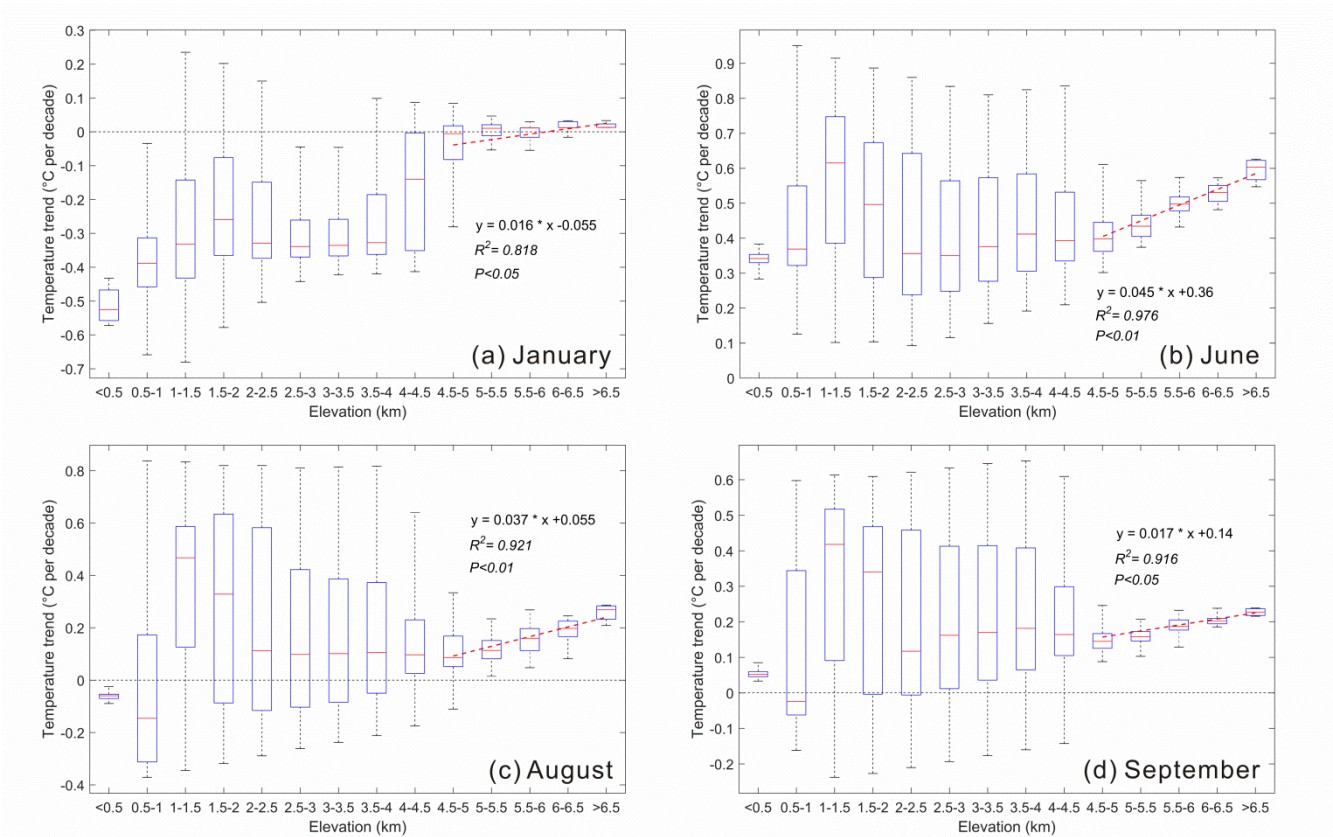

Figure 3: Box plots of monthly mean temperature trends at different altitudes from 1979 to 2016. (a) January, (b) June, (c) August, and (d) September. Thick horizontal lines in boxes show the median values. Boxes indicate the inner-quartile range (25% to 75%) and the whiskers show the full range of the values. The red dashed lines represent the significance of EDW by fitting the temperature trends and altitude groups above 4500 m.

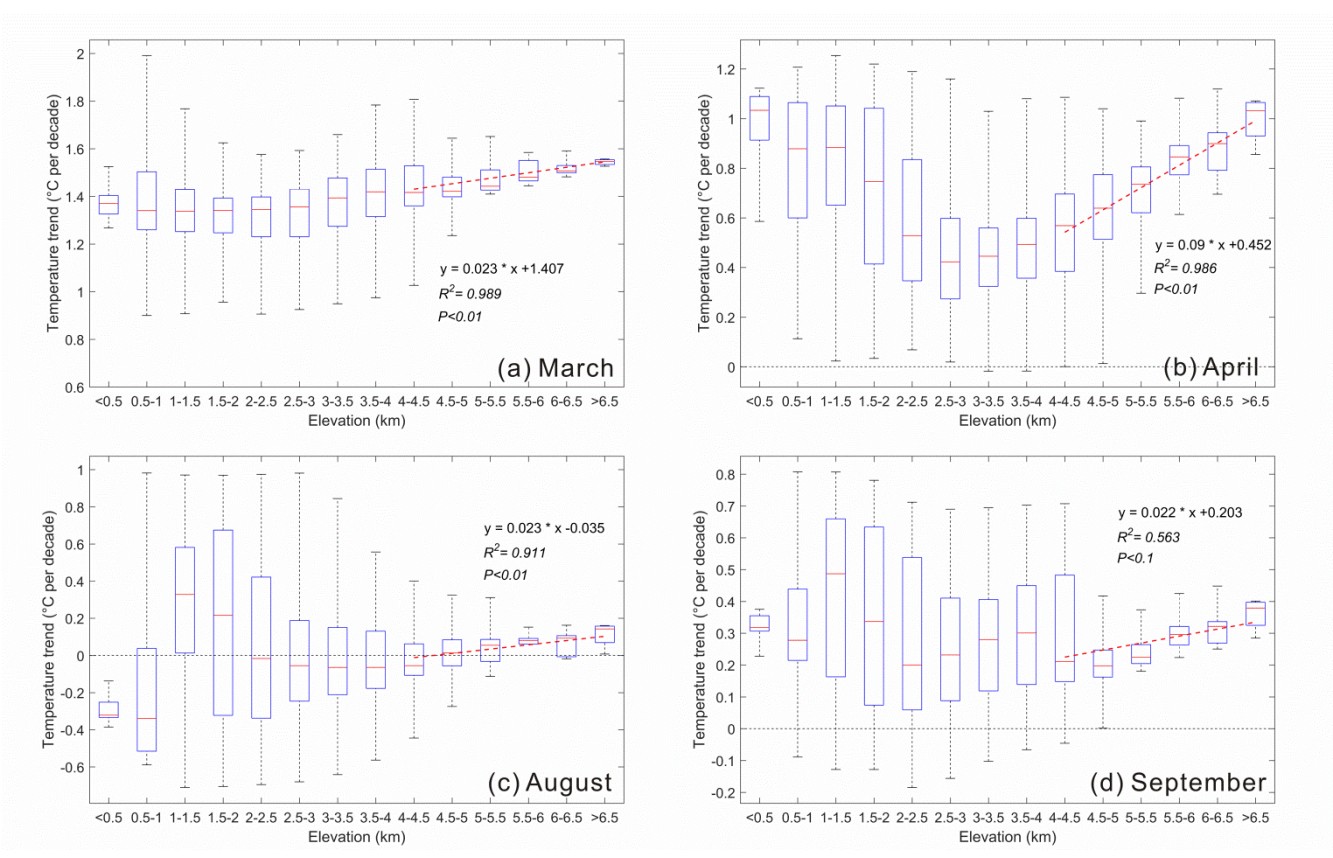

**Figure 4: Box plots of monthly maximum temperature trends at different altitudes from 1979 to 2016. (a) March, (b) April, (c) August, and (d) September. Thick horizontal lines in boxes show the median values. Boxes indicate the inner-quartile (25% to 75%) and the whiskers show the full range of the values. The red dashed lines represent the significance of EDW by fitting the temperature trends and altitude groups above 4000 m.**

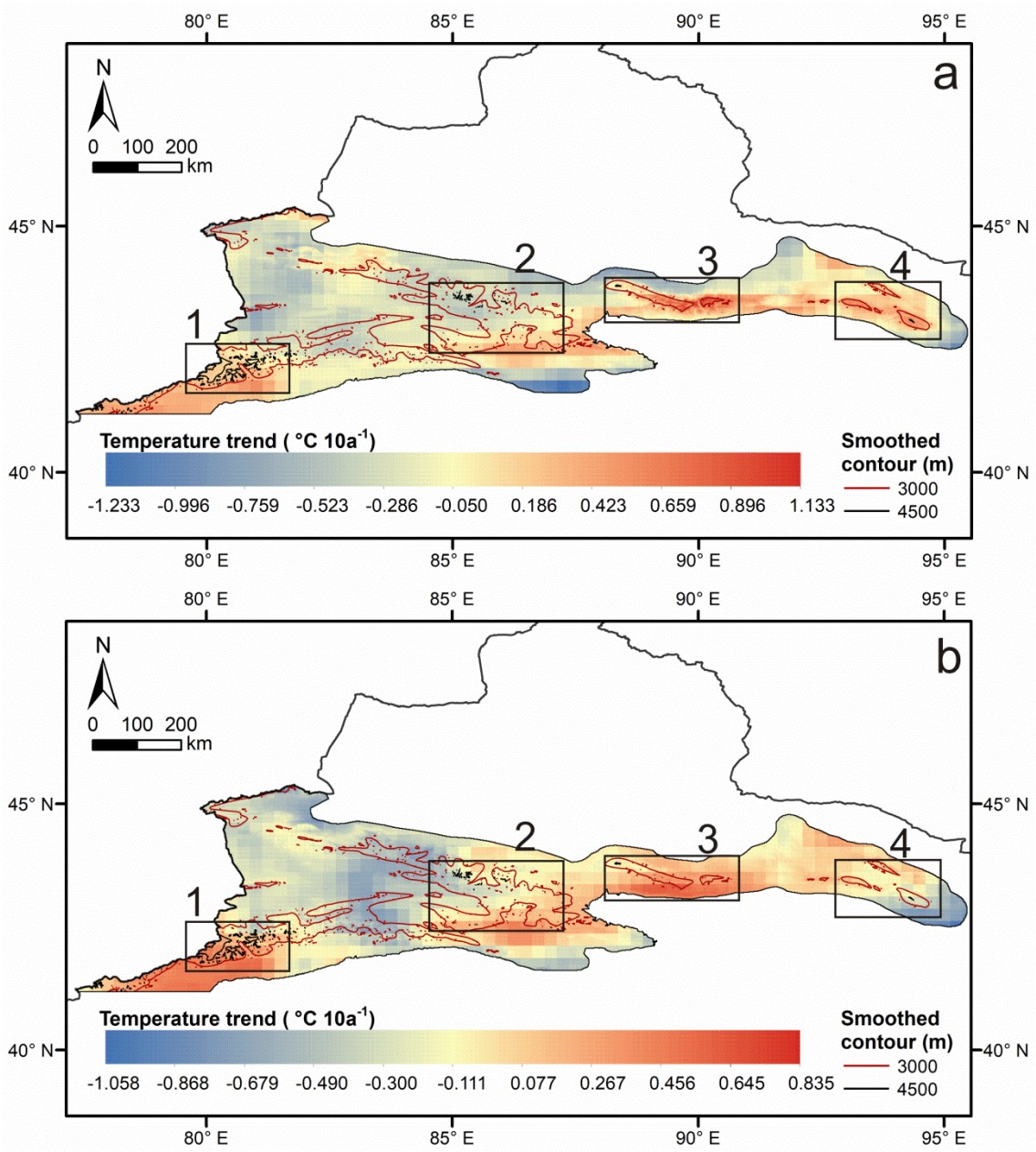

**Figure 5: Monthly minimum temperature trends (a) January and (b) December for the entire CTM from 1979 to 2016.**

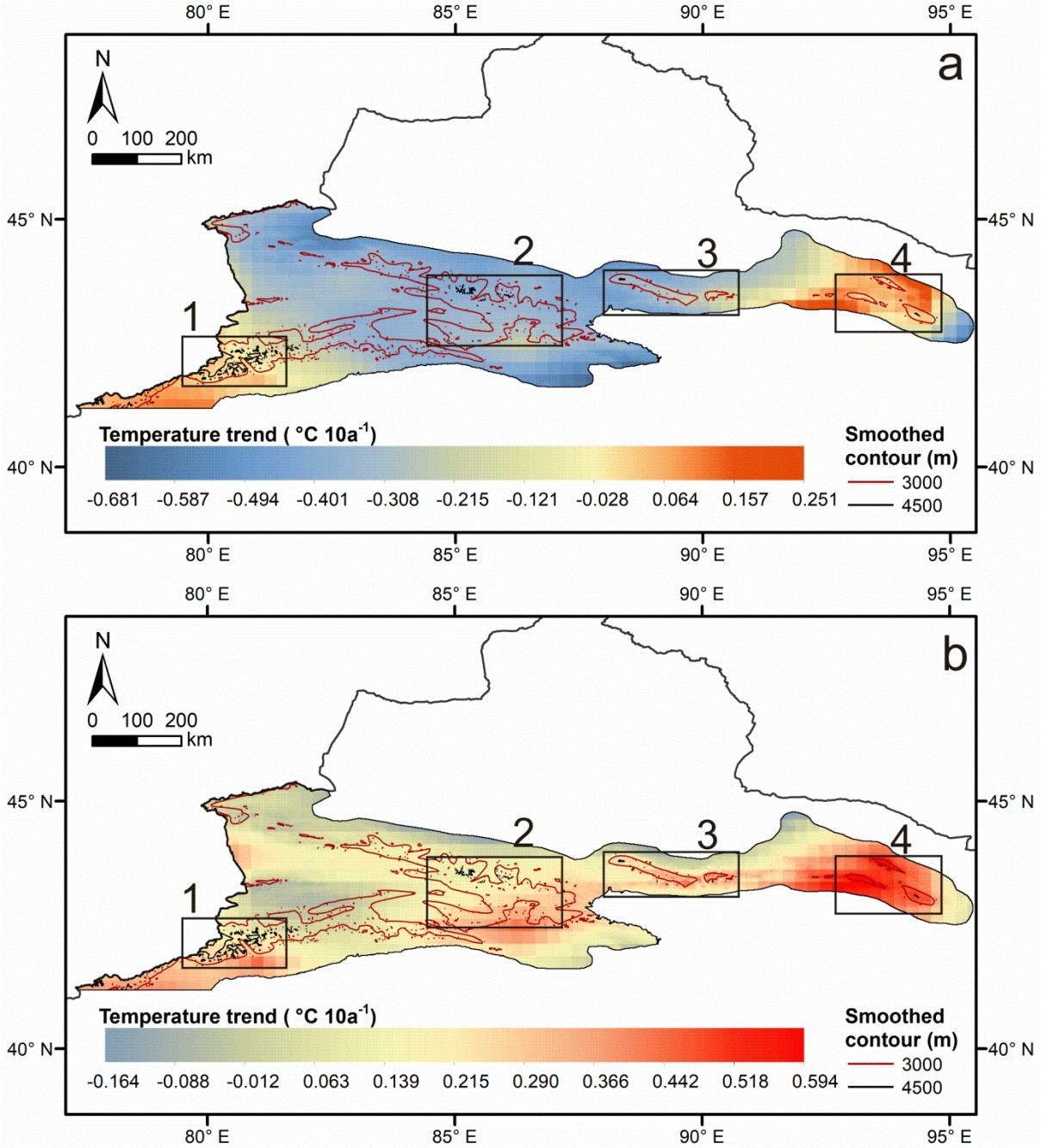

Figure 6: Monthly mean temperature trends (a) January and (b) February for the entire CTM from 1979 to 2016.

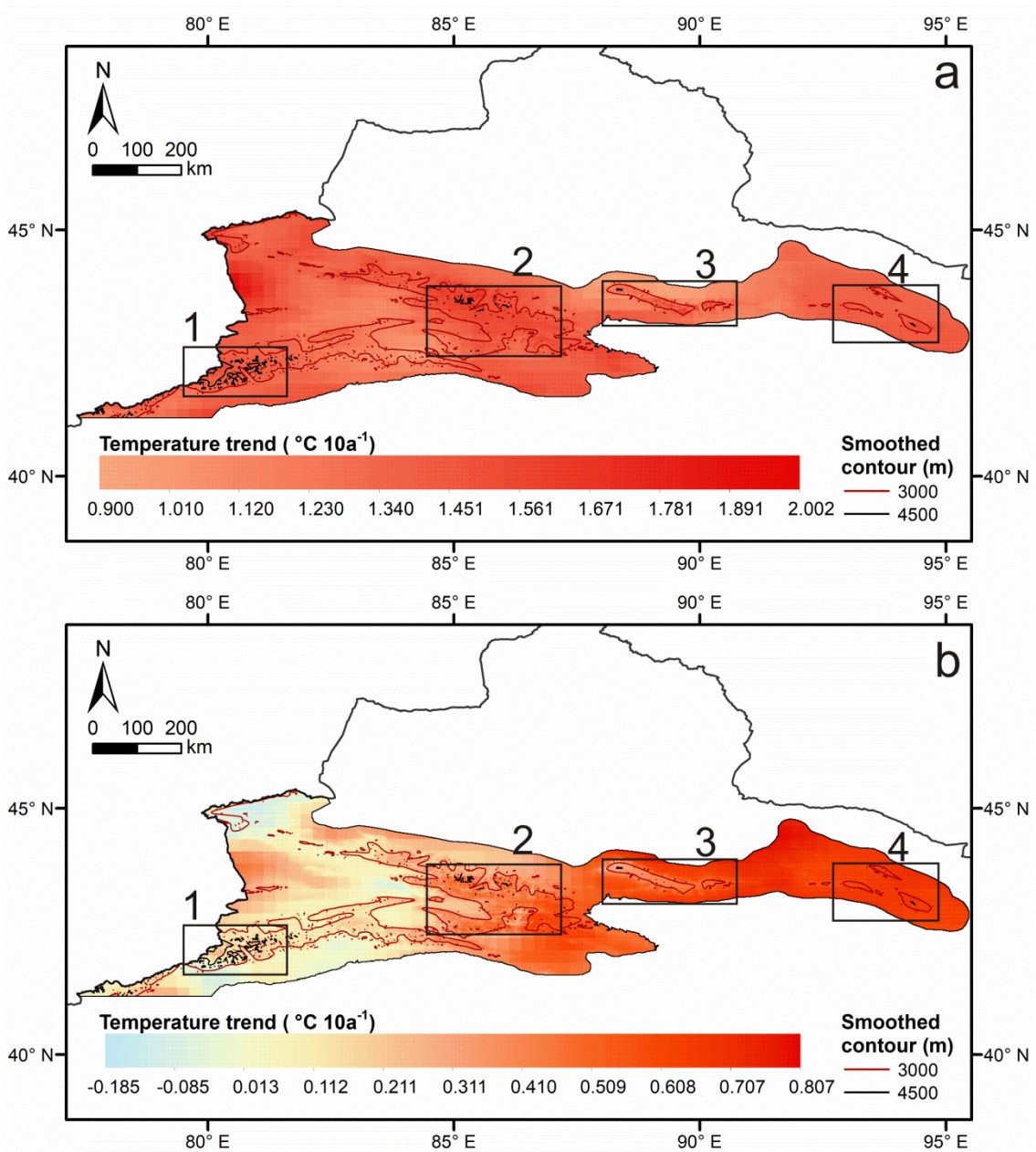

**Figure 7: Monthly maximum temperature trends (a) March and (b) September for the entire CTM from 1979 to 2016.**

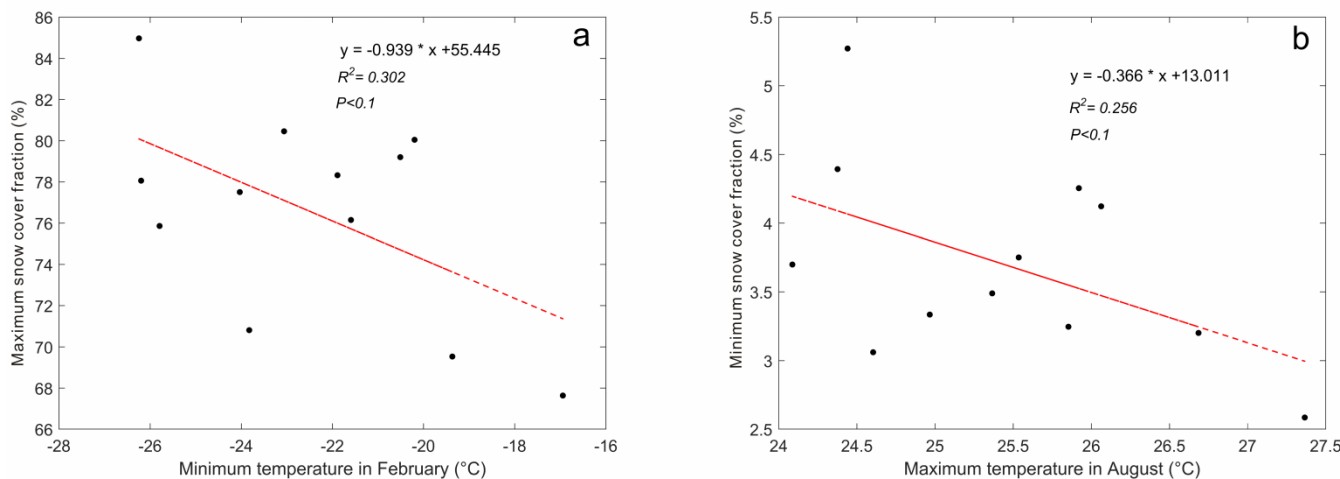

**Figure 8: Relationship of temperature and snow cover fraction (a) minimum temperature in February vs. maximum snow cover fraction and (b) maximum temperature in August vs. minimum snow cover fraction from 2002 to 2013.**

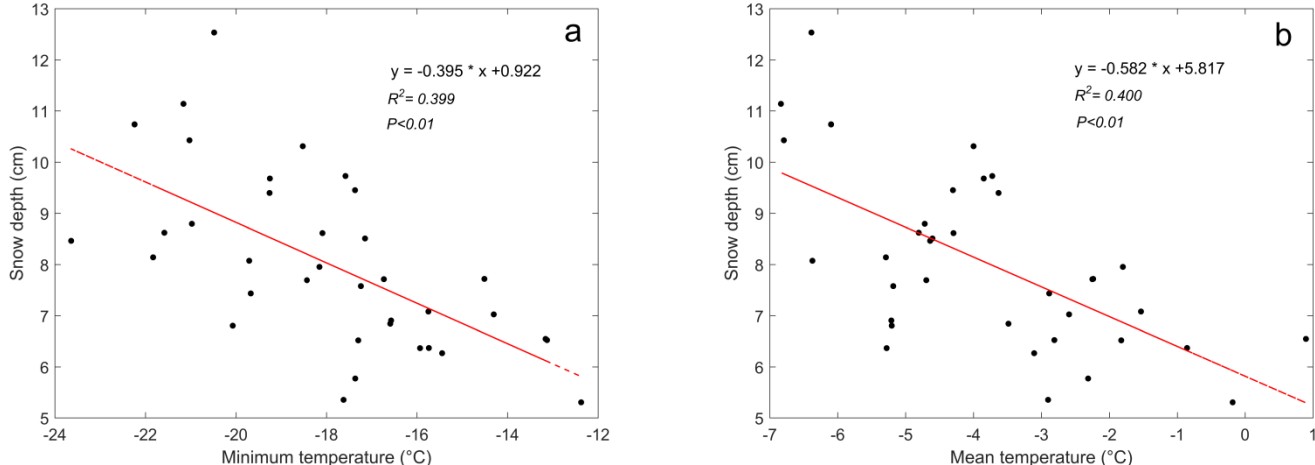

**Figure 9: Relationship of snow depth and (a) Tmin in March and (b) Tmean in March from 1979 to 2016.**