# Peer review of "Evidence of elevation-dependent warming from the Chinese Tianshan Mountains"

_The Cryosphere, 2020_

## Short Comment (SC1) · 13 Aug 2020

This paper is generally well-structured and well-described for all the parts. The authors detect EDW features in the Chinese Tianshan Mountains (CTM) using a unique high-resolution (1 km, 6-hourly) air temperature data set (CTMD). Authors provide an interesting study under the global warming.

---

## Short Comment (SC2) · 13 Aug 2020

Many studies have suggested EDW phenomena in mountain regions of the world based on the multi-source data and climate models. However, the phenomena still lack systematic detected in the Tianshan Mountains, as a "water tower" of Central Asia. This study systematically investigated the EDW characteristics for different indicators on different time scales in Tianshan Mountains using a unique high-resolution dataset. It is worth mentioning that authors provide a multi-scale perspective of the EDW phenomena.

---

## Short Comment (SC3) · 25 Aug 2020

Many thanks to Dr. Jia. Thanks a lot for your interest in our work. We also appreciate your positive comments.

At present, there are few studies on the elevation-dependent warming in the Chinese Tianshan Mountains (CTM). We hope the present research can provide a reference for other researchers. We also expect this research to help others better understand mountain climate change.

Thanks a lot for your comment again.
* * *

---

## Short Comment (SC4) · 25 Aug 2020

Many thanks to Dr. Yao. We appreciate your positive comments.

Dr. Yao pointed out a very important issue, which is the EDW detection at monthly scale. However, most previous studies focused on the seasonal EDW features. The unique high-resolution temperature dataset allows us to detect the EDW phenomenon at multi-temporal scales. Thus, we reveal the significant EDW features during the transition of seasons. This may explain the accelerated melting of glaciers and snow in the CTM. We expect this research to help others better understand the EDW in high mountains.

Thanks a lot for your comment again.

---

## Short Comment (SC5) · 29 Aug 2020

Climate change likely alters regional hydrologic regime and affects access to water resources. Using a unique high-resolution (1 km, 6-hourly) air temperature data set (CTMD) produced by previous studies, this work investigated the existence of the elevation-dependence warming (EDW) in the Chinese Tianshan Mountains (CTM) at the monthly, seasonal, and annual time scales, which is of significance for understanding glacier melting and water resources management in the vast arid area around CTM. The authors found that EDW was significant at different altitudes on different time scales. Overall, the present study is an interesting forward attempt to deepen our understanding of EDW phenomenon. Few queries to the authors, which might be helpful to improve this manuscript are given below:

[Figure]

1. In sections 3.1 and 3.2, the authors provided the significance of the linear regression models (equation (1)), it would also be helpful to report if the trends (coefficient alpha) were statistically different from zero (or to report the confidence intervals of the trends).

2. Please explain the definition of seasons by grouping months in CTM region (this may be helpful for international readers).

3. Please elaborate a bit on how the trends for CTM (Table 1 &2) and China were calculated. If my understanding were correct, the authors applied equation (1) to estimate the trend for each pixel, but I am not clear how they aggregated the pixel-scale trends to the trends for CTM.

---

## Referee Comment (RC1) · Anonymous Referee #1 · 30 Aug 2020

This study intends to reveal EDW in the Chinese Tianshan Mountains using a high-resolution data that are developed in the previous study based on ERA-I data in combination with topographic correction method. Despite merits such as clear structure and better writing to be easy to follow, I have three comments in the following:

(1) My major concern is the accuracy of data used. This paper does not do a detailed introduce to the high-resolution data, which results in that I cannot evaluate its accuracy or reliability. After a look at the reference provided, it shows that the high-resolution data are based on ERA-I reanalysis. ERA-I is developed based on model simulation in addition to weather station observations, so it generally has large uncertainties in such a small region, especially for mountainous region. Because ERA-I includes in suit observations at some weather stations, it may be unsurprised there

are very seasonable performance for evaluation using observed data from perhaps the same weather stations.

(2) This paper discusses the mechanism only, if data can be used to reveal some mechanism in the research region, it will be a better progress. The mechanism discussed may be suitable for other regions, but is not always in the case for the research region in the present study.

(3) Some expressions are not very rigorous. Such as Line 83-85, the author say that satellite data have low spatial resolution, which is questionable. Some satellite data with 1 km resolution are the same resolution as data used in this study. The author also say large system errors with satellite data, which needs analyses or references to confirm.

---

## Referee Comment (RC2) · Anonymous Referee #2 · 1 Sep 2020

General comments:

This article analyses whether elevation-dependent warming (EDW) is present in the Chinese Tianshan Mountains, both overall and at a regional level. The authors present a compelling case for research into this phenomenon, as increased warming in higher regions may have detrimental effects on glacier melt. EDW is judged based on the criteria of regional warming amplification and altitude warming amplification, and these two criteria are assessed for the entirety of the Chinese Tianshan Mountains on a monthly timescale. Furthermore, spatial differences in EDW are assessed across the mountain range. Overall, the paper is well presented and structured, and the discussion and conclusions of this spatially and temporally complicated problem are interesting. However, there are some issues which I think need to be addressed before publication,

most importantly the definition of EDW used in the paper and how it relates to the con-clusions reached in the paper, and the suitability of the data set used for this analysis, as highlighted below.

Specific comments:

1. Whole paper: The authors have carefully defined elevation-dependent warming (EDW) immediately in the article, namely that two criteria should be met: regional warming amplification and altitude warming amplification. Section 3.1 concludes that regional warming amplification is only present in any of the minimum, mean and max-imum daily temperatures in the months from February to June. However, in section 3.2, warming amplification with altitude is now described as EDW, for example line 183 "The prevalence of EDW is most significant in December...". This is then used for the remainder of the paper, especially in the conclusions. The authors should identify the months which satisfy both regional warming and altitude warming amplification, and these months should be set out clearly as the months where EDW is present.

This needs to be altered throughout the paper, and has substantial implications for the conclusions, as I think there are only one or two months which satisfy both conditions.

2. Methods/CTMD dataset: I think there should be some discussion of the suitability and limitation of the CTMD dataset for this analysis, given that the paper is reliant on it. Two particular points stand out:

o Gao et al., 2018 gives an analysis of the data set compared to a number of stations; however they are all under 3000 m asl. I do appreciate the difficulty of finding high-elevations stations, but do the authors have any evidence that this data set is suitable at elevations of 5000 m asl and above? In addition, Gao et al., 2018 also indicates that the lapse rate from ERA-interim (the correction term used to downscale ERA-Interim to the 1 km scale) is steeper than that seen in the observations. It is often the case that the free atmosphere lapse rate is steeper than the near-ground lapse rate of temperature with elevation, and this difference may cause errors in the 1 km data set used in this

paper.

o Gao et al., 2018 acknowledge that the trends in the ERA-Interim data, and therefore the CTMD, do not always follow those of the observations. For example, in the minimum daily temperature, the trend in the CTMD considerably underestimates that of the observations. It is not clear whether this bias is constant with elevation, which is essential to the results presented in this manuscript.

3. Table 1 and 2: given the variation over time, it would be useful to know which of these trends is statistically significant

4. Line 128: How were 6-hourly data aggregated to the minimum and maximum temperature? Was any consideration given to the minimum/maximum temperature not occurring at 00, 6, 12, 18 UTC?

5. Related to points 3 and 4: I'm surprised that in some cases, the warming increase in Tmin and Tmax are both greater than the warming increase in Tmean. This suggests some unusual shift in the shape of the diurnal cycle. Could the authors hypothesise as to why this might be?

6. Section 3.3: This analysis of the spatial variations is interesting, and Figures 5-7 quite well represent the first requirement for EDW, that the warming in the region is greater than the surrounding area. However, it is difficult to see the altitude warming amplification from these plots unless you are well-acquainted with the topography of the region (e.g. from figure 5b it's only really possible to see a north-south gradient in area 1, it's not clear that that corresponds with high-low). Would it be possible to add (small) plots such as those in figures 2-4 to figures 5-7 for each region? If it's not possible to fit the graphs on, perhaps the trends and significance could be calculated, such as in figure 2-4? As in point 1, only those areas which fit both criteria should be described as EDW.

Smaller remarks, technical comments and suggestions:

[Figure]

7. Figure 1: Does the bottom right hand corner map show the extent of the CMA05 used in this analysis? If so, please add to the caption. If not, could this be altered to show the CMA05 extent?

8. Introduction: It would be useful to make clear earlier on in the paper that EDW is referring to the rate of warming over a multi-annual scale (rather than, say, rate of warming during the day). This is made clear on line 58 with 'warming trend of annual mean temperature' but could be mentioned earlier.

9. Around line 120 onwards-perhaps mention that the topography comes from SRTM.

10. Line 136-137: it might be sensible to combine the highest two elevation bands, given that the highest only contains 4 points (which may not be representative in general).

11. Line 268-270: I think this sentence can be removed as you're only talking about surface albedo here.

12. Line 113: remove 'because' (either 'because the system. . ..,the bias could be' or 'The system bias of. . .. Thus, the bias. . .'

13. Please consider changing the colour bars in Figs. 5 to 7 so that they are all the same (and ideally centred around 0, so that red is positive, blue is negative and yellow around zero). At first glance it seems that the maximum temperature trends in March have both positive and negative values, which as you point out in the text is not the case. In addition, please flip the colour bars so that negative values are on the left and positive on the right.

14. Give the location of the Ili valley where it first appears on line 208, rather than 210.

References Gao, L., Wei, J., Wang, L., Bernhardt, M., Schulz, K., and Chen, X.: A high-resolution air temperature data set for the Chinese Tian Shan in 1979–2016, Earth System Science Data, 10, 2097-2114, 2018.

---

## Author Comment (AC1) · 10 Sep 2020

**Response to Referee 1**

We would like to thank anonymous Referee 1 for reviewing our manuscript. These professional comments are really helpful for improving the manuscript. In the following, we address all comments point-by-point according to referee's comments.

This study intends to reveal EDW in the Chinese Tianshan Mountains using a high resolution data that are developed in the previous study based on ERA-I data in combination with topographic correction method. Despite merits such as clear structure and better writing to be easy to follow, I have three comments in the following:

1. My major concern is the accuracy of data used. This paper does not do a detailed introduce to the high-resolution data, which results in that I cannot evaluate its accuracy or reliability. After a look at the reference provided, it shows that the high resolution data are based on ERA-I reanalysis. ERA-I is developed based on model simulation in addition to weather station observations, so it generally has large uncertainties in such a small region, especially for mountainous region. Because ERA-I includes in suit observations at some weather stations, it may be unsurprised there are very seasonable performance for evaluation using observed data from perhaps the same weather stations.

-Answer: Thanks a lot for the comments. The referee pointed out a very important issue on the accuracy of data, which is the foundation of presented study. That is true that we just provided limited information on the data set because we leave more space to EDW analysis. We accept the comment and will add more data set information in the revision.

Here, we would like to introduce the data set briefly. It is true that the data set

is produced based on ERA-Interim data and an elevation correction method. We also agree that the uncertainty is large for original ERA-Interim. Our previous studies revealed that there are around 3-4 °C systematic bias from original ERA-Interim (Gao et al., 2014, 2017). Thus, a correction is necessary before local application. The correction approach based on the internal lapse rates derived from ERA-Interim has been proven to be effective in the mountains. Although, there is still a less than 2 °C bias after elevation correction, the warming trends could be captured very well (Gao et al., 2018, Table 1). 24 meteorological stations are applied for data set validation in the CTM from 1979-2013. The averaged trend difference between observation and CTMD is only 0.07 °C $10a^{-1}$ respects to annual and seasonal temperatures. Although the CTMD tends to underestimate the trends for minimum temperatures, we still believe that CTMD is reliable to capture the EDW trend.

Table 1. Trends (°C $10a^{-1}$) of annual and seasonal temperatures over the 24 sites in 1979-2013.

|             | Annual | Spring | Summer | Autumn | Winter |
|-------------|--------|--------|--------|--------|--------|
| observation | 0.420  | 0.664  | 0.432  | 0.532  | 0.018  |
| ERA-Interim | 0.378  | 0.659  | 0.530  | 0.448  | -0.153 |
| correction  | 0.349  | 0.638  | 0.478  | 0.443  | -0.195 |

Meanwhile, we would like to emphasize that the CTM is not a small mountain region (larger than 350,000 $km^2$) which only has less than 30 meteorological stations. Most of them are located in the piedmont plains or valleys. Thus, the validation based on limited surface meteorological station may be not objective. We also know that the analysis of future climate change scenarios relies on model data such as GCM outputs, which have large uncertainties. However, the GCM models are still the most powerful tool for climate change analysis, and the trends modeled by the GCM are still credible. Thus, we believe that although ERA-Interim has errors, it has the ability to reveal regional climate changes after elevation correction. Furthermore, ERA-Interim assimilated

ground observation data, which can more accurately reflect local climate change.

The referee also raised another very important issue that it is unsurprised about the seasonable performance of ERA-Interim at certain stations. Here, we would like to clarify briefly. The ERA-Interim applied ECMWF Integrated Forecast System (IFS) which could assimilate observations in the model. However, only a very small part of observations was assimilated. 9 of 24 sites were possible assimilated by IFS for ERA-Interim in the CTM according to ECMWF assimilation records (Gao et al., 2018, Table 2). Only 4 sites with long-term observations (more than 30 years) while other 5 sites (less than 15 years) were assimilated. In other words, ERA-Interim is a relative independent data set (considering the ratio of ground stations amount to the whole CTM area). We believe the performance of ERA-Interim sometimes is "surprised" in such a complex terrains and it is reliable for regional climate change detection.

Table 2. Possible assimilated sites in the CTM in ERA-Interim.

| Name | WMO id | starting date | ending date |
|---|---|---|---|
| Jinghe | 51334 | 1979-06-21 | 1993-01-21 |
| Qitai | 51379 | 1979-06-03 | 1985-05-20 |
| Yining | 51431 | 1978-12-31 | 2011-12-31 |
| Urumqi | 51463 | 1978-12-31 | 2011-12-31 |
| Qijiaojing | 51495 | 1979-04-07 | 1993-04-24 |
| Turfan | 51573 | 1981-06-30 | 1984-08-08 |
| Kuche | 51644 | 1978-12-31 | 2011-12-31 |
| Kuerle | 51656 | 1979-01-03 | 1994-12-30 |
| Hami | 52203 | 1978-12-31 | 2011-12-31 |

2. This paper discusses the mechanism only, if data can be used to reveal some mechanism in the research region, it will be a better progress. The mechanism discussed may be suitable for other regions, but is not always in the case for the research region in the present study.

-Answer: Thanks a lot for the comments. The referee is definitely right that the mechanism is the key issue for EDW. There are two steps in our research plan.

The first step is detecting EDW phenomenon and selecting typical EDW regions. The second step is the mechanism investigation. We plan to apply WRF model and statistical methods to assess the mechanisms of local physical processes and large-scale circulation, respectively. The WRF model has been widely used for simulating the energy transfer processes at regional scales (~3km). We plan to simulate the surface energy balances under different land surface covers, especially the snow/ice cover effect via WRF model. Besides, we plan to use statistical methods such as SVD, linear regression, lead-lag correlation to assess the impact of atmospheric circulation factors (such as the North Atlantic Oscillation, Arctic Oscillation) on the EDW in typical EDW areas. Thus, we try to comprehensively explore the mechanisms of EDW from large scale (NAO, AO) and local scale (land surface cover). Therefore, the main purpose of this study is to detect the EDW phenomenon, and to prove the existence of EDW in the CTM.

3. Some expressions are not very rigorous. Such as Line 83-85, the author say that satellite data have low spatial resolution, which is questionable. Some satellite data with 1 km resolution are the same resolution as data used in this study. The author also say large system errors with satellite data, which needs analyses or references to confirm.

--Answer: Thanks a lot for pointing this out. We agree that some parts are not very rigorous. We will check the full text and revise them in the subsequent revision.

Gao, L., Hao, L., and Chen, X.W.: Evaluation of ERA-interim monthly temperature data over the Tiberan Plateau, Journal of Mountain Science, 11(5): 1154-1168, 2014.

Gao, L., Bernbardt, M., Schulz, K., and Chen, X.W.: Elevation correction of ERA-Interim temperature data in the Tibetan Plateau, International Journal of

Climatology, 37(9): 3540-3552, 2017.

Gao, L., Wei, J., Wang, L., Bernhardt, M., Schulz, K., and Chen, X.: A high-resolution air temperature data set for the Chinese Tian Shan in 1979–2016, Earth System Science Data, 10, 2097-2114, 2018.

---

## Short Comment (SC6) · 12 Sep 2020

We would like to thank Dr. Chen for the valuable comment. In the following, we try to address all comments point-by-point.

Climate change likely alters regional hydrologic regime and affects access to water resources. Using a unique high-resolution (1 km, 6-hourly) air temperature data set (CTMD) produced by previous studies, this work investigated the existence of the elevation-dependence warming (EDW) in the Chinese Tianshan Mountains (CTM) at the monthly, seasonal, and annual time scales, which is of signiïficance for understanding glacier melting and water resources management in the vast arid area around CTM. The authors found that EDW was signiïficant at different altitudes on different

time scales. Overall, the present study is an interesting forward attempt to deepen our understanding of EDW phenomenon. Few queries to the authors, which might be helpful to improve this manuscript are given below:

1. In sections 3.1 and 3.2, the authors provided the significance of the linear regression models (equation (1)), it would also be helpful to report if the trends (coefiňĄcient alpha) were statistically different from zero (or to report the confidence intervals of the trends).

-Answer: Thanks a lot for Dr. Chen's suggestion. We provided the coefficients and confidence levels for figures (section 3.2 and 3.3), not for tables (section 3.1). We will add the regression coefficient information for Table 1 and 2 in the section 3.1 as well as the revised supplementary material.

2. Please explain the definition of seasons by grouping months in CTM region (this may be helpful for international readers).

-Answer: Thanks a lot for the comment. We are sorry for this unclear part. We defined the seasons as: December, January and February for winter, March to May for spring, June to August for summer, and September to November for autumn. We will add this definition in the subsequent revision.

3. Please elaborate a bit on how the trends for CTM (Table 1 &2) and China were calculated. If my understanding were correct, the authors applied equation (1) to estimate the trend for each pixel, but I am not clear how they aggregated the pixel-scale trends to the trends for CTM.

-Answer: Thanks a lot for the comment. Dr. Chen is right that the calculation of the trends in Table 1 and Table 2 is not clear. We used the equation (2) for the trend calculation for each pixel (grid) for minimum temperature, maximum temperature, and mean temperature at monthly, seasonal and annual temporal scales. In order to compare the trends between CTMD and CMA05 for the whole region scale, we used the averaged

trend values for these two data sets. We will add this information in the revision.

---

## Author Comment (AC2) · 19 Oct 2020

**Response to Referee 2**

We would like to thank anonymous Referee 2 for reviewing our manuscript. These constructive comments are very important for us to improve the present manuscript. In the following, we address all comments point-by-point according to referee's comments.

General comments:

This article analyses whether elevation-dependent warming (EDW) is present in the Chinese Tianshan Mountains, both overall and at a regional level. The authors present a compelling case for research into this phenomenon, as increased warming in higher regions may have detrimental effects on glacier melt. EDW is judged based on the criteria of regional warming amplification and altitude warming amplification, and these two criteria are assessed for the entirety of the Chinese Tianshan Mountains on a monthly time scale. Furthermore, spatial differences in EDW are assessed across the mountain range. Overall, the paper is well presented and structured, and the discussion and conclusions of this spatially and temporally complicated problem are interesting. However, there are some issues which I think need to be addressed before publication, most importantly the definition of EDW used in the paper and how it relates to the conclusions reached in the paper, and the suitability of the data set used for this analysis, as highlighted below.

Specific comments:

1. Whole paper: The authors have carefully defined elevation-dependent warming (EDW) immediately in the article, namely that two criteria should be met: regional warming amplification and altitude warming amplification. Section 3.1 concludes that regional warming amplification is only present in any of the minimum, mean and maximum daily temperatures in the months

from February to June. However, in section 3.2, warming amplification with altitude is now described as EDW, for example line 183 "The prevalence of EDW is most significant in December...". This is then used for the remainder of the paper, especially in the conclusions. The authors should identify the months which satisfy both regional warming and altitude warming amplification, and these months should be set out clearly as the months where EDW is present.

This needs to be altered throughout the paper, and has substantial implications for the conclusions, as I think there are only one or two months which satisfy both conditions.

-Answer: Thanks a lot for the comments. The reviewer pointed a very important issue. After carefully reviewing the literatures again, we have to admit that our previous definition of EDW were a bit arbitrary. To be precise, regional warming amplification and altitude warming amplification are the two basic characteristics (or "fundamental questions" from Rangwala and Miller, 2012) of EDW. In the previous literatures, although there are many discussions on altitude warming amplification in high mountains, no literature clearly states that regional warming amplification is one of criteria for EDW. We will revise this part through the whole paper in the revision.

2. Methods/CTMD dataset: I think there should be some discussion of the suitability and limitation of the CTMD dataset for this analysis, given that the paper is reliant on it. Two particular points stand out: o Gao et al., 2018 gives an analysis of the data set compared to a number of stations; however they are all under 3000 m asl. I do appreciate the difficulty of finding high elevations stations, but do the authors have any evidence that this data set is suitable at elevations of 5000 m asl and above? In addition, Gao et al., 2018 also indicates that the lapse rate from ERA-interim (the correction term used to downscale ERA-Interim to the 1km scale) is steeper than that seen in the

observations. It is often the case that the free atmosphere lapse rate is steeper than the near-ground lapse rate of temperature with elevation, and this difference may cause errors in the 1 km data set used in this paper.

Gao et al., 2018 acknowledge that the trends in the ERA-Interim data, and therefore the CTMD, do not always follow those of the observations. For example, in the minimum daily temperature, the trend in the CTMD considerably underestimates that of the observations. It is not clear whether this bias is constant with elevation, which is essential to the results presented in this manuscript.

-Answer: Thanks a lot for the comments. The reviewer raised a challenge issue on the quality of CTMD. We must admit that the credibility of data in high-altitude areas is always a huge challenge. In Gao et al., 2018, we used 24 sites to validate the CTMD. It is true that all the sites are lower than 3000 m. We are looking for reliable observation data all the time to further verify the quality of CTMD. We plan to update a more high resolution data V2.0 (100m, 6-hourly) since the CTMD V1.0 that released in 2018. However, as far as we know, there are only very few automatic weather observation stations between 3000-5000m. The time series of these observational data is always short than 10 years with some data gaps. Meanwhile, we have to clarify that these observations data are difficult to access due to permission issues. Therefore, we only could evaluate the credibility of CTMD based on limited observations. In general, we could conclude that the CTMD has a small large-scale bias because of small large-scale errors of ERA-Interim. Previous studies claimed that the large-scale errors of ERA-Interim are acceptable with respect to long-term trends (Gao et al., 2012; Simmons et al., 2010).

To response the question raised by the Referee, we plan to use the Land Surface Data Assimilation System (CLDAS-V2.0) near real-time product data set from China Meteorological Administration to verify the higher areas in the

revision. This data set applied multiple resources including more than 2400 surface observations, ECMWF/GFS reanalysis, and remote sensing data. Unfortunately, this data set only begins from 2008. The temporal and spatial resolution is 0.0625°×0.0625° and 1 hour, respectively. We hope we could provide more information on the quality of CTMD via comparing with CLDAS-V2.0.

About the lapse rate, the referee is right that the lapse rate from ERA-interim is steeper than the observations. Figure 4 in Gao et al., 2018 has shown that the lapse rates of ERA-Interim are greater than observations from September to December. Generally, the influence of elevation on temperature is basically unchanged at a smaller spatial resolution of 1km, while slope and aspect of the terrain become the dominant factors at hundreds meters. It is true that the free atmosphere lapse rate is steeper than the near-ground lapse rate of temperature because of the different radiation mechanism. To overcome this limitation, the downscaling model used different spatial spans, that is, from the near surface layer (~925hPa) to the free atmosphere (~500hPa). The selection of the lapse rate (such as $\Gamma_{700\_925}$) for each grid is completely dependent on its altitude, which reflects a larger elevation range as much as possible for a more real temperature lapse rate as possible. In the downscaling model, we used the ERA-Interim 2-m temperature instead of site temperature. Therefore, the downscaling model is completely independent of ground stations. However, we agree that the ERA-Interim lapse rate may be part of the source of error. Meanwhile, it is a challenge to distinguish this error quantitatively from the ERA-Interim model errors.

The referee is right that the trend of minimum temperature in CTMD does not follow that of observations in Gao et al., 2018. The CTMD in Gao et al., 2018 covers a larger area (818126 km$^2$), which includes such as the plains on the northern slope of the CTM and the basins on the southern slope of CTM. A "Cold Lake" effect may occur within the basin in winter. The lapse rate may be

positive rather than negative. For example, the Turfan Basin (below mean sea level) may have a temperature inversion layer in winter. The present study re-defines the Tianshan boundary according to Deng et al (2019). The CTM contains numerous inter-valley basins and oasis. Thus, the trends of minimum temperature in low terrains may be problematic. However, this study focuses on the trend over the whole CTM, and CTMD may not be good enough on the site scale, but it is still representative on the entire region. Again, we will introduce the CLDAS-V2.0 data set to further valid the reliability of CTMD in the revision.

3. Table 1 and 2: given the variation over time, it would be useful to know which of these trends is statistically significant.

-Answer: Thanks a lot for pointing this out. We will mark the significance levels with asterisk in Table 1 and 2 in the revision.

**Table 1. Annual and seasonal temperature trends (°C 10a$^{-1}$) in the CTM (based on CTMD) and continental China (based on CMA05) from 1979–2016.**

|  | CTMD | | | CMA05 | | |
| --- | --- | --- | --- | --- | --- | --- |
|  | Tmin | Tmean | Tmax | Tmin | Tmean | Tmax |
| Spring | **0.633** *** | **0.522** *** | **0.640** *** | 0.557 *** | 0.513 *** | 0.518 *** |
| Summer | 0.441 *** | 0.342 *** | 0.266 ** | 0.472 *** | 0.388 *** | 0.378 *** |
| Autumn | 0.302 | 0.200 * | 0.270 | 0.551 *** | 0.458 *** | 0.420 *** |
| Winter | 0.014 | -0.085 | 0.115 | 0.432 *** | 0.361 *** | 0.327 *** |
| Annual | 0.347 *** | 0.245 *** | 0.323 *** | 0.503 *** | 0.430 *** | 0.411 *** |

Note: the bold and underlined value indicates a greater warming trend in the CTM than continental China. * denotes the significance level $p<0.1$, ** denotes the significance level $p<0.05$, and *** denotes the significance level $p<0.01$.

**Table 2. Monthly temperature trends (°C 10a$^{-1}$) in the CTM (based on CTMD) and the continental China (based on CMA05) from 1979–2016.**

|  | CTMD | | | CMA05 | | |
| --- | --- | --- | --- | --- | --- | --- |
|  | Tmin | Tmean | Tmax | Tmin | Tmean | Tmax |
| January | -0.133 | -0.269 | -0.235 | 0.343 ** | 0.256 | 0.212 |
| February | 0.313 | 0.177 | **0.605** ** | 0.558 *** | 0.523 *** | 0.549 ** |
| March | **0.835** ** | **0.818** *** | **1.339** *** | 0.651 *** | 0.672 *** | 0.752 *** |
| April | 0.441 | **0.537** *** | **0.664** * | 0.547 *** | 0.522 *** | 0.516 *** |
| May | **0.624** ** | 0.211 | -0.082 | 0.475 *** | 0.345 *** | 0.284 *** |

| Month | | | | | | |
|---|---|---|---|---|---|---|
| June | **0.752** *** | **0.476** *** | **0.422** *** | 0.516 *** | 0.390 *** | 0.344 *** |
| July | 0.227 | 0.331 *** | 0.280 | 0.472 *** | 0.411 *** | 0.416 *** |
| August | 0.342 | 0.217 * | 0.095 | 0.429 *** | 0.363 *** | 0.375 *** |
| September | 0.246 | 0.237 | 0.330 | 0.559 *** | 0.486 *** | 0.495 *** |
| October | 0.273 | 0.180 | 0.227 | 0.524 *** | 0.434 *** | 0.398 *** |
| November | 0.386 | 0.183 | 0.252 | 0.569 *** | 0.455 *** | 0.368 ** |
| December | -0.137 | -0.164 | -0.025 | 0.394 *** | 0.303 ** | 0.219 |

Note: the bold and underlined value indicates a greater warming trend in the CTM than continental China. * denotes the significance level $p<0.1$, ** denotes the significance level $p<0.05$, and *** denotes the significance level $p<0.01$.

4. Line 128: How were 6-hourly data aggregated to the minimum and maximum temperature? Was any consideration given to the minimum/maximum temperature not occurring at 00, 6, 12, 18 UTC?

--Answer: Thanks a lot for pointing this out. The minimum and maximum temperatures are calculated from four temperature records. The observation standard of the China Meteorological Administration is also the instantaneous temperature four times a day, from 20 o'clock of previous day to 20 o'clock of current day at local time (UTC+8 Beijing time). The minimum/maximum temperature possible occurs at other time, rather than 00, 6, 12, 18 UTC. However, normally, the maximum temperature occurs around 14 o'clock (06:00 UTC). The minimum temperature occurs around 4 to 5 o'clock in the morning, which is close to 18 UTC (2 o'clock at Beijing time). Therefore, there is only limited effect for minimum and maximum temperature calculation from the 6-hourly data set.

5. Related to points 3 and 4: I'm surprised that in some cases, the warming increase in Tmin and Tmax are both greater than the warming increase in Tmean. This suggests some unusual shift in the shape of the diurnal cycle. Could the authors hypothesise as to why this might be?

--Answer: Thanks a lot for pointing this out. We checked the data carefully again. We found that the header of table does not correspond to the data. It means that the data is in the wrong column. We are very sorry for this kind of

mistake that shouldn't be. We correct it in the revision.

**Table 1. Annual and seasonal temperature trends (°C 10a$^{-1}$) in the CTM (based on CTMD) and continental China (based on CMA05) from 1979–2016.**

| | CTMD | | | CMA05 | | |
|---|---|---|---|---|---|---|
| | Tmin | Tmean | Tmax | Tmin | Tmean | Tmax |
| Spring | **0.633** *** | **0.522** *** | **0.640** *** | 0.557 *** | 0.513 *** | 0.518 *** |
| Summer | 0.441 *** | 0.342 *** | 0.266 ** | 0.472 *** | 0.388 *** | 0.378 *** |
| Autumn | 0.302 | 0.200 * | 0.270 | 0.551 *** | 0.458 *** | 0.420 *** |
| Winter | 0.014 | -0.085 | 0.115 | 0.432 *** | 0.361 *** | 0.327 *** |
| Annual | 0.347 *** | 0.245 *** | 0.323 *** | 0.503 *** | 0.430 *** | 0.411 *** |

Note: the bold and underlined value indicates a greater warming trend in the CTM than continental China. * denotes the significance level $p<0.1$, ** denotes the significance level $p<0.05$, and *** denotes the significance level $p<0.01$.

**Table 2. Monthly temperature trends (°C 10a$^{-1}$) in the CTM (based on CTMD) and the continental China (based on CMA05) from 1979–2016.**

| | CTMD | | | CMA05 | | |
|---|---|---|---|---|---|---|
| | Tmin | Tmean | Tmax | Tmin | Tmean | Tmax |
| January | -0.133 | -0.269 | -0.235 | 0.343 ** | 0.256 | 0.212 |
| February | 0.313 | 0.177 | **0.605** ** | 0.558 *** | 0.523 *** | 0.549 ** |
| March | **0.835** ** | **0.818** *** | **1.339** *** | 0.651 *** | 0.672 *** | 0.752 *** |
| April | 0.441 | **0.537** *** | **0.664** * | 0.547 *** | 0.522 *** | 0.516 *** |
| May | **0.624** ** | 0.211 | -0.082 | 0.475 *** | 0.345 *** | 0.284 *** |
| June | **0.752** *** | **0.476** *** | **0.422** *** | 0.516 *** | 0.390 *** | 0.344 *** |
| July | 0.227 | 0.331 *** | 0.280 | 0.472 *** | 0.411 *** | 0.416 *** |
| August | 0.342 | 0.217 * | 0.095 | 0.429 *** | 0.363 *** | 0.375 *** |
| September | 0.246 | 0.237 | 0.330 | 0.559 *** | 0.486 *** | 0.495 *** |
| October | 0.273 | 0.180 | 0.227 | 0.524 *** | 0.434 *** | 0.398 *** |
| November | 0.386 | 0.183 | 0.252 | 0.569 *** | 0.455 *** | 0.368 ** |
| December | -0.137 | -0.164 | -0.025 | 0.394 *** | 0.303 ** | 0.219 |

Note: the bold and underlined value indicates a greater warming trend in the CTM than continental China. * denotes the significance level $p<0.1$, ** denotes the significance level $p<0.05$, and *** denotes the significance level $p<0.01$.

6. Section 3.3: This analysis of the spatial variations is interesting, and Figures 5-7 quite well represent the first requirement for EDW, that the warming in the region is greater than the surrounding area. However, it is difficult to see the altitude warming amplification from these plots unless you are well-acquainted with the topography of the region (e.g. from figure 5b it's only really possible to see a north-south gradient in area 1, it's not clear that that corresponds with

high-low). Would it be possible to add (small) plots such as those in figures 2-4 to figures 5-7 for each region? If it's not possible to fit the graphs on, perhaps the trends and significance could be calculated, such as in figure 2-4? As in point 1, only those areas which fit both criteria should be described as EDW.

--Answer: Thanks a lot for the comment. The reviewer provided a very good suggestion to show the difference in spatial variations. The sub-plot is feasible. We select a certain direction in typical zone 2, and then establish a terrain profile with the corresponding temperature trend. Fox example:

[Figure]

**Figure 5: Monthly minimum temperature trends (a) January and (b) December for the**

entire CTM from 1979–2016. The top two sub-plots show the elevation and temperature trend along the terrain profile (black arrow) in Zone 2, respectively.

[Figure]

**Figure 6: Monthly maximum temperature trends (a) March and (b) September for the entire CTM from 1979–2016. The top two sub-plots show the elevation and temperature trend along the terrain profile (black arrow) in Zone 2, respectively.**

[Figure]

**Figure 7: Monthly mean temperature trends (a) January and (b) February for the entire CTM from 1979–2016. The top two sub-plots show the elevation and temperature trend along the terrain profile (black arrow) in Zone 2, respectively.**

Smaller remarks, technical comments and suggestions:

7. Figure 1: Does the bottom right hand corner map show the extent of the CMA05 used in this analysis? If so, please add to the caption. If not, could this be altered to show the CMA05 extent?

--Answer: Thanks a lot for pointing this out. We will revise and add the grid

points of CMA05 in the Figure 1.

[Figure]

**Figure1: Location of the Chinese Tianshan Mountains (CTM).The elevation ranges from 204 m to 7100 m a.s.l., with a DEM resolution of 1 km from SRTM. The grey sub-plot show the extent of the CMA05 at 0.5 °×0.5 °grid.**

8. Introduction: It would be useful to make clear earlier on in the paper that EDW is referring to the rate of warming over a multi-annual scale (rather than, say, rate of warming during the day). This is made clear on line 58 with 'warming trend of annual mean temperature' but could be mentioned earlier.

--Answer: Thanks a lot for pointing this out. We will modify the expression for the whole text to clarify the warming trend over a multi-annual scale in the revision.

9. Around line 120 onwards-perhaps mention that the topography comes from SRTM.

--Answer: Thanks a lot for pointing this out. We will clarify the source of DEM that comes from SRTM. It also will be noted in the caption of Figure 1.

10. Line 136-137: it might be sensible to combine the highest two elevation bands, given that the highest only contains 4 points (which may not be

--Answer: Thanks a lot for pointing this out. The referee is right that only four grids above than 7000m. However, we tend to keep these 4 grids because they represent the highest peaks in the entire CTM. Meanwhile, these four grids basically have the similar performance as that of 6500-7000m group.

11. Line 268-270: I think this sentence can be removed as you're only talking about surface albedo here.

--Answer: Thanks a lot for pointing this out. We will revise it in the revision.

12. Line 113: remove 'because' (either 'because the system....,the bias could be' or 'The system bias of.... Thus, the bias...'

--Answer: Thanks a lot for pointing this out. We will revise it in the revision.

13. Please consider changing the colour bars in Figs. 5 to 7 so that they are all the same (and ideally centred around 0, so that red is positive, blue is negative and yellow around zero). At first glance it seems that the maximum temperature trends in March have both positive and negative values, which as you point out in the text is not the case. In addition, please flip the colour bars so that negative values are on the left and positive on the right.

--Answer: Thanks a lot for the comment. The suggestion is excellent for improving the readable of figures. We will revise the color bar in the revision. For example:

[Figure]

**Figure 5: Monthly minimum temperature trends (a) January and (b) December for the entire CTM from 1979–2016.   The top two sub-plots show the elevation and temperature trend along the terrain profile (black arrow) in Zone 2, respectively.**

14. Give the location of the Ili valley where it first appears on line 208, rather than 210. References Gao, L., Wei, J., Wang, L., Bernhardt, M., Schulz, K., and Chen, X.: A high-resolution air temperature data set for the Chinese TianShan in1979–2016, Earth System Science Data, 10, 2097-2114, 2018.

--Answer: Thanks a lot for pointing this out. We will revise it in the revision.

Reference:

Deng, H., Chen, Y., and Li, Y.: Glacier and snow variations and their impacts on regional water resources in mountains, Journal of Geographical Sciences, 29(1): 84-100, 2019.

Gao, L., Bernhardt, M., and Schulz, L.: Elevation correction of ERA-interim temperature data in complex terrain, Hydrology and Earth System Sciences, 16(12): 4661-4673, 2012.

Simmons, A. J., Willett, K. M., Jones, P. D., Thorne, P. W., and Dee, D. P.: Low-frequency variations in surface atmospheric humidity, temperature, and precipitation: Inferences from reanalyses and monthly gridded observational data sets, J. Geophys. Res.-Atmos., 115, D01110, doi:10.1029/2009jd012442, 2010.

---

## Referee Comment (RC3) · Anonymous Referee #3 · 5 Nov 2020

Gao and co-authors present an analysis of decadal air temperature trends against elevation to explore the case for elevation dependent warming (EDW) in the Chinese Tianshan Mountains (CTM). The authors explore this across a large domain using a recent 1km resolution product derived based upon ERA-Interim reanalysis and station data up to 3000 m a.s.l. They find that for given months and sub-domains of the CTM, EDW is evident, though is complex and not consistent or clear for all domains or seasons. The manuscript is well written in parts and explores a very interesting and relevant topic within the cryosphere. While the work has particular value to be published in the journal, I believe much more needs to be done to explain the data sources and their limitations, to convince the reader of the validity of CTMD product and therefore the uncertainty and limitations of their results as well as providing more

justification and better presentation of the key findings.

General Comments I think the manuscript has promise and could be substantially improved based upon some key things.

1) The authors give general reference to their ESSD paper for details about the CMTD product, but a much stronger section of the data and methods need to be presented for this manuscript in order to summarise the key details about how the CMTD was derived, for what time scale it is processed and what the major assumptions or limitations are that might affect the analysis of EDW. It's apparent to me that the authors are already considering these limitations etc, based upon their responses to other reviewers on the open-discussions. To the reader of this manuscript, there is not enough information presented to judge the quality of the CTMD and assess the validity of the results that are based upon it.

2) I have the same issue as 1), but also for the CMA05 product. I am left questioning the comparability of the two for the tabular information presented (the first criterion of EDW that is the regionally amplified warming). For the CMA05, all pixels are averaged to produce a temperature/warming trend for all elevations across the entirety of China? Is this dataset also derived from ERA-I? Does it include the CTM as well, or all the rest of China except the study domain? If it is all of China, this then also includes other mountain regions of the country? In general, I like the succinct and to-the-point paper, but there are a lot of important pieces of information that are missing and without them, the reader cannot gain a good appreciation of the scientific rigour and value of the authors work. Being clearer about some of those elements will greatly aid the scientific conclusions.

3) In some places, a justification for showing some months and not others are needed. Figures for Tmin, Tmax and Tmean all show different months, for example. Is this purely just to show the months with the strongest trends? Some work needs to go into the figures as well. I see that that has begun already based upon comments from
reviewer#2. In each figure, the authors show different scales (y-axis limits are different in Figures 2-4 and colour scales are different in each subplot for Figures 5-7), and it becomes hard for the reader to easily compare and understand them, and take away the key message(s). See specific comments on the figures below.

4) The manuscript presents a rather general discussion with little further exploration of possible mechanisms. There is a repetition of general comments regarding, for example, the albedo's role on the surface energy balance, but this never links with why we may see EDW in certain months or why the strongest warming may occur only for Tmin in January/December and why Tmax trends or regional (east-west) temperature trends (e.g. Figure 5) might occur. A reference of Deng et al. 2019 is given, for example, but it is not elaborated upon much. Can this or other datasets or analyses regarding snow cover/albedo from MODIS tell us more about why EDW might be occurring for certain seasons/mountains/zones? I don't suggest that the authors do a full analysis of snow cover, but some additional and more in-depth discussion points are definitely required.

5) Finally, throughout the manuscript, the terminology of EDW and trends/gradients shifts somewhat and consistency is required throughout (following a clear initial definition). Moreover, the use of the word 'significantly' comes up a lot to refer to differences in trends across space (for the maps) and time (for seasons / months). Unless these differences are tested for significance and values reported, care should be taken for the wording and adjusted appropriately.

Specific Comments

Abstract L26 -What are EDW 'Features'? I would consider rewording this. L26-27 – Please add here the time period over which CMTD was derived and analysed (1979-2016?) L28 – Statistically significant elevation dependence? Add that if so. L34 – While I do not disagree that this is a likely contributor to glacier melt in the CTM, the authors do not explicitly 'explain' this link, especially as the EDW trends are not so clear for all summer months. It's possible that stronger trends in warming at high elevations

in April could have a key influence on some more precipitation falling as rain, but again, the authors cannot (based upon the presented work) state this. I would rephrase this to something like "This new evidence could partly explain the accelerated melting of glaciers in the CTM, though the mechanisms remain to be explored" or similar.

Introduction

L36 – two 'criteria

L50 – Current 'evidence'

L54 – Please elaborate here and add some reasoning of seasonal significance from those studies.

L58 – What is global mountain detection? Do the authors refer to detection of trends or 'observations' in general for mountain regions? Please clarify and reword.

L58-74 This paragraph reads rather disjointed without a clear flow or argument. Because it recounts several other instances of studies exploring EDW, the overview might be more valuable to the reader in a tabular format? I would suggest to restructure this paragraph and improve the flow of the writing.

L72-73 – Please clarify what satellite data the authors refer to and how that shows EDW/climate warming at specified elevations. How does this point fit into the context of the manuscript discussion and/or the strengths/limitations of the presented dataset?

L81 – Do the authors refer to 56 gridded points of a given product presented by You et al.? Please clarify and rewrite.

L87 – To me the "largest independent latitudinal mountain system" is not clear. Can the authors clarify its meaning or remove it?

Data and Methods

L109 – CTMD is briefly defined at the end of the introduction, but should be described in

sufficient detailed before introducing other datasets to compare to it. See my general comment about elaborating on the CTMD product, especially on its derivation and potential limitations for exploring EDW in this manuscript.

L111-Taking all elevations of CMA05? It is not clear how comparable these products are (see general comment). For the CTMD product, the definition of mountain domain is all of the CTMD pixels (including low elevations)? I am left questioning whether the comparison of the CTMD and CMA05 trends are valid and how the values for Table 1 were derived for each of them. More information is required here.

L112 – Can the authors define what is a small large scale error? Small biases over large domains?

L113 – systematic?

L116-118 – It would be valuable to recount that winter lapse rates were not well estimated by CTMD compared to the station data as shown by Fig. 4 of Gao et al., 2018. Some mention here (or in the discussion) needs to explore the potential impact that this might have on your results. If, for example, your temperatures at the highest elevations were estimated by the station lapse rates, would they be largely different from what the CTMD gives you? Could this strongly affect the EDW trends for the highest elevations in January/December? I don't expect that the authors should use the low-elevation stations to derive the high elevation temperatures for their analyses, but some discussion on the limitations of CTMD for the current analyses are required somewhere in the manuscript.

L126 – reword to 'six-hourly timestep'

L136-138 – fine, but maybe neaten, use of table?

L139 – statistical significance of the linear regression? What p-value defines your statistical significance when you use the term significant in the abstract?

L141- averaged is mean or median? (cf boxplots with median red line plotted)

Results

L150 – This needs clarification. Do the authors refer to the elevation gradient of decadal temperature trends or the gradient (slope) of the regression line that quantifies the trend in each elevation band? If referring to the latter, please use the word trend (or similar) instead to not confuse with temperature gradient/lapse rate.

L174 – Why those months only? How are they 'representative'? Representative of what? I don't see a clear segregation of season, January and December both have negative trends for the whole domain (converse to the CMA05), April is not as large an increase as March... More justification is needed. Are the authors simply showing all of the results which have more warming somewhere?

L176 – Is your average a Mean? Median? Note that median is displayed for boxplots.

L185 - Figure 3 now investigates March, April, August and September. Why are the same months not compared and what is the justification this time?

L193 – Months of interest for Tmean are again different.

L203 – Statistically significantly different? If so, by what test and what significance? Same comment throughout the paragraph, please clarify the significance or reword it.

L207 – are warmer on average, the figure rather shows a higher rate of warming. Check sentence.

Possible hypotheses and mechanisms

I feel that this section should be under the general header of 'discussion'. Please see general comments on this section. I believe that much more is needed for this section. It is very general and I don't go away feeling that I learned anything new.

L255 – Also the snow cover and snow albedo here affect this... This is mentioned in the next paragraph and the information is essentially repeated with no additional information gain.

L264-265 – Sure, this could be a mechanism, but has there been any other studies demonstrating snow cover changes and albedo changes in the CTM? I note that the Deng paper is cited but not investigated further. Because the CTMD product is generated through station observations at lower elevations, would this not bias representation of high elevation changes? Of course, I appreciate that there are no available data at those higher elevations, but this needs to be mentioned and limitations of the dataset/study need to be linked with a more in depth interpretation of the most noteworthy results.

L273 – Could be? Are these model simulations of idealised conditions or did authors find this specifically for that zone? Reword to 'estimated glacier mass loss...'

L275 – 'In summary'

Discussion and Conclusions

In my opinion, this section needs splitting into; 1) a greater discussion with section 4 (see general comment and above) and, 2) a clear and concise, separate conclusions section.

L284 – 'DO' not (in the case of CTM) clearly reflect EDW. Not cannot.

L285-286 – This belongs to the previous section. The authors should elaborate whether earlier spring snow melt is significant (and quantify significance) or at least demonstrate if past work suggests that warming at those higher elevations is more likely. Comparing some general estimates of snow line elevation or from previous findings to those same elevation bands would be of value, though I'm sceptical if the CTMD product will reflect that change.

L288 – Replace gradients with trends unless referring specifically to the difference across the elevation bands (Figures 2-4). In general, the terminology needs clarification.

L297 – I think that this is a crucial point. Above 5000 m, there are always positive

changes. . . but for lower elevations in those winter months, there are largely negative trends for minimum and some mean temperatures (Figures 2 and 4). I would like to see more discussion as to why we might expect to have a general cooling (negative) trend for the winter minimum below 3000 m. The lack of discussion regarding the mechanisms is a major drawback to the current manuscript version.

L297-298 – Or could be warming as a result of snow cover depletion (feedback)?

L297-302 – This reads like a results section again.

Figures

-My general issue with the figures is the lack of standardisation (i.e. different colour and y-axis scales) and the ever changing months presented. It leaves the reader with no strong idea as to the key findings.

-I recommend maintaining the same y-axis limits to all sub-plots in Figures 2-4, labelling the months on the plots for easier interpretation.

-For Figures 5-7, please adjust the colour scale from left (blue – negative) to right (red – positive) following the reviewer#2 comments and also set the same total scale limit for each plot (i.e. -1.5 - +1.5°C 10a-1) with 0°C trend always being the same colour (pale yellow or white). Do the authors also report trends that are not statistically significant? If so, I would also represent these as white or blank pixels if possible. This will aid the reader's ability to interpret and compare the magnitudes of trends between sub-plots/figures as well as areas that aren't statistically significant trends.

- I would suggest adding some other figure(s) that shows the interannual variability of Tmin/Tmax/Tmean for some of the highest elevation pixels so we can better interpret how the suspected EDW warming for March/April/(or month of most interest) looks compared to some lower elevations, or compared to the 'background' change of 'non-mountain' regions shown from the CMA05, if the CMA05 and CTMD are indeed comparable (see general comment). These are the two criteria for EDW and need to

be more convincingly demonstrated and discussed.

---

## Author Comment (AC3) · 25 Nov 2020

**Response to Referee 3**

We would like to thank anonymous Referee 3 for reviewing our manuscript. These constructive comments are very important for us to improve the present manuscript. In the following, we address all comments point-by-point according to referee's comments.

General comments:

Gao and co-authors present an analysis of decadal air temperature trends against elevation to explore the case for elevation dependent warming (EDW) in the Chinese Tianshan Mountains (CTM). The authors explore this across a large domain using a recent 1km resolution product derived based upon ERA-Interim reanalysis and station data up to 3000 m a.s.l. They find that for given months and sub-domains of the CTM, EDW is evident, though is complex and not consistent or clear for all domains or seasons. The manuscript is well written in parts and explores a very interesting and relevant topic within the cryosphere. While the work has particular value to be published in the journal, I believe much more needs to be done to explain the data sources and their limitations, to convince the reader of the validity of CTMD product and therefore the uncertainty and limitations of their results as well as providing more justification and better presentation of the key findings.

General Comments I think the manuscript has promise and could be substantially improved based upon some key things.

1) The authors give general reference to their ESSD paper for details about the CMTD product, but a much stronger section of the data and methods need to be presented for this manuscript in order to summarise the key details about how the CMTD was derived, for what time scale it is processed and what the major assumptions or limitations are that might affect the analysis of EDW. It's

apparent to me that the authors are already considering these limitations etc, based upon their responses to other reviewers on the open-discussions. To the reader of this manuscript, there is not enough information presented to judge the quality of the CTMD and assess the validity of the results that are based upon it.

-Answer: Thanks a lot for the comments. The reviewer raised a very important issue as the referee 2 has pointed out before. The data set CTMD is the most important basis for EDW analysis in this study. We know that the credibility of the data set determines the reliability of EDW detection. Indeed, we did not provide much information (such as data production process) on the data set while we focused more on EDW analysis. We agree with the referee that the limitations of the CTMD should be fully demonstrated in the manuscript for better understanding of readers especially who are the potential data users. In the response to referee 2, we planned to use the Land Surface Data Assimilation System (CLDAS-V2.0) from the near real-time product data set from China Meteorological Administration to verify the higher elevations of the CTMD. However, we found this data set (in 2008-2016, we checked last time) is not available since it only begins in 2017, although it applied multiple data resources since 2007 in the data production process. Therefore, we have to seek other data sources to strengthen the verification. What we are struggling with is whether there is really a data set suitable for validating our CTMD product. Due to the lack of ground stations in high mountains (above 3500m), any other reproduce data sets (such as CRU data set at monthly and 0.5 degree spatial-temporal resolution) are flawed. We have always been very worried that the quality of CTMD seems to become an unproven issue. We appreciate that the referee 3 also pointed out the difficulty of observation acquisition. However, we agree that we should present the limitations of the CTMD without reservation in the revision.

2) I have the same issue as 1), but also for the CMA05 product. I am left

questioning the comparability of the two for the tabular information presented (the first criterion of EDW that is the regionally amplified warming). For the CMA05, all pixels are averaged to produce a temperature/warming trend for all elevations across the entirety of China? Is this dataset also derived from ERA-I? Does it include the CTM as well, or all the rest of China except the study domain? If it is all of China, this then also includes other mountain regions of the country? In general, I like the succinct and to-the-point paper, but there are a lot of important pieces of information that are missing and without them, the reader cannot gain a good appreciation of the scientific rigour and value of the authors work. Being clearer about some of those elements will greatly aid the scientific conclusions.

-Answer: Thanks a lot for the comments. The referee is right that the information on the CMA05 is not enough for the readers. We will add more details on the processes of CMA05. In this study, the CMA05 which covers the whole continental China (including the CTM) was compared to CTMD. We think the referee provides a good idea that the CMA05 without the CTM can also be compared. Thus, we will add the trend analysis using the CMA05 excluded the CTM as well as the CMA05 excluded the Tibetan Plateau (The TP is considered to be one of the most intense warming regions in China) in the section 3.1 and also update the results in Table 1 and Table 2 in the revision.

3) In some places, a justification for showing some months and not others are needed. Figures for Tmin, Tmax and Tmean all show different months, for example. Is this purely just to show the months with the strongest trends? Some work needs to go into the figures as well. I see that that has begun already based upon comments from reviewer#2. In each figure, the authors show different scales (y-axis limits are different in Figures 2-4 and colour scales are different in each subplot for Figures 5-7), and it becomes hard for the reader to easily compare and understand them, and take away the key

message(s). See specific comments on the figures below.

-Answer: Thanks a lot for pointing this issue out. We must admit that the representative months we selected indeed have a significant warming trend. But it is not limited to these four months. We have shown the warming trend for all months in the Supplementary material. Here we want to clarify that we did not use a uniform scale (y-axis limits). We have tried. But the temperature increasing trend for some months at some elevation groups are negative. If a uniform scale used, the possible range could be -1.6 to 2 ℃ 10a$^{-1}$. Thus, for some months, the box plot will appear very crowded and small, which is in a poor readable for the percentile ranges (25% to 75%). Thus, we keep the different y-axis ranges. However, the referee's comment is reasonable. We figure out a good way to show the trend comparison for all month is adding a table which including all slope and significance levels. The table is as following:

**Table 3. Monthly temperature trends (℃ 10a$^{-1}$) in different elevations based on CTMD from 1979–2016.**

|  | Tmin | Tmean | Tmax |
|---|---|---|---|
| January | 0.039*** | 0.036*** | 0.037*** |
| February | 0.033*** | 0.012 | 0.008*** |
| March | 0.023 | 0.009** | 0.017*** |
| April | 0.021*** | -0.02*** | **0.069***** |
| May | -0.056*** | -0.022*** | -0.045*** |
| June | -0.025*** | 0.007 | -0.046*** |
| July | 0.0 | -0.017** | -0.019** |
| August | -0.011 | **0.037*** | **0.023*** |
| September | -0.006 | **0.017** | **0.038*** |
| October | -0.073*** | -0.018*** | **0.017** |
| November | -0.032*** | -0.031*** | -0.018*** |
| December | 0.064*** | **0.006** | -0.018*** |

Note: the bold and underlined value indicates a warming trend for higher elevations, not for the whole elevation range. More details could be found in Figure 2 to 4 and Figure S1 to S12. * denotes the significance level $p<0.1$, ** denotes the significance level $p<0.05$, and *** denotes the significance level $p<0.01$.

4) The manuscript presents a rather general discussion with little further exploration of possible mechanisms. There is a repetition of general comments

regarding, for example, the albedo's role on the surface energy balance, but this never links with why we may see EDW in certain months or why the strongest warming may occur only for Tmin in January/December and why Tmax trends or regional (east-west) temperature trends (e.g. Figure 5) might occur. A reference of Deng et al. 2019 is given, for example, but it is not elaborated upon much. Can this or other datasets or analyses regarding snow cover/albedo from MODIS tell us more about why EDW might be occurring for certain seasons/mountains/zones? I don't suggest that the authors do a full analysis of snow cover, but some additional and more in-depth discussion points are definitely required.

-Answer: Thanks a lot for the comments. The reviewer pointed a very key issue. The physical mechanism of EDW is indeed a challenge issue. The current researches are more about the hypothetical mechanism, rather than quantitative physical mechanism investigation. From our view, surface energy balance is the core mechanism. Among them, snow/ice covers that resulting in surface albedo changes may be the core influencing factors. Deng et al. (2019) did preliminary research using simple statistical analysis, which is not enough to explain the physical mechanism. That is exactly what we want to do in the future, that is, using dynamic models (e.g. WRF) to simulate the relationship between surface ground cover and near surface air temperature. The reviewer's comment is very constructive. We plan to use the remote sensing data (MODIS) to explain the possible impacts of snow/ice cover on temperature changes in the revision.

5) Finally, throughout the manuscript, the terminology of EDW and trends/gradients shifts somewhat and consistency is required throughout (following a clear initial definition). Moreover, the use of the word 'significantly' comes up a lot to refer to differences in trends across space (for the maps) and time (for seasons/months). Unless these differences are tested for significance and values reported, care should be taken for the wording and adjusted

appropriately.

-Answer: Thanks a lot for the comments. The referee 2 also pointed out the terminology problem. We admit that we did not give a very clear definition on EDW, even some misunderstanding. In the revision, we will clarify the EDW definition as well as its features. The trends indeed represent different means respect to space and temporal scale. We will specifically state in the result part in the revision.

Specific comments:

6) Abstract L26 -What are EDW 'Features'? I would consider rewording this.

-Answer: Thanks a lot for the comments. To be precise, regional warming amplification and altitude warming amplification are the two basic EDW characteristics. We will reword this part in the revision.

7) L26-27 – Please add here the time period over which CMTD was derived and analysed (1979- 2016?)

-Answer: Thanks a lot for the comments. We will add the time series 1979-2016 in the revision.

8) L28 – Statistically significant elevation dependence? Add that if so.

-Answer: Thanks a lot for the comments. We will add the statistical significances in the revision.

9) L34 – While I do not disagree that this is a likely contributor to glacier melt in the CTM, the authors do not explicitly 'explain' this link, especially as the EDW trends are not so clear for all summer months. It's possible that stronger trends in warming at high elevations in April could have a key influence on some more precipitation falling as rain, but again, the authors cannot (based upon the presented work) state this. I would rephrase this to something like "This new

evidence could partly explain the accelerated melting of glaciers in the CTM, though the mechanisms remain to be explored" or similar.

-Answer: Thanks a lot for the comments. Our conclusion may be a little bit arbitrary. We will revise this part in the revision.

Introduction

10) L36 – two 'criteria

-Answer: Thanks for pointing this out. We will revise it in the revision.

11) L50 – Current 'evidence'

-Answer: Thanks for pointing this out. We will revise it in the revision.

12) L54 – Please elaborate here and add some reasoning of seasonal significance from those studies.

-Answer: Thanks for pointing this out. We will add more information on it in the revision.

13) L58 – What is global mountain detection? Do the authors refer to detection of trends or 'observations' in general for mountain regions? Please clarify and reword.

-Answer: Thanks for pointing this out. "Global mountain detection" means the researcher investigated the temperature trends for most of large mountains over the world. We will clarify this literature in the revision.

14) L58-74 This paragraph reads rather disjointed without a clear flow or argument. Because it recounts several other instances of studies exploring EDW, the overview might be more valuable to the reader in a tabular format? I would suggest to restructure this paragraph and improve the flow of the writing.

-Answer: Thanks for pointing this out. We will restructure this paragraph and improve the flow of the writing in the revision.

15) L72-73 – Please clarify what satellite data the authors refer to and how that shows EDW/climate warming at specified elevations. How does this point fit into the context of the manuscript discussion and/or the strengths/limitations of the presented dataset?

-Answer: Thanks for pointing this out. We will check the literature in the revision.

16) L81 – Do the authors refer to 56 gridded points of a given product presented by You et al.? Please clarify and rewrite.

-Answer: Thanks for pointing this out. We will clarify this literature in the revision.

17) L87 – To me the "largest independent latitudinal mountain system" is not clear. Can the authors clarify its meaning or remove it?

-Answer: Thanks for pointing this out. We remove it in the revision.

Data and Methods

18) L109 – CTMD is briefly defined at the end of the introduction, but should be described insufficient detailed before introducing other datasets to compare to it. See my general comment about elaborating on the CTMD product, especially on its derivation and potential limitations for exploring EDW in this manuscript.

-Answer: Thanks a lot for the comments. We will add more information in the revision.

19) L111-Taking all elevations of CMA05? It is not clear how comparable these

products are (see general comment). For the CTMD product, the definition of mountain domain is all of the CTMD pixels (including low elevations)? I am left questioning whether the comparison of the CTMD and CMA05 trends are valid and how the values for Table 1 were derived for each of them. More information is required here.

-Answer: Thanks a lot for the comments. We will clarify this part and add more analysis in the revision.

20) L112 – Can the authors define what is a small large scale error? Small biases over large domains?

-Answer: Thanks for pointing this out. We will clarify this part in the revision.

21) L113 – systematic?

-Answer: Thanks for pointing this out. Yes, we will correct it in the revision.

22) L116-118 – It would be valuable to recount that winter lapse rates were not well estimated by CTMD compared to the station data as shown by Fig. 4 of Gao et al., 2018. Some mention here (or in the discussion) needs to explore the potential impact that this might have on your results. If, for example, your temperatures at the highest elevations were estimated by the station lapse rates, would they be largely different from what the CTMD gives you? Could this strongly affect the EDW trends for the highest elevations in January/December? I don't expect that the authors should use the low-elevation stations to derive the high elevation temperatures for their analyses, but some discussion on the limitations of CTMD for the current analyses are required somewhere in the manuscript.

-Answer: Thanks a lot for the comments. The reviewer is right that the limitation of CTMD should be fully demonstrated in the discussion, especially the poor simulation of lapse rate by CTMD in winter.

-Answer: Thanks for pointing this out. We will correct it in the revision.

-Answer: Thanks for pointing this out. We will neaten it in the revision.

25) L139 – statistical significance of the linear regression? What p-value defines your statistical significance when you use the term significant in the abstract?

-Answer: Thanks a lot for the comments. We used 0.1, 0.05, and 0.01 for p-value to define statistical significance. We add this information for Table 1 and Table 2, as well as the abstract in the revision.

26) L141- averaged is mean or median? (cf boxplots with median red line plotted)

-Answer: Thanks a lot for the comments. Yes, the boxplots show the median value. We used the mean value for consistent trend calculation.

Results

27) L150 – This needs clarification. Do the authors refer to the elevation gradient of decadal temperature trends or the gradient (slope) of the regression line that quantifies the trend in each elevation band? If referring to the latter, please use the word trend (or similar) instead to not confuse with temperature gradient/lapse rate.

-Answer: Thanks for pointing this out. We will clarify this part in the revision.

28) L174 – Why those months only? How are they 'representative'? Representative of what? I don't see a clear segregation of season, January and December both have negative trends for the whole domain (converse to

the CMA05), April is not as large an increase as March: : : More justification is needed. Are the authors simply showing all of the results which have more warming somewhere?

-Answer: Thanks a lot for the comments. The representative months we selected indeed have a significant warming trend. But it is not limited to these four months. We have shown the warming trend for all months in the Supplementary material. We will add more information on this part. We also add a table for Figure 2-4.

**Table 3. Monthly temperature trends (°C 10a$^{-1}$) in different elevations based on CTMD from 1979–2016.**

|  | Tmin | Tmean | Tmax |
|---|---|---|---|
| January | 0.039*** | 0.036*** | 0.037*** |
| February | 0.033*** | 0.012 | 0.008*** |
| March | 0.023 | 0.009** | 0.017*** |
| April | 0.021*** | -0.02*** | **0.069***** |
| May | -0.056*** | -0.022*** | -0.045*** |
| June | -0.025*** | 0.007 | -0.046*** |
| July | 0.0 | -0.017** | -0.019** |
| August | -0.011 | **0.037***** | **0.023***** |
| September | -0.006 | **0.017**** | **0.038***** |
| October | -0.073*** | -0.018*** | **0.017**** |
| November | -0.032*** | -0.031*** | -0.018*** |
| December | 0.064*** | **0.006**** | -0.018*** |

Note: the bold and underlined value indicates a warming trend for higher elevations, not for the whole elevation range. More details could be found in Figure 2 to 4 and Figure S1 to S12. * denotes the significance level $p<0.1$, ** denotes the significance level $p<0.05$, and *** denotes the significance level $p<0.01$.

29) L176 – Is your average a Mean? Median? Note that median is displayed for boxplots.

-Answer: Thanks a lot for the comments. Because we calculated the monthly and seasonal temperature trends for each grid based on averaged 6-hourly data. Thus, we want to keep the consistent trend calculation for all parts. The boxplot shows the 25% to 75% range with the median value. The regression based on mean value reflects extra information for the whole figure.

30) L185 - Figure 3 now investigates March, April, August and September. Why are the same months not compared and what is the justification this time?

-Answer: Thanks for pointing this out. It illustrates the complexity and variability of EDW. Because the performance of different temperature type (Tmin, Tmean and Tmax) is diverse for different months. We try to select the months with the most significant temperature warming trend.

31) L193 – Months of interest for Tmean are again different.

-Answer: Thanks for pointing this out. Yes, the months of interest are different because the diverse performances for different months. We believe it is better to let the readers know which month has the intense warming trend.

32) L203 – Statistically significantly different? If so, by what test and what significance? Same comment throughout the paragraph, please clarify the significance or reword it.

-Answer: Thanks for pointing this out. We will clarify this paragraph and provide the p-value in the revision.

33) L207 – are warmer on average, the figure rather shows a higher rate of warming. Check sentence.

-Answer: Thanks for pointing this out. We reword this sentence in the revision.

Possible hypotheses and mechanisms

34) I feel that this section should be under the general header of 'discussion'. Please see general comments on this section. I believe that much more is needed for this section. It is very general and I don't go away feeling that I learned anything new.

-Answer: Thanks a lot for the comments. We will move this part to the

discussion section. We will also add some new discussion on the mechanisms, for example, the snow/ice cover changes.

35) L255 – Also the snow cover and snow albedo here affect this: : : This is mentioned in the next paragraph and the information is essentially repeated with no additional information gain.

-Answer: Thanks for pointing this out. We will reword this paragraph in the revision.

36) L264-265 – Sure, this could be a mechanism, but has there been any other studies demonstrating snow cover changes and albedo changes in the CTM? I note that the Deng paper is cited but not investigated further. Because the CTMD product is generated through station observations at lower elevations, would this not bias representation of high elevation changes? Of course, I appreciate that there are no available data at those higher elevations, but this needs to be mentioned and limitations of the dataset/study need to be linked with a more in depth interpretation of the most noteworthy results.

-Answer: Thanks a lot for the comments. We will add more discussion about the impacts of snow/ice cover on the temperature changes in the revision. It is true that there are quite few observations at higher elevation to validate the CTMD. The limitation of the CTMD will be fully demonstrated in the revision.

37) L273 – Could be? Are these model simulations of idealised conditions or did authors find this specifically for that zone? Reword to 'estimated glacier mass loss: : :'

-Answer: Thanks for pointing this out. This value is derived from a glacial model that provided by Dr. Deng (2019). We have contacted Dr. Deng that he will provide more data for our further analysis in the revision.

38) L275 – 'In summary'

-Answer: Thanks for pointing this out. We correct it in the revision.

39) In my opinion, this section needs splitting into; 1) a greater discussion with section 4 (see general comment and above) and, 2) a clear and concise, separate conclusions section.

-Answer: Thanks for pointing this out. We take this suggestion and will rewrite this section in the revision.

40) L284 – 'DO' not (in the case of CTM) clearly reflect EDW. Not cannot.

-Answer: Thanks for pointing this out. We correct it in the revision.

41) L285-286 – This belongs to the previous section. The authors should elaborate whether earlier spring snow melt is significant (and quantify significance) or at least demonstrate if past work suggests that warming at those higher elevations is more likely. Comparing some general estimates of snow line elevation or from previous findings to those same elevation bands would be of value, though I'm sceptical if the CTMD product will reflect that change.

-Answer: Thanks a lot for the comments. We try to find some snow/ice cover data in spring (there is some data that be possible provided by Dr. Deng at Tianshan No. 1 Glacier station in the Urumqi River Basin in the Zone 2) and to check more literatures to validate our conclusions. The ability of CTMD will be discussed comprehensively in the revision. We still keep cautious confidence in the CTMD.

42) L288 – Replace gradients with trends unless referring specifically to the difference across the elevation bands (Figures 2-4). In general, the terminology needs clarification.

-Answer: Thanks for pointing this out. We correct it in the revision.

43) L297 – I think that this is a crucial point. Above 5000 m, there are always positive trends for minimum and some mean temperatures (Figures 2 and 4). I would like to see more discussion as to why we might expect to have a general cooling (negative) trend for the winter minimum below 3000 m. The lack of discussion regarding the mechanisms is a major drawback to the current manuscript version.

-Answer: Thanks a lot for the comments. The reviewer pointed a very important issue. It is true that the discussion on the mechanism is not enough. The land surface process plays a key role regarding the mechanism. The air at high altitudes is similar to the free atmosphere and the dry adiabatic process is dominant. In low-altitude areas, the impact of underlying surface characteristics (e.g. terrain and land cover) is more significant. We will try to improve this part in the revision.

44) L297-298 – Or could be warming as a result of snow cover depletion (feedback)?

-Answer: Thanks a lot for the comments. The melting and retreat of the snow cover will affect the surface albedo, which changes the surface energy balance. We will discuss more on the snow cover in the revision.

45) L297-302 – This reads like a results section again.

-Answer: Thanks for pointing this out. We will reword it in the revision.

Figures

46) -My general issue with the figures is the lack of standardisation (i.e. different colour and y-axis scales) and the ever changing months presented. It leaves the reader with no strong idea as to the key findings.

-I recommend maintaining the same y-axis limits to all sub-plots in Figures 2-4, labelling the months on the plots for easier interpretation.

-Answer: Thanks a lot for the comments. We have responded before. We want to clarify that we did not use a uniform scale (y-axis limits). We have tried. But the temperature increasing trend for some months at some elevation groups are negative. If a uniform scale would be used, the possible range could be -1.6 to 2 ℃ 10a$^{-1}$. Thus, for some months, the box plot will appear very crowded and small, which is in a poor readable for the percentile ranges (25% to 75%). Thus, we keep the different y-axis ranges. However, the referee's comment is reasonable. We figure out a good way to show the trend comparison for all month is adding a table which including all slope and significance levels. The table is as following:

**Table 3. Monthly temperature trends (℃ 10a$^{-1}$) in different elevations based on CTMD from 1979–2016.**

|           | Tmin        | Tmean       | Tmax        |
|-----------|-------------|-------------|-------------|
| January   | 0.039[***]  | 0.036[***]  | 0.037[***]  |
| February  | 0.033[***]  | 0.012       | 0.008[***]  |
| March     | 0.023       | 0.009[**]   | 0.017[***]  |
| April     | 0.021[***]  | -0.02[***]  | **0.069**[***] |
| May       | -0.056[***] | -0.022[***] | -0.045[***] |
| June      | -0.025[***] | 0.007       | -0.046[***] |
| July      | 0.0         | -0.017[**]  | -0.019[**]  |
| August    | -0.011      | **0.037**[***] | **0.023**[***] |
| September | -0.006      | **0.017**[**]  | **0.038**[***] |
| October   | -0.073[***] | -0.018[***] | **0.017**[**]  |
| November  | -0.032[***] | -0.031[***] | -0.018[***] |
| December  | 0.064[***]  | **0.006**[**]  | -0.018[***] |

Note: the bold and underlined value indicates a warming trend for higher elevations, not for the whole elevation range. More details could be found in Figure 2 to 4 and Figure S1 to S12. [*] denotes the significance level $p<0.1$, [**] denotes the significance level $p<0.05$, and [***] denotes the significance level $p<0.01$.

47) -For Figures 5-7, please adjust the colour scale from left (blue – negative) to right (red – positive) following the reviewer#2 comments and also set the same total scale limit for each plot (i.e. -1.5 - +1.5_C 10a-1) with 0_C trend always being the same colour (pale yellow or white). Do the authors also report

trends that are not statistically significant? If so, I would also represent these as white or blank pixels if possible. This will aid the reader's ability to interpret and compare the magnitudes of trends between sub-plots/figures as well as areas that aren't statistically significant trends.

-Answer: Thanks for the suggestion. We have revised the figures (e.g. Figure 5). We will try to set the not statistically significant values to white color in the revision.

[Figure]

**Figure 5: Monthly minimum temperature trends (a) January and (b) December for the entire CTM from 1979–2016. The top two sub-plots show the elevation and temperature**

**trend along the terrain profile (black arrow) in Zone 2, respectively.**

48) -I would suggest adding some other figure(s) that shows the interannual variability of Tmin/Tmax/Tmean for some of the highest elevation pixels so we can better interpret how the suspected EDW warming for March/April/(or month of most interest) looks compared to some lower elevations, or compared to the 'background' change of 'non-mountain' regions shown from the CMA05, if the CMA05 and CTMD are indeed comparable (see general comment). These are the two criteria for EDW and need to be more convincingly demonstrated and discussed.

-Answer: Thanks a lot for the comments. The reviewer provides a very good suggestion. We will add more analysis on the comparison of warming trends in high altitudes and lower elevations in the revision.

---

## Referee Report (RR1)

**Re-review of Gao et al. Evidence of elevation-dependent warming from the Chinese Tianshan Mountains.**

The manuscript revision by Lu Gao and co-authors has improved significantly based upon the detailed comments of the editor and all reviewers. The authors have done a lot of additional work to incorporate the reviewer's points and this reflects in a more robust article that better explains the methodology and limitations of EDW exploration in the Tianshan mountain range. While I am generally happy with the changes made to the manuscript, a few comments remain, as well as some small minor text changes. With these changes made, I would recommend the manuscript be accepted for publication.

**General comments:**

1) The results section is nicely divided into a regional, altitudinal and sub-domain focus. However, each section is a little too descriptive and the authors should attempt to shorten each section, focusing upon the main features and utilising the tables and figures to explain all of the individual values of warming and significance etc.

2) I believe that the figures have improved slightly, though I think small changes could still help the reader to navigate the information more easily. See specific comments on figures below.

3) The authors have incorporated information regarding snow following my initial review. I think this is a good additional to the manuscript, though a few small pieces of information are still missing in my opinion. For example, the snow cover rate (or rather fraction) and average(?) depth is provided for the whole CTM for a given month in each year and compared to the mean (all pixel) warming rate for each month? I think the authors could show the elevation of snow cover in those years vs. the EDW without too much additional effort, but adding extra value to the study. I think the authors should in fact add this as an additional (but succinct) results section rather than just in the discussion. Especially as the data are presented earlier in the manuscript.

**Specific comments:**

L24: "..typical high mountain regions…"

L40, add semi-colon after "characteristics".

L43: "..Outside of these mountain ranges."

L45: as L24

L45: add "..BOTH" before " observations and models"

L50: I think a more recent reference regarding water towers (Immerzeel et al. 2020) would be suitable here. Immerzeel, W. W. *et al.* (2020) 'Importance and vulnerability of the world's water towers', *Nature*, 577(7790), pp. 364–369. doi: 10.1038/s41586-019-1822-y.

L60-65: There are a lot of short sentences that could be merged and improved for flow.

L97: The Immerzeel reference would also be appropriate here.

L99: How do the authors quantify "water resources" here? Is this a water equivalent of ice volume? It is not clear to me and should be revised.

L101: warming at what elevation? A mean of the entire CTM? Perhaps clarify that here.

L115: change "system" to "systematic"

L145: change to "low elevation terrain"

L160: Reference needed here to support the 'reliability' of CMA05.

L162: first step? I do not understand what the authors refer to here. Please revise this to clarify.

L167: snow cover rate? The authors describe a snow cover fraction here. This terminology should be ideally used throughout the manuscript. It is not explicit whether this fraction is just for the entire CTM or another area. The authors need to briefly clarify this. As mentioned, I think the authors could better leverage this information, if only simply, in order to show the snow cover fraction by elevation bands. I believe that this would more appropriately indicate the relationship to altitudinally resolved EDW.

L189: but y was just given as variable estimate from equation 2

L203-204: Not clear, please re-write this more clearly. The authors mean that although some pixels did not have significant change, all pixels in CTM were averaged and compared to WCC and LCC?

L231: better to write as "regional warming"

L297: hilltop? The authors refer to Mountain peaks?

L324: not types – metrics or indicators (as previously written)

L326: "terrain" not "terrains"

L335: "for" Tmin

L358: A reference is required for this statement.

L370: These are very small snow depth values and likely within the uncertainty of the microwave measurements? Perhaps the changes in depth are therefore not significant? This is another example where elevation bands of depth could be more informative than the average for the whole CTM.

L377: the reported value is the significance (p) value, not the "remarkable" correlation value.

L396: How can the authors state a higher accuracy of monthly EDW here? There is no evidence that the monthly values are more accurate, rather that they allow the exploration of sub-seasonal trends that are obscured when averaging over several months/the whole year.

**Figures:**

The figures have improved a little, though I still find figures 2-4 could be improved. I think that each subplot could have a title that specifies the month, rather than having to look to the caption, especially as the group of months changes for each figure. I think a righthand axes with a shaded area or bar could be used to indicate the percentage of the pixels in each elevation band as a product of the total area (total pixels). As this does not change between each panel, it could also be added to figure 1.

Figure 1 should also be referred to in the text, as it is not currently.

Figures 5-7: I understand the reviewers point regarding the colour bar scaling being different in each figure. However, I think the authors should still consider setting 0°C 10a-1 to yellow in all figures, so the divergent colour scale (blue negative, red positive) is always equal and the intensity of blues and reds can still be compared for different figures, even though the scale limits are different.

---

## Referee Report (RR2)

The paper is improved by the addition of more detail of the datasets used, and I commend the authors on their careful and considered conclusions, which rightly highlight the complexity of this phenomena. While I still have some reservations about the dataset due to the fundamental importance of the lapse rates used to create the dataset in determining EDW, I understand the difficulties in validating such a dataset, and the limitations are well discussed in the paper now. Some comments are discussed below.

I am clearer now about the purpose of looking at regional warming amplification, however I think the distinction between regional warming amplification and altitude warming amplification could still be made clearer. Unless the terminology of 'regional warming amplification' and 'altitude warming amplification' are used elsewhere in the literature, it might be clearer to stick to the terminology used in Rangwala and Miller (2012), and only use 'elevation dependent warming' to describe altitude warming amplification. It could be made clearer that section 3.1 is considering whether the Chinese Tianshan Mountains are warming faster than the surrounding lowland areas as a whole, and then that 3.2 and 3.3 are looking at elevation dependence within these mountains.

Table 3 and 4: There still seem to be some instances where the warming trends are larger in both Tmin and Tmax than in Tmean. This may be what the data show, but I think it needs some discussion. It suggests either a fundamental change to the diurnal cycle, or that the results may be overly dependent on the hours chosen.

Table 5: It is very useful to have all these put numbers in one table and makes a good addition to the paper. However, the method used to determine the trends is suggesting startling differences between the trends, which are being exasperated by elevation bands used to determine the trend.

For example, April in table 5, there is a suggestion of increased warming with elevation in Tmin and Tmax, but decreased in Tmean. This discrepancy seems to be due to the authors taking the gradient of the slope for minimum and mean temperature from all the elevation bands, but the gradient of the slope for the mean temperature only from 2500 m upward (Fig S4). Could you explain why you chose a different method for Tmax and Tmean? I think the values in table 5 should compare similar slopes, otherwise they are somewhat confusing. Fig S6 is also somewhat surprising, in that in the highest elevation band, the trends for minimum and maximum daily temperature are both smaller than the trend for mean daily temperature.

Figure 5: While the subplots are added are striking, I am not wholly convinced that they are representative of the whole subregion being examined. For example, figure 5 b, in zone 2, if you took a similar transect at the very northern region of zone 2, would you see the opposite results? These subplots would be better based on average temperatures with elevation within each zone, rather than unique transects.

**Minor comments**

Line 75: please provide some references relating to the Alps, Andes and Rockies.

Line 80: is this trend in minimum and maximum temperature differences a worldwide phenomena?

Line 137: some words missing in this sentence 'for example, the lapse rates of ERA_Interim are greater than those from September to December'.

**References**

Rangwala, I., and Miller, J. R.: Climate change in mountains: a review of elevation-dependent warming and its possible causes, Climatic Change, 114, 527-547, 2012.

---

## Referee Report (RR3)

The manuscript has been improved with additional explanations of the methods and limitations since last reviewed. I believe that the conclusions are correct, however I still have some comments about the analysis used, particularly for table 5.

**Major comments**

Table 5, lines 269-276: The authors have clearly addressed the different altitudes over which warming amplification was detected. However I am still concerned about the use of different elevations to determine significant trends. The authors point to studies where EDW was found at high elevations only, and I would suggest that they conduct similar analysis to form a stronger and more meaningful conclusion. If there is a suggestion that altitude dependent warming occurs above a certain threshold (in the discussion, you suggest 4500 m), that threshold can be chosen and all trends analysed on the same grid points above the threshold. Even if this results in fewer significant trends, I believe it will result in more meaningful conclusions. Where many trends are fitted and significant ones are chosen, it increases the likelihood that those that appear significant have occurred by chance. I think the authors have made this clear and evident in their conclusions, but that the analysis could be more focussed to support these conclusions.

Figure 5: It is not clear to me the advantage of showing one transect where elevation and warming trends are both increasing in the subplots. The transects shown here do not give an indication of whether the elevation is the determining factor, just that (e.g. in figure 5), both elevation and temperature trends increase from west to east. The figure 1 included in the authors response suggests to me that there may be many transects where the temperature trend is decreasing with altitude, indicating there may be another control on warming rates, rather than elevation. I would suggest these subplots are removed, or changed to plots of elevation versus temperature trend over the entire sub-region.

Table 2: I can't quite work out what this is showing, what is the significance of, and where are the different significance levels? Is this the percentage of grid points in the entire CTM that show significant warming trends? If so, which significance level is being used?

**Minor comments**

Line 179: please make it clear here that while the 6-hourly data were aggregated to monthly, seasonal and annual time scales, Tmin and Tmax were picked as one of the 4 available UTC times (and which these were). Similarly for line 197

Line 209: grammatical error: " WCC and LCC that represented by excluding the CTM and the QTP from the WCC" -> "WCC and LCC which is represented by excluding the CTM and the QTP from the WCC". Also a general note, that removing some of the acronyms in this paper would greatly improve readability.

Figures 5-6: Please indicate either in the figure captions how the place names correspond to the zones (e.g. where is the Ili valley, Tolm mountains?)

---

## Author Response (AR2)

**Response to Editor**

We would like to thank Editor Dr. Francesca Pellicciotti for reviewing our revised manuscript as well as the responses. In the following, we address all comments point-by-point according to editor's comments.

Thank you very much for your revised manuscript and response to the reviewers. The paper is much improved, and I thank you for your efforts. I still see few problematic issues, however, and I am sending the manuscript to the reviewers again for a second review.

I feel that some of your statements in the responses would need some more evidence to back them, e.g .the reply to Reviewer 1 comment on the accuracy of the high resolution dataset is not entirely convincing; I am not sure I understand what you did in response to the Reviewer 2 comment on the two criteria you used to define EDW; and finally the English still needs major improvements (see e.g. line 224, 232; new lines 272-275; 312-315; etc).

-Answer: Thanks a lot for the comments. Firstly, about the accuracy of the data set, we added more information on the method section according to Referee 1's comment. According to the authors' previous studies, ERA-Interim has a high degree of credibility. In order to enhance credibility and persuasiveness, we collected more research cases from other scholars, especially the evaluation of ERA-Interim over China. For example, Hairiguli et al (2019) concluded that the ERA-Interim could capture the inter-annual variations of monthly mean temperature in the CTM from via comparing with 45 observation sites from 1984 to 2016. Bai et al (2013) found that ERA-Interim temperature data is better than NCEP/NCAR data based on a comparison with 9 observation sites in the CTM from 2004 to 2006.

We understand the concerns of reviewers, especially on the data accuracy.

We still want to explain that most of the model data (GCMs, reanalysis) have errors, and a large part of the errors are systematic biases. After the systematic biases are effectively eliminated, the corrected model data has a good ability to capture the long-term climate change trend. The core of EDW refers to the increasing trend of temperature within a certain time period (generally longer than 30 years). Thus, the model data as well as the corrected data is still the first choice for climate analysis.

We have been working hard to collect more high-resolution data for validation, but with little success. Other gridded data sets (e.g. CRU at 0.5 degree) do not meet the validation requirements (because the CTMD is at 1 km grid). Therefore, this is the result we can get at the present stage. We thank the referee's consideration.

Secondly, about the EDW definition, we tried to clarify in the revision. In the review paper published by Rangwala and Miller (2012) and Mountain Research Initiative EDW Working Group (2015), they claimed that the regional warming amplification and altitude warming amplification are the two fundamental questions for the EDW. It means that these two most important characteristics need to be detected, respectively. Furthermore, they also concluded that the EDW is existent in some typical mountains (e.g. Alps) because the altitude warming amplification can be detected by observation and climate models. However, the regional warming amplification is still difficult to detect because of limited observations at global scale. Therefore, from our understanding, the EDW could be determined once the regional warming amplification or altitude warming amplification could be detected. But it should be pointed out that the regional warming amplification is difficult to judge because of limited observations. Therefore, in this study, the regional warming amplification is detected by comparing the CTM with the continental China, rather than the global. We revised this part in the revision.

Thirdly, the manuscript for language, grammar, clarity, readability, and an appropriate tone are revised by a native speaker via the Elsevier language editing service (https://webshop.elsevier.com/).

[Figure]

Last but not least, we admit that the CTMD is not so perfect that should be further validated by more observations. But we still believe this is a useful and meaningful attempt to reveal the potential EDW phenomenon in the arid high mountains.

In response to a comment by Reviewer 2 you state that "We plan to update a more high resolution data V2.0 (100m, 6-hourly) since the CTMD V1.0 that released in 2018." Is this new dataset available now and did you use or will you be able to use it?

-Answer: Thanks a lot for the comments. Yes, we are working on the CTMD V2.0 for a long time. Here we want to briefly introduce the methodology on this data. The influence of altitude on temperature is basically unchanged at a below 1km spatial resolution. On the contrary, slope and aspect become the

dominant factors. Thus, the CTMD V1.0 solves the problem of the influence of altitude on temperature. But CTMD V1.0 has a relative poor ability to reflect the temperature changes at micro-terrains. Thus, the CTMD V2.0 tends to further consider the impact of the micro-topographic factors (slope and aspect) on the air temperature. The CTMD spatial resolution is upgraded from 1km to 100m grid via the model as follows:

$$T_{100} = T_{1km} \frac{\cos i}{\cos z} \tag{1}$$

$$\cos i = \cos a \cos z + \sin a \sin z \cos(r - b) \tag{2}$$

where, $T_{100}$ is the CTMD V2.0 temperature at 100 m grid, and $T_{1km}$ is the CTMD V1.0 temperature at 1km grid. $i$ is the angle between the normal line of earth's surface and sun's rays; $a$ is the slope; $z$ is the solar zenith angle; $r$ is the sun's azimuth angle, and $b$ is the slope. Previous studies always used fixed zenith angles and azimuth angles, ignoring the diurnal changes of temperature. In the CTMD V2.0, $z$ and $r$ is calculated according to the earth parameters (e.g. earth radius) and the time point of the data set (00, 06, 12, 18 UTC). However, because of the COVID-19, this work is significantly affected and it lags behind our schedule. An important and necessary work is that we must go to the field to collect ground observation data (cooperation with the Glacier Observatory of the Chinese Academy of Sciences) to verify the data set. We hope to release this data set no later than next spring.

References:

Hairiguli, N., Yusufujiang, R., Madiniyati, D., and Roukeyamu, A.: Adaptability Analysis of ERA-Interim and GHCN-CAM Reanalyzed Data Temperature Values in Tianshan Mountains Area,China,Mountain Research, 37(4): 613-621, 2019. (In Chinese with English abstract)

Bai, L., Wang, W., Yao, Y., Ma, J., and Li, L.: Reliability of NCEP/NCAR and ERA-Interim Reanalysis Data on Tianshan Mountainous Area, Desert and

Oasis Meteorology, 7(6): 51-56, 2013. (In Chinese with English abstract)

Mountain Research Initiative EDW Working Group: Elevation-dependent warming in mountain regions of the world, Nature Climate Change, 5, 424-430, 2015.

Rangwala, I., and Miller, J. R.: Climate change in mountains: a review of elevation-dependent warming and its possible causes, Climatic Change, 114, 527-547, 2012.

**Response to Referee 1**

We would like to thank anonymous Referee 1 for reviewing our manuscript. These professional comments are really helpful for improving the manuscript. In the following, we address all comments point-by-point according to referee's comments.

This study intends to reveal EDW in the Chinese Tianshan Mountains using a high resolution data that are developed in the previous study based on ERA-I data in combination with topographic correction method. Despite merits such as clear structure and better writing to be easy to follow, I have three comments in the following:

1. My major concern is the accuracy of data used. This paper does not do a detailed introduce to the high-resolution data, which results in that I cannot evaluate its accuracy or reliability. After a look at the reference provided, it shows that the high resolution data are based on ERA-I reanalysis. ERA-I is developed based on model simulation in addition to weather station observations, so it generally has large uncertainties in such a small region, especially for mountainous region. Because ERA-I includes in suit observations at some weather stations, it may be unsurprised there are very seasonable performance for evaluation using observed data from perhaps the same weather stations.

-Answer: Thanks a lot for the comments. The referee pointed out a very important issue on the accuracy of data, which is the foundation of presented study. That is true that we just provided limited information on the data set because we leave more space to EDW analysis. We accept the comment and will add more data set information in the revision.

Here, we would like to introduce the data set briefly. It is true that the data set

is produced based on ERA-Interim data and an elevation correction method. We also agree that the uncertainty is large for original ERA-Interim. Our previous studies revealed that there are around 3-4 °C systematic bias from original ERA-Interim (Gao et al., 2014, 2017). Thus, a correction is necessary before local application. The correction approach based on the internal lapse rates derived from ERA-Interim has been proven to be effective in the mountains. Although, there is still a less than 2 °C bias after elevation correction, the warming trends could be captured very well (Gao et al., 2018, Table 1). 24 meteorological stations are applied for data set validation in the CTM from 1979-2013. The averaged trend difference between observation and CTMD is only 0.07 °C $10a^{-1}$ respects to annual and seasonal temperatures. Although the CTMD tends to underestimate the trends for minimum temperatures, we still believe that CTMD is reliable to capture the EDW trend.

Table 1. Trends (°C $10a^{-1}$) of annual and seasonal temperatures over the 24 sites in 1979-2013.

|             | Annual | Spring | Summer | Autumn | Winter |
|-------------|--------|--------|--------|--------|--------|
| observation | 0.420  | 0.664  | 0.432  | 0.532  | 0.018  |
| ERA-Interim | 0.378  | 0.659  | 0.530  | 0.448  | -0.153 |
| correction  | 0.349  | 0.638  | 0.478  | 0.443  | -0.195 |

Meanwhile, we would like to emphasize that the CTM is not a small mountain region (larger than 350,000 $km^2$) which only has less than 30 meteorological stations. Most of them are located in the piedmont plains or valleys. Thus, the validation based on limited surface meteorological station may be not objective. We also know that the analysis of future climate change scenarios relies on model data such as GCM outputs, which have large uncertainties. However, the GCM models are still the most powerful tool for climate change analysis, and the trends modeled by the GCM are still credible. Thus, we believe that although ERA-Interim has errors, it has the ability to reveal regional climate changes after elevation correction. Furthermore, ERA-Interim assimilated

ground observation data, which can more accurately reflect local climate change.

The referee also raised another very important issue that it is unsurprised about the seasonable performance of ERA-Interim at certain stations. Here, we would like to clarify briefly. The ERA-Interim applied ECMWF Integrated Forecast System (IFS) which could assimilate observations in the model. However, only a very small part of observations was assimilated. 9 of 24 sites were possible assimilated by IFS for ERA-Interim in the CTM according to ECMWF assimilation records (Gao et al., 2018, Table 2). Only 4 sites with long-term observations (more than 30 years) while other 5 sites (less than 15 years) were assimilated. In other words, ERA-Interim is a relative independent data set (considering the ratio of ground stations amount to the whole CTM area). We believe the performance of ERA-Interim sometimes is "surprised" in such a complex terrains and it is reliable for regional climate change detection.

Table 2. Possible assimilated sites in the CTM in ERA-Interim.

| Name | WMO id | starting date | ending date |
|------|--------|---------------|-------------|
| Jinghe | 51334 | 1979-06-21 | 1993-01-21 |
| Qitai | 51379 | 1979-06-03 | 1985-05-20 |
| Yining | 51431 | 1978-12-31 | 2011-12-31 |
| Urumqi | 51463 | 1978-12-31 | 2011-12-31 |
| Qijiaojing | 51495 | 1979-04-07 | 1993-04-24 |
| Turfan | 51573 | 1981-06-30 | 1984-08-08 |
| Kuche | 51644 | 1978-12-31 | 2011-12-31 |
| Kuerle | 51656 | 1979-01-03 | 1994-12-30 |
| Hami | 52203 | 1978-12-31 | 2011-12-31 |

2. This paper discusses the mechanism only, if data can be used to reveal some mechanism in the research region, it will be a better progress. The mechanism discussed may be suitable for other regions, but is not always in the case for the research region in the present study.

-Answer: Thanks a lot for the comments. The referee is definitely right that the mechanism is the key issue for EDW. We added the snow cover rate and snow

depth data in the analysis, and the relationship between snow and temperature is discussed in the revision. For example:

[revised manuscript text omitted]

3. Some expressions are not very rigorous. Such as Line 83-85, the author say that satellite data have low spatial resolution, which is questionable. Some satellite data with 1 km resolution are the same resolution as data used in this study. The author also say large system errors with satellite data, which needs analyses or references to confirm.

--Answer: Thanks a lot for pointing this out. We agree that some parts are not very rigorous. We checked the full text and revise them in the subsequent revision.

Gao, L., Hao, L., and Chen, X.W.: Evaluation of ERA-interim monthly temperature data over the Tiberan Plateau, Journal of Mountain Science, 11(5): 1154-1168, 2014.

Gao, L., Bernbardt, M., Schulz, K., and Chen, X.W.: Elevation correction of ERA-Interim temperature data in the Tibetan Plateau, International Journal of Climatology, 37(9): 3540-3552, 2017.

Gao, L., Wei, J., Wang, L., Bernhardt, M., Schulz, K., and Chen, X.: A high-resolution air temperature data set for the Chinese Tian Shan in 1979–2016, Earth System Science Data, 10, 2097-2114, 2018.

**Response to Referee 2**

We would like to thank anonymous Referee 2 for reviewing our manuscript. These constructive comments are very important for us to improve the present manuscript. In the following, we address all comments point-by-point according to referee's comments. The detail responses please see the supplement.

General comments:

This article analyses whether elevation-dependent warming (EDW) is present in the Chinese Tianshan Mountains, both overall and at a regional level. The authors present a compelling case for research into this phenomenon, as increased warming in higher regions may have detrimental effects on glacier melt. EDW is judged based on the criteria of regional warming amplification and altitude warming amplification, and these two criteria are assessed for the entirety of the Chinese Tianshan Mountains on a monthly time scale. Furthermore, spatial differences in EDW are assessed across the mountain range. Overall, the paper is well presented and structured, and the discussion and conclusions of this spatially and temporally complicated problem are interesting. However, there are some issues which I think need to be addressed before publication, most importantly the definition of EDW used in the paper and how it relates to the conclusions reached in the paper, and the suitability of the data set used for this analysis, as highlighted below.

Specific comments:

1. Whole paper: The authors have carefully defined elevation-dependent warming (EDW) immediately in the article, namely that two criteria should be met: regional warming amplification and altitude warming amplification. Section 3.1 concludes that regional warming amplification is only present in

any of the minimum, mean and maximum daily temperatures in the months from February to June. However, in section 3.2, warming amplification with altitude is now described as EDW, for example line 183 "The prevalence of EDW is most significant in December...". This is then used for the remainder of the paper, especially in the conclusions. The authors should identify the months which satisfy both regional warming and altitude warming amplification, and these months should be set out clearly as the months where EDW is present.

This needs to be altered throughout the paper, and has substantial implications for the conclusions, as I think there are only one or two months which satisfy both conditions.

-Answer: Thanks a lot for the comments. The reviewer pointed a very important issue. After carefully reviewing the literatures again, we have to admit that our previous definition of EDW were a bit arbitrary. To be precise, regional warming amplification and altitude warming amplification are the two basic characteristics (or "fundamental questions" from Rangwala and Miller, 2012) of EDW. In the previous literatures, although there are many discussions on altitude warming amplification in high mountains, no literature clearly states that regional warming amplification is one of criteria for EDW. We revised this part through the whole paper in the revision.

2. Methods/CTMD dataset: I think there should be some discussion of the suitability and limitation of the CTMD dataset for this analysis, given that the paper is reliant on it. Two particular points stand out: o Gao et al., 2018 gives an analysis of the data set compared to a number of stations; however they are all under 3000 m asl. I do appreciate the difficulty of finding high elevations stations, but do the authors have any evidence that this data set is suitable at elevations of 5000 m asl and above? In addition, Gao et al., 2018 also indicates that the lapse rate from ERA-interim (the correction term used to

downscale ERA-Interim to the 1km scale) is steeper than that seen in the observations. It is often the case that the free atmosphere lapse rate is steeper than the near-ground lapse rate of temperature with elevation, and this difference may cause errors in the 1 km data set used in this paper.

Gao et al., 2018 acknowledge that the trends in the ERA-Interim data, and therefore the CTMD, do not always follow those of the observations. For example, in the minimum daily temperature, the trend in the CTMD considerably underestimates that of the observations. It is not clear whether this bias is constant with elevation, which is essential to the results presented in this manuscript.

-Answer: Thanks a lot for the comments. The reviewer raised a challenge issue on the quality of CTMD. We must admit that the credibility of data in high-altitude areas is always a huge challenge. In Gao et al., 2018, we used 24 sites to validate the CTMD. It is true that all the sites are lower than 3000 m. We are looking for reliable observation data all the time to further verify the quality of CTMD. We plan to update a more high resolution data V2.0 (100m, 6-hourly) since the CTMD V1.0 that released in 2018. However, as far as we know, there are only very few automatic weather observation stations between 3000-5000m. The time series of these observational data is always short than 10 years with some data gaps. Meanwhile, we have to clarify that these observations data are difficult to access due to permission issues. Therefore, we only could evaluate the credibility of CTMD based on limited observations. In general, we could conclude that the CTMD has a small large-scale bias because of small large-scale errors of ERA-Interim. Previous studies claimed that the large-scale errors of ERA-Interim are acceptable with respect to long-term trends (Gao et al., 2012; Simmons et al., 2010).

About the lapse rate, the referee is right that the lapse rate from ERA-interim is steeper than the observations. Figure 4 in Gao et al., 2018 has shown that the

lapse rates of ERA-Interim are greater than observations from September to December. Generally, the influence of elevation on temperature is basically unchanged at a smaller spatial resolution of 1km, while slope and aspect of the terrain become the dominant factors at hundreds meters. It is true that the free atmosphere lapse rate is steeper than the near-ground lapse rate of temperature because of the different radiation mechanism. To overcome this limitation, the downscaling model used different spatial spans, that is, from the near surface layer (~925hPa) to the free atmosphere (~500hPa). The selection of the lapse rate (such as $\Gamma_{700\_925}$) for each grid is completely dependent on its altitude, which reflects a larger elevation range as much as possible for a more real temperature lapse rate as possible. In the downscaling model, we used the ERA-Interim 2-m temperature instead of site temperature. Therefore, the downscaling model is completely independent of ground stations. However, we agree that the ERA-Interim lapse rate may be part of the source of error. Meanwhile, it is a challenge to distinguish this error quantitatively from the ERA-Interim model errors.

The referee is right that the trend of minimum temperature in CTMD does not follow that of observations in Gao et al., 2018. The CTMD in Gao et al., 2018 covers a larger area (818126 km$^2$), which includes such as the plains on the northern slope of the CTM and the basins on the southern slope of CTM. A "Cold Lake" effect may occur within the basin in winter. The lapse rate may be positive rather than negative. For example, the Turfan Basin (below mean sea level) may have a temperature inversion layer in winter. The present study re-defines the Tianshan boundary according to Deng et al (2019). The CTM contains numerous inter-valley basins and oasis. Thus, the trends of minimum temperature in low terrains may be problematic. However, this study focuses on the trend over the whole CTM, and CTMD may not be good enough on the site scale, but it is still representative on the entire region. We added more information on the limitation of CTMD in the revision. For example:

Although, the CTMD was validated by 24 meteorological stations on a daily scale, indicating a high reliability for the climatology trend investigations, the limitations should be fully demonstrated. Whether the lapse rate can accurately reflect the temperature changes at altitudes is worth discussing. For example, the lapse rates of ERA-Interim are greater than observations from September to December, while lapse rate in the free atmosphere is steeper than that near ground because of the different radiation mechanism (Gao et al., 2018a). The lapse rate may be positive rather than negative since the "Cold Lake" effect in winter such as in the Turfan Basin, which may have a temperature inversion layer at night. Under this situation, the downscaling model may be disabled for winter. Therefore, an opposite trend for minimum temperature during winter is captured by the CTMD compared to the slight positive warming trend from 24 sites. Meanwhile, the trend of diurnal temperature range (DTR) is not captured very well by the CTMD in spring and autumn (Gao et al., 2018a). We want to emphasize that the CTMD is only validated by 24 sites, which are mainly in low terrains. The credibility of the CTMD in the high peaks is difficult to evaluate because of few observations exist. However, we believe that the CTMD is still creditable since it could capture the distribution characteristics of temperatures as well as the general warming trends.

3. Table 1 and 2: given the variation over time, it would be useful to know which of these trends is statistically significant.

-Answer: Thanks a lot for pointing this out. We will mark the significance levels with asterisk in Table 3 and 4 in the revision.

**Table 3. Annual and seasonal temperature trends (°C 10a$^{-1}$) in the CTM (based on CTMD) and the whole continental China (WCC) and low-altitude areas (LCC) by excluding the CTM and the QTP from the WCC (both based on CMA05) from 1979–2016.**

|  | CTM | | | WCC | | | LCC | | |
|---|---|---|---|---|---|---|---|---|---|
|  | Tmin | Tmean | Tmax | Tmin | Tmean | Tmax | Tmin | Tmean | Tmax |
| Spring | 0.633 *** | 0.522 *** | 0.640 *** | 0.557 *** | 0.513 *** | 0.518 *** | 0.543 *** | 0.498 *** | 0.505 *** |
| Summer | 0.441 *** | 0.342 *** | 0.266 ** | 0.472 *** | 0.388 *** | 0.378 *** | 0.404 *** | 0.336 *** | 0.348 *** |
| Autumn | 0.302 | 0.200 * | 0.270 | 0.551 *** | 0.458 *** | 0.420 *** | 0.506 *** | 0.411 *** | 0.371 *** |
| Winter | 0.014 | -0.085 | 0.115 | 0.432 *** | 0.361 *** | 0.327 *** | 0.333 ** | 0.257 | 0.211 |
| Annual | 0.347 *** | 0.245 *** | 0.323 *** | 0.503 *** | 0.430 *** | 0.411 *** | 0.446 *** | 0.376 *** | 0.359 *** |

Note: the bold and underlined value indicates a greater warming trend in the CTM than WCC and LCC. * denotes the significance level $p<0.1$, ** denotes the significance level $p<0.05$, and ***

denotes the significance level $p<0.01$.

**Table 4. Monthly temperature trends (°C 10a$^{-1}$) in the CTM (based on CTMD) and the whole continental China (WCC) and low-altitude areas (LCC) by excluding the CTM and the QTP from the WCC (both based on CMA05) from 1979–2016.**

| | CTMD | | | WCC | | | LCC | | |
|---|---|---|---|---|---|---|---|---|---|
| | Tmin | Tmean | Tmax | Tmin | Tmean | Tmax | Tmin | Tmean | Tmax |
| **January** | -0.133 | -0.269 | -0.235 | 0.343 ** | 0.256 | 0.212 | 0.225 | 0.143 | 0.102 |
| **February** | 0.313 | 0.177 | **0.605** ** | 0.558 *** | 0.523 *** | 0.549 ** | 0.486 ** | 0.456 * | 0.475 * |
| **March** | **0.835** ** | **0.818** *** | **1.339** *** | 0.651 *** | 0.672 *** | 0.752 *** | 0.661 *** | 0.673 *** | 0.738 *** |
| **April** | 0.441 | **0.537** *** | **0.664** * | 0.547 *** | 0.522 *** | 0.516 *** | 0.520 *** | 0.503 *** | 0.508 *** |
| **May** | **0.624** ** | 0.211 | -0.082 | 0.475 *** | 0.345 *** | 0.284 *** | 0.447 *** | 0.317 *** | 0.270 *** |
| **June** | **0.752** *** | **0.476** *** | **0.422** *** | 0.516 *** | 0.390 *** | 0.344 *** | 0.467 *** | 0.348 *** | 0.320 *** |
| **July** | 0.227 | 0.331 *** | 0.28 | 0.472 *** | 0.411 *** | 0.416 *** | 0.402 *** | 0.343 *** | 0.359 *** |
| **August** | 0.342 | 0.217 * | 0.095 | 0.429 *** | 0.363 *** | 0.375 *** | 0.343 *** | 0.318 *** | 0.363 *** |
| **September** | 0.246 | 0.237 | 0.33 | 0.559 *** | 0.486 *** | 0.495 *** | 0.517 *** | 0.445 *** | 0.456 *** |
| **October** | 0.273 | 0.18 | 0.227 | 0.524 *** | 0.434 *** | 0.398 *** | 0.496 *** | 0.407 *** | 0.372 ** |
| **November** | 0.386 | 0.183 | 0.252 | 0.569 *** | 0.455 *** | 0.368 ** | 0.503 *** | 0.381 ** | 0.285 |
| **December** | -0.137 | -0.164 | -0.025 | 0.394 *** | 0.303 ** | 0.219 | 0.287 * | 0.171 | 0.055 |

Note: the bold and underlined value indicates a greater warming trend in the CTM than WCC and LCC. * denotes the significance level $p<0.1$, ** denotes the significance level $p<0.05$, and *** denotes the significance level $p<0.01$.

4. Line 128: How were 6-hourly data aggregated to the minimum and maximum temperature? Was any consideration given to the minimum/maximum temperature not occurring at 00, 6, 12, 18 UTC?

--Answer: Thanks a lot for pointing this out. The minimum and maximum temperatures are calculated from four temperature records. The observation standard of the China Meteorological Administration is also the instantaneous temperature four times a day, from 20 o'clock of previous day to 20 o'clock of current day at local time (UTC+8 Beijing time). The minimum/maximum temperature possible occurs at other time, rather than 00, 6, 12, 18 UTC. However, normally, the maximum temperature occurs around 14 o'clock (06:00 UTC). The minimum temperature occurs around 4 to 5 o'clock in the morning, which is close to 18 UTC (2 o'clock at Beijing time). Therefore, there is only limited effect for minimum and maximum temperature calculation from the 6-hourly data set.

5. Related to points 3 and 4: I'm surprised that in some cases, the warming increase in Tmin and Tmax are both greater than the warming increase in Tmean. This suggests some unusual shift in the shape of the diurnal cycle. Could the authors hypothesise as to why this might be?

--Answer: Thanks a lot for pointing this out. We checked the data carefully again. We found that the header of table does not correspond to the data. It means that the data is in the wrong column. We are very sorry for this kind of mistake that shouldn't be. We correct it in the revision.

**Table 3. Annual and seasonal temperature trends (°C 10a$^{-1}$) in the CTM (based on CTMD) and the whole continental China (WCC) and low-altitude areas (LCC) by excluding the CTM and the QTP from the WCC (both based on CMA05) from 1979–2016.**

| | CTM | | | WCC | | | LCC | | |
|---|---|---|---|---|---|---|---|---|---|
| | Tmin | Tmean | Tmax | Tmin | Tmean | Tmax | Tmin | Tmean | Tmax |
| Spring | **0.633** *** | **0.522** *** | **0.640** *** | 0.557 *** | 0.513 *** | 0.518 *** | 0.543 *** | 0.498 *** | 0.505 *** |
| Summer | 0.441 *** | 0.342 *** | 0.266 ** | 0.472 *** | 0.388 *** | 0.378 *** | 0.404 *** | 0.336 *** | 0.348 *** |
| Autumn | 0.302 | 0.200 * | 0.270 | 0.551 *** | 0.458 *** | 0.420 *** | 0.506 *** | 0.411 *** | 0.371 *** |
| Winter | 0.014 | -0.085 | 0.115 | 0.432 *** | 0.361 *** | 0.327 *** | 0.333 ** | 0.257 | 0.211 |
| Annual | 0.347 *** | 0.245 *** | 0.323 *** | 0.503 *** | 0.430 *** | 0.411 *** | 0.446 *** | 0.376 *** | 0.359 *** |

Note: the bold and underlined value indicates a greater warming trend in the CTM than WCC and LCC. * denotes the significance level $p<0.1$, ** denotes the significance level $p<0.05$, and *** denotes the significance level $p<0.01$.

**Table 4. Monthly temperature trends (°C 10a$^{-1}$) in the CTM (based on CTMD) and the whole continental China (WCC) and low-altitude areas (LCC) by excluding the CTM and the QTP from the WCC (both based on CMA05) from 1979–2016.**

| | CTMD | | | WCC | | | LCC | | |
|---|---|---|---|---|---|---|---|---|---|
| | Tmin | Tmean | Tmax | Tmin | Tmean | Tmax | Tmin | Tmean | Tmax |
| **January** | -0.133 | -0.269 | -0.235 | 0.343 ** | 0.256 | 0.212 | 0.225 | 0.143 | 0.102 |
| **February** | 0.313 | 0.177 | **0.605** ** | 0.558 *** | 0.523 *** | 0.549 ** | 0.486 ** | 0.456 * | 0.475 * |
| **March** | **0.835** ** | **0.818** *** | **1.339** *** | 0.651 *** | 0.672 *** | 0.752 *** | 0.661 *** | 0.673 *** | 0.738 *** |
| **April** | 0.441 | **0.537** *** | **0.664** * | 0.547 *** | 0.522 *** | 0.516 *** | 0.520 *** | 0.503 *** | 0.508 *** |
| **May** | **0.624** ** | 0.211 | -0.082 | 0.475 *** | 0.345 *** | 0.284 *** | 0.447 *** | 0.317 *** | 0.270 *** |
| **June** | **0.752** *** | **0.476** | **0.422** *** | 0.516 *** | 0.390 *** | 0.344 *** | 0.467 *** | 0.348 *** | 0.320 *** |
| **July** | 0.227 | 0.331 *** | 0.28 | 0.472 *** | 0.411 *** | 0.416 *** | 0.402 *** | 0.343 *** | 0.359 *** |
| **August** | 0.342 | 0.217 * | 0.095 | 0.429 *** | 0.363 *** | 0.375 *** | 0.343 *** | 0.318 *** | 0.363 *** |
| **September** | 0.246 | 0.237 | 0.33 | 0.559 *** | 0.486 *** | 0.495 *** | 0.517 *** | 0.445 *** | 0.456 *** |
| **October** | 0.273 | 0.18 | 0.227 | 0.524 *** | 0.434 *** | 0.398 *** | 0.496 *** | 0.407 *** | 0.372 ** |
| **November** | 0.386 | 0.183 | 0.252 | 0.569 *** | 0.455 *** | 0.368 ** | 0.503 *** | 0.381 ** | 0.285 |

| December | -0.137 | -0.164 | -0.025 | 0.394 *** | 0.303 ** | 0.219 | 0.287 * | 0.171 | 0.055 |

Note: the bold and underlined value indicates a greater warming trend in the CTM than WCC and LCC. * denotes the significance level $p<0.1$, ** denotes the significance level $p<0.05$, and *** denotes the significance level $p<0.01$.

6. Section 3.3: This analysis of the spatial variations is interesting, and Figures 5-7 quite well represent the first requirement for EDW, that the warming in the region is greater than the surrounding area. However, it is difficult to see the altitude warming amplification from these plots unless you are well-acquainted with the topography of the region (e.g. from figure 5b it's only really possible to see a north-south gradient in area 1, it's not clear that that corresponds with high-low). Would it be possible to add (small) plots such as those in figures 2-4 to figures 5-7 for each region? If it's not possible to fit the graphs on, perhaps the trends and significance could be calculated, such as in figure 2-4? As in point 1, only those areas which fit both criteria should be described as EDW.

--Answer: Thanks a lot for the comment. The reviewer provided a very good suggestion to show the difference in spatial variations. The sub-plot is feasible. We select a certain direction in typical zone 2, and then establish a terrain profile with the corresponding temperature trend. Fox example:

[Figure]

**Figure 5: Monthly minimum temperature trends (a) January and (b) December for the entire CTM from 1979–2016. The ordinate of two sub-plots show the elevation trend and temperature trend along the terrain profile (black arrow indicates the direction) in Zone 2, respectively. The abscissa represents the distance in multiples of the scale.**

[Figure]

**Figure 6: Monthly maximum temperature trends (a) March and (b) September for the entire CTM from 1979–2016. The ordinate of two sub-plots show the elevation trend and temperature trend along the terrain profile (black arrow indicates the direction) in Zone 2, respectively. The abscissa represents the distance in multiples of the scale.**

[Figure]

**Figure 7: Monthly mean temperature trends (a) January and (b) February for the entire CTM from 1979–2016. The ordinate of two sub-plots show the elevation trend and temperature trend along the terrain profile (black arrow indicates the direction) in Zone 2, respectively. The abscissa represents the distance in multiples of the scale.**

Smaller remarks, technical comments and suggestions:

7. Figure 1: Does the bottom right hand corner map show the extent of the CMA05 used in this analysis? If so, please add to the caption. If not, could this be altered to show the CMA05 extent?

--Answer: Thanks a lot for pointing this out. We will revise and add the grid

points of CMA05 in the Figure 1.

[Figure]

**Figure1: Location of the Chinese Tianshan Mountains (CTM).The elevation ranges from 204 m to 7100 m a.s.l., with a DEM resolution of 1 km from SRTM. The grey sub-plot show the extent of the CMA05 at 0.5 °×0.5 °grid.**

8. Introduction: It would be useful to make clear earlier on in the paper that EDW is referring to the rate of warming over a multi-annual scale (rather than, say, rate of warming during the day). This is made clear on line 58 with 'warming trend of annual mean temperature' but could be mentioned earlier.

--Answer: Thanks a lot for pointing this out. We modified the expression in the section 2.4 Analytical methods in the revision.

9. Around line 120 onwards-perhaps mention that the topography comes from SRTM.

--Answer: Thanks a lot for pointing this out. We clarify the source of DEM that comes from SRTM. It is also noted in the caption of Figure 1.

10. Line 136-137: it might be sensible to combine the highest two elevation bands, given that the highest only contains 4 points (which may not be representative in general).

--Answer: Thanks a lot for pointing this out. The referee is right that only four grids above than 7000m. However, we tend to keep these 4 grids because they represent the highest peaks in the entire CTM. Meanwhile, these four grids basically have the similar performance as that of 6500-7000m group.

11. Line 268-270: I think this sentence can be removed as you're only talking about surface albedo here.

--Answer: Thanks a lot for pointing this out. We removed it in the revision.

12. Line 113: remove 'because' (either 'because the system....,the bias could be' or 'The system bias of.... Thus, the bias...'

--Answer: Thanks a lot for pointing this out. We corrected it in the revision.

13. Please consider changing the colour bars in Figs. 5 to 7 so that they are all the same (and ideally centred around 0, so that red is positive, blue is negative and yellow around zero). At first glance it seems that the maximum temperature trends in March have both positive and negative values, which as you point out in the text is not the case. In addition, please flip the colour bars so that negative values are on the left and positive on the right.

--Answer: Thanks a lot for the comment. The suggestion is excellent for improving the readable of figures. We revised the color bar in the revision. For example:

[Figure]

**Figure 5: Monthly minimum temperature trends (a) January and (b) December for the entire CTM from 1979–2016. The ordinate of two sub-plots show the elevation trend and temperature trend along the terrain profile (black arrow indicates the direction) in Zone 2, respectively. The abscissa represents the distance in multiples of the scale.**

14. Give the location of the Ili valley where it first appears on line 208, rather than 210. References Gao, L., Wei, J., Wang, L., Bernhardt, M., Schulz, K., and Chen, X.: A high-resolution air temperature data set for the Chinese TianShan in1979–2016, Earth System Science Data, 10, 2097-2114, 2018.

--Answer: Thanks a lot for pointing this out. We revised it in the revision.

-Answer: Thanks a lot for the comments. The referee is right that the information on the CMA05 is not enough for the readers. We will add more details on the processes of CMA05. In this study, the CMA05 which covers the whole continental China (including the CTM) was compared to CTMD. We think the referee provides a good idea that the CMA05 without the CTM can also be compared. Thus, we added the trend analysis using the CMA05 excluded the CTM as well as the CMA05 excluded the Tibetan Plateau (The TP is considered to be one of the most intense warming regions in China) in the revision.

**Table 3. Annual and seasonal temperature trends (°C 10a$^{-1}$) in the CTM (based on CTMD) and the whole continental China (WCC) and low-altitude areas (LCC) by excluding the CTM and the QTP from the WCC (both based on CMA05) from 1979–2016.**

|  | CTM | | | WCC | | | LCC | | |
|---|---|---|---|---|---|---|---|---|---|
|  | Tmin | Tmean | Tmax | Tmin | Tmean | Tmax | Tmin | Tmean | Tmax |
| Spring | 0.633 *** | 0.522 *** | 0.640 *** | 0.557 *** | 0.513 *** | 0.518 *** | 0.543 *** | 0.498 *** | 0.505 *** |
| Summer | 0.441 *** | 0.342 *** | 0.266 ** | 0.472 *** | 0.388 *** | 0.378 *** | 0.404 *** | 0.336 *** | 0.348 *** |
| Autumn | 0.302 | 0.200 * | 0.270 | 0.551 *** | 0.458 *** | 0.420 *** | 0.506 *** | 0.411 *** | 0.371 *** |
| Winter | 0.014 | -0.085 | 0.115 | 0.432 *** | 0.361 *** | 0.327 *** | 0.333 ** | 0.257 | 0.211 |

| | | | | | | | | | |
|---|---|---|---|---|---|---|---|---|---|
| Annual | 0.347 *** | 0.245 *** | 0.323 *** | 0.503 *** | 0.430 *** | 0.411 *** | 0.446 *** | 0.376 *** | 0.359 *** |

Note: the bold and underlined value indicates a greater warming trend in the CTM than WCC and LCC. * denotes the significance level *p*<0.1, ** denotes the significance level *p*<0.05, and *** denotes the significance level *p*<0.01.

**Table 4. Monthly temperature trends (°C 10a⁻¹) in the CTM (based on CTMD) and the whole continental China (WCC) and low-altitude areas (LCC) by excluding the CTM and the QTP from the WCC (both based on CMA05) from 1979–2016.**

| | CTMD | | | WCC | | | LCC | | |
|---|---|---|---|---|---|---|---|---|---|
| | Tmin | Tmean | Tmax | Tmin | Tmean | Tmax | Tmin | Tmean | Tmax |
| **January** | -0.133 | -0.269 | -0.235 | 0.343 ** | 0.256 | 0.212 | 0.225 | 0.143 | 0.102 |
| **February** | 0.313 | 0.177 | **0.605** ** | 0.558 *** | 0.523 *** | 0.549 ** | 0.486 ** | 0.456 * | 0.475 * |
| **March** | **0.835** ** | **0.818** *** | **1.339** *** | 0.651 *** | 0.672 *** | 0.752 *** | 0.661 *** | 0.673 *** | 0.738 *** |
| **April** | 0.441 | **0.537** *** | **0.664** * | 0.547 *** | 0.522 *** | 0.516 *** | 0.520 *** | 0.503 *** | 0.508 *** |
| **May** | **0.624** ** | 0.211 | -0.082 | 0.475 *** | 0.345 *** | 0.284 *** | 0.447 *** | 0.317 *** | 0.270 *** |
| **June** | **0.752** *** | **0.476** *** | **0.422** *** | 0.516 *** | 0.390 *** | 0.344 *** | 0.467 *** | 0.348 *** | 0.320 *** |
| **July** | 0.227 | 0.331 *** | 0.28 | 0.472 *** | 0.411 *** | 0.416 *** | 0.402 *** | 0.343 *** | 0.359 *** |
| **August** | 0.342 | 0.217 * | 0.095 | 0.429 *** | 0.363 *** | 0.375 *** | 0.343 *** | 0.318 *** | 0.363 *** |
| **September** | 0.246 | 0.237 | 0.33 | 0.559 *** | 0.486 *** | 0.495 *** | 0.517 *** | 0.445 *** | 0.456 *** |
| **October** | 0.273 | 0.18 | 0.227 | 0.524 *** | 0.434 *** | 0.398 *** | 0.496 *** | 0.407 *** | 0.372 ** |
| **November** | 0.386 | 0.183 | 0.252 | 0.569 *** | 0.455 *** | 0.368 ** | 0.503 *** | 0.381 ** | 0.285 |
| **December** | -0.137 | -0.164 | -0.025 | 0.394 *** | 0.303 ** | 0.219 | 0.287 * | 0.171 | 0.055 |

Note: the bold and underlined value indicates a greater warming trend in the CTM than WCC and LCC. * denotes the significance level *p*<0.1, ** denotes the significance level *p*<0.05, and *** denotes the significance level *p*<0.01.

3) In some places, a justification for showing some months and not others are needed. Figures for Tmin, Tmax and Tmean all show different months, for example. Is this purely just to show the months with the strongest trends? Some work needs to go into the figures as well. I see that that has begun already based upon comments fromreviewer#2. In each figure, the authors show different scales (y-axis limits are different in Figures 2-4 and colour scales are different in each subplot for Figures 5-7), and it becomes hard for the reader to easily compare and understand them, and take away the key message(s). See specific comments on the figures below.

-Answer: Thanks a lot for pointing this issue out. We must admit that the representative months we selected indeed have a significant warming trend. But it is not limited to these four months. We have shown the warming trend for

all months in the Supplementary material. Here we want to clarify that we did not use a uniform scale (y-axis limits). We have tried. But the temperature increasing trend for some months at some elevation groups are negative. If a uniform scale used, the possible range could be -1.6 to 2 $^\circ$C $10a^{-1}$. Thus, for some months, the box plot will appear very crowded and small, which is in a poor readable for the percentile ranges (25% to 75%). Thus, we keep the different y-axis ranges. However, the referee's comment is reasonable. We figure out a good way to show the trend comparison for all month is adding a table which including all slope and significance levels. The table is as following:

**Table 5. Monthly temperature trends ($^\circ$C $10a^{-1}$) over different elevations based on CTMD from 1979–2016.**

| | Tmin | Tmean | Tmax |
|---|---|---|---|
| January | 0.039*** | 0.036*** | 0.037*** |
| February | 0.033*** | 0.012 | 0.008*** |
| March | 0.023 | 0.009** | 0.017*** |
| April | 0.021*** | -0.02*** | **0.069***** |
| May | -0.056*** | -0.022*** | -0.045*** |
| June | -0.025*** | 0.007 | -0.046*** |
| July | 0.0 | -0.017** | -0.019** |
| August | -0.011 | **0.037***** | **0.023***** |
| September | -0.006 | **0.017**** | **0.038***** |
| October | -0.073*** | -0.018*** | **0.017**** |
| November | -0.032*** | -0.031*** | -0.018*** |
| December | 0.064*** | **0.006**** | -0.018*** |

Note: the bold and underlined value indicates a warming trend for higher elevations, not for the whole elevation range. More details could be found in Figure 2 to 4 and Figure S1 to S12. * denotes the significance level $p<0.1$, ** denotes the significance level $p<0.05$, and *** denotes the significance level $p<0.01$.

4) The manuscript presents a rather general discussion with little further exploration of possible mechanisms. There is a repetition of general comments regarding, for example, the albedo's role on the surface energy balance, but this never links with why we may see EDW in certain months or why the strongest warming may occur only for Tmin in January/December and why Tmax trends or regional (east-west) temperature trends (e.g. Figure 5) might

occur. A reference of Deng et al. 2019 is given, for example, but it is not elaborated upon much. Can this or other datasets or analyses regarding snow cover/albedo from MODIS tell us more about why EDW might be occurring for certain seasons/mountains/zones? I don't suggest that the authors do a full analysis of snow cover, but some additional and more in-depth discussion points are definitely required.

-Answer: Thanks a lot for the comments. The reviewer pointed a very key issue. The physical mechanism of EDW is indeed a challenge issue. The current researches are more about the hypothetical mechanism, rather than quantitative physical mechanism investigation. The reviewer's comment is very constructive. We added the snow cover rate and snow depth data in the analysis, and the relationship between snow and temperature is discussed in the revision. For example:

[revised manuscript text omitted]

5) Finally, throughout the manuscript, the terminology of EDW and trends/gradients shifts somewhat and consistency is required throughout (following a clear initial definition). Moreover, the use of the word 'significantly' comes up a lot to refer to differences in trends across space (for the maps) and time (for seasons/months). Unless these differences are tested for significance and values reported, care should be taken for the wording and adjusted appropriately.

-Answer: Thanks a lot for the comments. The referee 2 also pointed out the terminology problem. We admit that we did not give a very clear definition on

EDW, even some misunderstanding. In the revision, we clarified the EDW definition as well as its features. The trends indeed represent different means respect to space and temporal scale.

Specific comments:

6) Abstract L26 -What are EDW 'Features'? I would consider rewording this.

-Answer: Thanks a lot for the comments. To be precise, regional warming amplification and altitude warming amplification are the two basic EDW characteristics. We reword this part in the revision.

7) L26-27 – Please add here the time period over which CMTD was derived and analysed (1979- 2016?)

-Answer: Thanks a lot for the comments. We added the time series 1979-2016 in the revision.

8) L28 – Statistically significant elevation dependence? Add that if so.

-Answer: Thanks a lot for the comments. We added the statistical significances in the revision.

9) L34 – While I do not disagree that this is a likely contributor to glacier melt in the CTM, the authors do not explicitly 'explain' this link, especially as the EDW trends are not so clear for all summer months. It's possible that stronger trends in warming at high elevations in April could have a key influence on some more precipitation falling as rain, but again, the authors cannot (based upon the presented work) state this. I would rephrase this to something like "This new evidence could partly explain the accelerated melting of glaciers in the CTM, though the mechanisms remain to be explored" or similar.

-Answer: Thanks a lot for the comments. Our conclusion may be a little bit arbitrary. We revised this part in the revision.

10) L36 – two 'criteria

-Answer: Thanks for pointing this out. We revised it in the revision.

11) L50 – Current 'evidence'

-Answer: Thanks for pointing this out. We revised it in the revision.

12) L54 – Please elaborate here and add some reasoning of seasonal significance from those studies.

-Answer: Thanks for pointing this out. We added specific information on it in the revision.

13) L58 – What is global mountain detection? Do the authors refer to detection of trends or 'observations' in general for mountain regions? Please clarify and reword.

-Answer: Thanks for pointing this out. "Global mountain detection" means the researcher investigated the temperature trends for most of large mountains over the world. We revised this literature.

14) L58-74 This paragraph reads rather disjointed without a clear flow or argument. Because it recounts several other instances of studies exploring EDW, the overview might be more valuable to the reader in a tabular format? I would suggest to restructure this paragraph and improve the flow of the writing.

-Answer: Thanks for pointing this out. We restructured this paragraph and improve the flow of the writing in the revision.

15) L72-73 – Please clarify what satellite data the authors refer to and how that shows EDW/climate warming at specified elevations. How does this point fit

into the context of the manuscript discussion and/or the strengths/limitations of the presented dataset?

-Answer: Thanks for pointing this out. We revised the literatures in the revision.

16) L81 – Do the authors refer to 56 gridded points of a given product presented by You et al.? Please clarify and rewrite.

-Answer: Thanks for pointing this out. We clarified this literature in the revision.

17) L87 – To me the "largest independent latitudinal mountain system" is not clear. Can the authors clarify its meaning or remove it?

-Answer: Thanks for pointing this out. We removed it in the revision.

Data and Methods

18) L109 – CTMD is briefly defined at the end of the introduction, but should be described insufficient detailed before introducing other datasets to compare to it. See my general comment about elaborating on the CTMD product, especially on its derivation and potential limitations for exploring EDW in this manuscript.

-Answer: Thanks a lot for the comments. We added more information in the revision.

Although, the CTMD was validated by 24 meteorological stations on a daily scale, indicating a high reliability for the climatology trend investigations, the limitations should be fully demonstrated. Whether the lapse rate can accurately reflect the temperature changes at altitudes is worth discussing. For example, the lapse rates of ERA-Interim are greater than observations from September to December, while lapse rate in the free atmosphere is steeper than that near ground because of the different radiation mechanism (Gao et al., 2018a). The lapse rate may be positive rather than negative since the "Cold Lake" effect in winter such as in the Turfan Basin, which may have a temperature inversion layer at night. Under this situation, the downscaling model may be

disabled for winter. Therefore, an opposite trend for minimum temperature during winter is captured by the CTMD compared to the slight positive warming trend from 24 sites. Meanwhile, the trend of diurnal temperature range (DTR) is not captured very well by the CTMD in spring and autumn (Gao et al., 2018a). We want to emphasize that the CTMD is only validated by 24 sites, which are mainly in low terrains. The credibility of the CTMD in the high peaks is difficult to evaluate because of few observations exist. However, we believe that the CTMD is still creditable since it could capture the distribution characteristics of temperatures as well as the general warming trends.

19) L111-Taking all elevations of CMA05? It is not clear how comparable these products are (see general comment). For the CTMD product, the definition of mountain domain is all of the CTMD pixels (including low elevations)? I am left questioning whether the comparison of the CTMD and CMA05 trends are valid and how the values for Table 1 were derived for each of them. More information is required here.

-Answer: Thanks a lot for the comments. We clarified this part and add more analysis in the revision.

**Table 3. Annual and seasonal temperature trends (°C 10a$^{-1}$) in the CTM (based on CTMD) and the whole continental China (WCC) and low-altitude areas (LCC) by excluding the CTM and the QTP from the WCC (both based on CMA05) from 1979–2016.**

|  | CTM | | | WCC | | | LCC | | |
|---|---|---|---|---|---|---|---|---|---|
|  | Tmin | Tmean | Tmax | Tmin | Tmean | Tmax | Tmin | Tmean | Tmax |
| Spring | 0.633 *** | 0.522 *** | 0.640 *** | 0.557 *** | 0.513 *** | 0.518 *** | 0.543 *** | 0.498 *** | 0.505 *** |
| Summer | 0.441 *** | 0.342 *** | 0.266 ** | 0.472 *** | 0.388 *** | 0.378 *** | 0.404 *** | 0.336 *** | 0.348 *** |
| Autumn | 0.302 | 0.200 * | 0.270 | 0.551 *** | 0.458 *** | 0.420 *** | 0.506 *** | 0.411 *** | 0.371 *** |
| Winter | 0.014 | -0.085 | 0.115 | 0.432 *** | 0.361 *** | 0.327 *** | 0.333 ** | 0.257 | 0.211 |
| Annual | 0.347 *** | 0.245 *** | 0.323 *** | 0.503 *** | 0.430 *** | 0.411 *** | 0.446 *** | 0.376 *** | 0.359 *** |

Note: the bold and underlined value indicates a greater warming trend in the CTM than WCC and LCC. * denotes the significance level $p<0.1$, ** denotes the significance level $p<0.05$, and *** denotes the significance level $p<0.01$.

**Table 4. Monthly temperature trends (°C 10a$^{-1}$) in the CTM (based on CTMD) and the whole continental China (WCC) and low-altitude areas (LCC) by excluding the CTM and the QTP from the WCC (both based on CMA05) from 1979–2016.**

|  | **CTMD** | **WCC** | **LCC** |
|---|---|---|---|

|  | Tmin | Tmean | Tmax | Tmin | Tmean | Tmax | Tmin | Tmean | Tmax |
|---|---|---|---|---|---|---|---|---|---|
| **January** | -0.133 | -0.269 | -0.235 | 0.343 [**] | 0.256 | 0.212 | 0.225 | 0.143 | 0.102 |
| **February** | 0.313 | 0.177 | **0.605** [**] | 0.558 [***] | 0.523 [***] | 0.549 [**] | 0.486 [**] | 0.456 [*] | 0.475 [*] |
| **March** | **0.835** [**] | **0.818** [***] | **1.339** [***] | 0.651 [***] | 0.672 [***] | 0.752 [***] | 0.661 [***] | 0.673 [***] | 0.738 [***] |
| **April** | 0.441 | **0.537** [***] | **0.664** [*] | 0.547 [***] | 0.522 [***] | 0.516 [***] | 0.520 [***] | 0.503 [***] | 0.508 [***] |
| **May** | **0.624** [**] | 0.211 | -0.082 | 0.475 [***] | 0.345 [***] | 0.284 [***] | 0.447 [***] | 0.317 [***] | 0.270 [***] |
| **June** | **0.752** [***] | **0.476** [***] | **0.422** [***] | 0.516 [***] | 0.390 [***] | 0.344 [***] | 0.467 [***] | 0.348 [***] | 0.320 [***] |
| **July** | 0.227 | 0.331 [***] | 0.28 | 0.472 [***] | 0.411 [***] | 0.416 [***] | 0.402 [***] | 0.343 [***] | 0.359 [***] |
| **August** | 0.342 | 0.217 [*] | 0.095 | 0.429 [***] | 0.363 [***] | 0.375 [***] | 0.343 [***] | 0.318 [***] | 0.363 [***] |
| **September** | 0.246 | 0.237 | 0.33 | 0.559 [***] | 0.486 [***] | 0.495 [***] | 0.517 [***] | 0.445 [***] | 0.456 [***] |
| **October** | 0.273 | 0.18 | 0.227 | 0.524 [***] | 0.434 [***] | 0.398 [***] | 0.496 [***] | 0.407 [***] | 0.372 [**] |
| **November** | 0.386 | 0.183 | 0.252 | 0.569 [***] | 0.455 [***] | 0.368 [**] | 0.503 [***] | 0.381 [**] | 0.285 |
| **December** | -0.137 | -0.164 | -0.025 | 0.394 [***] | 0.303 [**] | 0.219 | 0.287 [*] | 0.171 | 0.055 |

Note: the bold and underlined value indicates a greater warming trend in the CTM than WCC and LCC. [*] denotes the significance level $p<0.1$, [**] denotes the significance level $p<0.05$, and [***] denotes the significance level $p<0.01$.

20) L112 – Can the authors define what is a small large scale error? Small biases over large domains?

-Answer: Thanks for pointing this out. We clarified the bias is $\pm2.5$ K and cited the reference in the revision.

21) L113 – systematic?

-Answer: Thanks for pointing this out. We corrected it in the revision.

22) L116-118 – It would be valuable to recount that winter lapse rates were not well estimated by CTMD compared to the station data as shown by Fig. 4 of Gao et al., 2018. Some mention here (or in the discussion) needs to explore the potential impact that this might have on your results. If, for example, your temperatures at the highest elevations were estimated by the station lapse rates, would they be largely different from what the CTMD gives you? Could this strongly affect the EDW trends for the highest elevations in January/December? I don't expect that the authors should use the low-elevation stations to derive the high elevation temperatures for their analyses, but some discussion on the limitations of CTMD for the current

analyses are required somewhere in the manuscript.

-Answer: Thanks a lot for the comments. The reviewer is right that the limitation of CTMD should be fully demonstrated in the discussion, especially the poor simulation of lapse rate by CTMD in winter.

Although, the CTMD was validated by 24 meteorological stations on a daily scale, indicating a high reliability for the climatology trend investigations, the limitations should be fully demonstrated. Whether the lapse rate can accurately reflect the temperature changes at altitudes is worth discussing. For example, the lapse rates of ERA-Interim are greater than observations from September to December, while lapse rate in the free atmosphere is steeper than that near ground because of the different radiation mechanism (Gao et al., 2018a). The lapse rate may be positive rather than negative since the "Cold Lake" effect in winter such as in the Turfan Basin, which may have a temperature inversion layer at night. Under this situation, the downscaling model may be disabled for winter. Therefore, an opposite trend for minimum temperature during winter is captured by the CTMD compared to the slight positive warming trend from 24 sites. Meanwhile, the trend of diurnal temperature range (DTR) is not captured very well by the CTMD in spring and autumn (Gao et al., 2018a). We want to emphasize that the CTMD is only validated by 24 sites, which are mainly in low terrains. The credibility of the CTMD in the high peaks is difficult to evaluate because of few observations exist. However, we believe that the CTMD is still creditable since it could capture the distribution characteristics of temperatures as well as the general warming trends.

23) L126 – reword to 'six-hourly timestep'

-Answer: Thanks for pointing this out. We corrected it in the revision.

24) L136-138 – fine, but maybe neaten, use of table?

-Answer: Thanks for pointing this out. We used a table here in the revision.

**Table 1. Altitude groups in the CTMD.**

|   | Altitude range (m) | Grid number |
|---|---|---|
| 1 | <500 | 3139 |

| 2 | 500–1000 | 30810 |
|---|---|---|
| 3 | 1000-1500 | 83018 |
| 4 | 1500-2000 | 70229 |
| 5 | 2000-2500 | 46545 |
| 6 | 2500-3000 | 43400 |
| 7 | 3000-3500 | 39579 |
| 8 | 3500-4000 | 28256 |
| 9 | 4000-4500 | 8789 |
| 10 | 4500-5000 | 1666 |
| 11 | 5000-5500 | 496 |
| 12 | 5500-6000 | 150 |
| 13 | 6000-6500 | 54 |
| 14 | >6500 | 4 |

25) L139 – statistical significance of the linear regression? What p-value defines your statistical significance when you use the term significant in the abstract?

-Answer: Thanks a lot for the comments. We used 0.1, 0.05, and 0.01 for p-value to define statistical significance. We add this information for Table 3 and Table 4, as well as the abstract in the revision.

26) L141- averaged is mean or median? (cf boxplots with median red line plotted)

-Answer: Thanks a lot for the comments. Yes, the boxplots show the median value. We used the mean value for consistent trend calculation.

Results

27) L150 – This needs clarification. Do the authors refer to the elevation gradient of decadal temperature trends or the gradient (slope) of the regression line that quantifies the trend in each elevation band? If referring to the latter, please use the word trend (or similar) instead to not confuse with temperature gradient/lapse rate.

-Answer: Thanks for pointing this out. We clarified and used the term "trend" in

the revision.

28) L174 – Why those months only? How are they 'representative'? Representative of what? I don't see a clear segregation of season, January and December both have negative trends for the whole domain (converse to the CMA05), April is not as large an increase as March: : : More justification is needed. Are the authors simply showing all of the results which have more warming somewhere?

-Answer: Thanks a lot for the comments. The representative months we selected indeed have a significant warming trend. But it is not limited to these four months. We have shown the warming trend for all months in the Supplementary material. We will add more information on this part. We also add a table for Figure 2-4.

**Table 5. Monthly temperature trends (°C 10a$^{-1}$) over different elevations based on CTMD from 1979–2016.**

|  | Tmin | Tmean | Tmax |
|---|---|---|---|
| January | 0.039$^{***}$ | 0.036$^{***}$ | 0.037$^{***}$ |
| February | 0.033$^{***}$ | 0.012 | 0.008$^{***}$ |
| March | 0.023 | 0.009$^{**}$ | 0.017$^{***}$ |
| April | 0.021$^{***}$ | -0.02$^{***}$ | **0.069**$^{***}$ |
| May | -0.056$^{***}$ | -0.022$^{***}$ | -0.045$^{***}$ |
| June | -0.025$^{***}$ | 0.007 | -0.046$^{***}$ |
| July | 0.0 | -0.017$^{**}$ | -0.019$^{**}$ |
| August | -0.011 | **0.037**$^{***}$ | **0.023**$^{***}$ |
| September | -0.006 | **0.017**$^{**}$ | **0.038**$^{***}$ |
| October | -0.073$^{***}$ | -0.018$^{***}$ | **0.017**$^{**}$ |
| November | -0.032$^{***}$ | -0.031$^{***}$ | -0.018$^{***}$ |
| December | 0.064$^{***}$ | **0.006**$^{**}$ | -0.018$^{***}$ |

Note: the bold and underlined value indicates a warming trend for higher elevations, not for the whole elevation range. More details could be found in Figure 2 to 4 and Figure S1 to S12. $^{*}$ denotes the significance level $p<0.1$, $^{**}$ denotes the significance level $p<0.05$, and $^{***}$ denotes the significance level $p<0.01$.

29) L176 – Is your average a Mean? Median? Note that median is displayed for boxplots.

-Answer: Thanks a lot for the comments. Because we calculated the monthly

and seasonal temperature trends for each grid based on averaged 6-hourly data. Thus, we want to keep the consistent trend calculation for all parts. The boxplot shows the 25% to 75% range with the median value. The regression based on mean value reflects extra information for the whole figure.

30) L185 - Figure 3 now investigates March, April, August and September. Why are the same months not compared and what is the justification this time?

-Answer: Thanks for pointing this out. It illustrates the complexity and variability of EDW. Because the performance of different temperature type (Tmin, Tmean and Tmax) is diverse for different months. We try to select the months with the most significant temperature warming trend.

31) L193 – Months of interest for Tmean are again different.

-Answer: Thanks for pointing this out. Yes, the months of interest are different because the diverse performances for different months. We believe it is better to let the readers know which month has the intense warming trend.

32) L203 – Statistically significantly different? If so, by what test and what significance? Same comment throughout the paragraph, please clarify the significance or reword it.

-Answer: Thanks for pointing this out. We provided the p-value in the revision.

33) L207 – are warmer on average, the figure rather shows a higher rate of warming. Check sentence.

-Answer: Thanks for pointing this out. We reword this sentence in the revision.

Possible hypotheses and mechanisms

34) I feel that this section should be under the general header of 'discussion'. Please see general comments on this section. I believe that much more is

needed for this section. It is very general and I don't go away feeling that I learned anything new.

-Answer: Thanks a lot for the comments. We changed this section to discussion and added the snow cover rate and snow depth data in the analysis. The relationship between snow and temperature is discussed in the revision. For example:

[revised manuscript text omitted]

35) L255 – Also the snow cover and snow albedo here affect this: : : This is mentioned in the next paragraph and the information is essentially repeated with no additional information gain.

-Answer: Thanks for pointing this out. We reworded this paragraph in the revision.

36) L264-265 – Sure, this could be a mechanism, but has there been any other studies demonstrating snow cover changes and albedo changes in the CTM? I note that the Deng paper is cited but not investigated further. Because the CTMD product is generated through station observations at lower elevations, would this not bias representation of high elevation changes? Of course, I appreciate that there are no available data at those higher elevations, but this needs to be mentioned and limitations of the dataset/study need to be linked with a more in depth interpretation of the most noteworthy results.

-Answer: Thanks a lot for the comments. We will add more discussion about the impacts of snow/ice cover on the temperature changes in the revision. It is true that there are quite few observations at higher elevation to validate the

CTMD. The limitation of the CTMD is fully demonstrated in the revision. Please see the answers for Q34 and Q22.

37) L273 – Could be? Are these model simulations of idealised conditions or did authors find this specifically for that zone? Reword to 'estimated glacier mass loss: : :'

-Answer: Thanks for pointing this out. We corrected it.

38) L275 – 'In summary'

-Answer: Thanks for pointing this out. We corrected it in the revision.

Discussion and Conclusions

39) In my opinion, this section needs splitting into; 1) a greater discussion with section 4 (see general comment and above) and, 2) a clear and concise, separate conclusions section.

-Answer: Thanks for pointing this out. We rewrite and add more information in Section 4 in the revision.

40) L284 – 'DO' not (in the case of CTM) clearly reflect EDW. Not cannot.

-Answer: Thanks for pointing this out. We corrected it in the revision.

41) L285-286 – This belongs to the previous section. The authors should elaborate whether earlier spring snow melt is significant (and quantify significance) or at least demonstrate if past work suggests that warming at those higher elevations is more likely. Comparing some general estimates of snow line elevation or from previous findings to those same elevation bands would be of value, though I'm sceptical if the CTMD product will reflect that change.

-Answer: Thanks a lot for the comments. We added more analysis on the

relationship between snow and temperature. Please check the Q34.

42) L288 – Replace gradients with trends unless referring specifically to the difference across the elevation bands (Figures 2-4). In general, the terminology needs clarification.

-Answer: Thanks for pointing this out. We corrected it in the revision.

43) L297 – I think that this is a crucial point. Above 5000 m, there are always positive trends for minimum and some mean temperatures (Figures 2 and 4). I would like to see more discussion as to why we might expect to have a general cooling (negative) trend for the winter minimum below 3000 m. The lack of discussion regarding the mechanisms is a major drawback to the current manuscript version.

-Answer: Thanks a lot for the comments. The reviewer pointed a very important issue. It is true that the discussion on the mechanism is not enough. The land surface process plays a key role regarding the mechanism. The air at high altitudes is similar to the free atmosphere and the dry adiabatic process is dominant. In low-altitude areas, the impact of underlying surface characteristics (e.g. terrain and land cover) is more significant. We added more in section 4 in the revision.

It is worth noting that the temperature trend is always positive at an altitude of 4500 m or higher. However, the temperature has a cooling trend in winter below 3000 m, especially Tmin (Fig. 2). The significant altitude warming amplification phenomenon only could be found above 4500 m for August Tmean (Fig. 3 and Table 5). The air at high altitudes is similar to the free atmosphere and the dry adiabatic process is dominant. The absorption and reflection of solar radiation by the surface mainly determine the temperature change. In low-altitude areas, the impact of underlying surface characteristics (e.g. terrain and land cover) is more significant. The CTM has a complex terrain with many mountain basins and canyons. Since the "Cold Lake" effect in winter, the lapse rate even is positive. A temperature inversion layer often happens in deep canyons at night. In

low-altitude areas, the more surface soil moisture results in latent heat fluxes increasing, which further causes more absorbed solar radiation and then temperature warming in winter (Rangwala et al., 2012). This mechanism is closely related to snowlines and treelines because the migration of snowline and treeline changes the surface albedo (Mountain Research Initiative EDW Working Group, 2015).

44) L297-298 – Or could be warming as a result of snow cover depletion (feedback)?

-Answer: Thanks a lot for the comments. The melting and retreat of the snow cover will affect the surface albedo, which changes the surface energy balance. We discussed more in the section 4 in the revision.

45) L297-302 – This reads like a results section again.

-Answer: Thanks for pointing this out. We reworded it in the revision.

Figures

46) -My general issue with the figures is the lack of standardisation (i.e. different colour and y-axis scales) and the ever changing months presented. It leaves the reader with no strong idea as to the key findings.

-I recommend maintaining the same y-axis limits to all sub-plots in Figures 2-4, labelling the months on the plots for easier interpretation.

-Answer: Thanks a lot for the comments. We have responded before. We want to clarify that we did not use a uniform scale (y-axis limits). We have tried. But the temperature increasing trend for some months at some elevation groups are negative. If a uniform scale would be used, the possible range could be -1.6 to 2 ℃ 10a$^{-1}$. Thus, for some months, the box plot will appear very crowded and small, which is in a poor readable for the percentile ranges (25% to 75%). Thus, we keep the different y-axis ranges. However, the referee's

comment is reasonable. We figure out a good way to show the trend comparison for all month is adding a table which including all slope and significance levels. The table is as following:

**Table 5. Monthly temperature trends (°C 10a⁻¹) at different elevations based on CTMD from 1979–2016.**

|  | Tmin | Tmean | Tmax |
|---|---|---|---|
| January | 0.039*** | 0.036*** | 0.037*** |
| February | 0.033*** | 0.012 | 0.008*** |
| March | 0.023 | 0.009** | 0.017*** |
| April | 0.021*** | -0.02*** | **0.069***** |
| May | -0.056*** | -0.022*** | -0.045*** |
| June | -0.025*** | 0.007 | -0.046*** |
| July | 0.0 | -0.017** | -0.019** |
| August | -0.011 | **0.037***** | **0.023***** |
| September | -0.006 | **0.017**** | **0.038***** |
| October | -0.073*** | -0.018*** | **0.017**** |
| November | -0.032*** | -0.031*** | -0.018*** |
| December | 0.064*** | **0.006**** | -0.018*** |

Note: the bold and underlined value indicates a warming trend for higher elevations, not for the whole elevation range. More details could be found in Figure 2 to 4 and Figure S1 to S12. * denotes the significance level $p<0.1$, ** denotes the significance level $p<0.05$, and *** denotes the significance level $p<0.01$.

47) -For Figures 5-7, please adjust the colour scale from left (blue – negative) to right (red – positive) following the reviewer#2 comments and also set the same total scale limit for each plot (i.e. -1.5 - +1.5_C 10a-1) with 0_C trend always being the same colour (pale yellow or white). Do the authors also report trends that are not statistically significant? If so, I would also represent these as white or blank pixels if possible. This will aid the reader's ability to interpret and compare the magnitudes of trends between sub-plots/figures as well as areas that aren't statistically significant trends.

-Answer: Thanks for the suggestion. We have specially added a table to count the number of grids that at significance levels in the revision. We have revised the figures (e.g. Figure 5). However, we did not t set the not statistically significant values to white color because it affects the readability of figures.

**Table 2. Ratio of sum of grids at different significance levels (*p*<0.1, *p*<0.05 and *p*<0.01) to total grids (356133).**

|  | Tmin | Tmean | Tmax |
|---|---|---|---|
| January | 3.28 | 3.65 | 6.48 |
| February | 9.66 | 0.55 | 56.65 |
| March | 52.02 | 99.35 | 100.00 |
| April | 3.76 | 69.16 | 46.36 |
| May | 46.97 | 29.21 | 7.63 |
| June | 80.33 | 92.37 | 49.63 |
| July | 46.86 | 51.97 | 38.82 |
| August | 35.58 | 56.37 | 40.84 |
| September | 19.87 | 47.77 | 35.32 |
| October | 11.78 | 25.52 | 11.41 |
| November | 12.00 | 14.07 | 14.12 |
| December | 0.38 | 0.00 | 0.00 |

[Figure]

**Figure 5: Monthly minimum temperature trends (a) January and (b) December for the entire CTM from 1979–2016. The ordinate of two sub-plots show the elevation trend and temperature trend along the terrain profile (black arrow indicates the direction) in Zone 2, respectively. The abscissa represents the distance in multiples of the scale.**

48) -I would suggest adding some other figure(s) that shows the interannual variability of Tmin/Tmax/Tmean for some of the highest elevation pixels so we can better interpret how the suspected EDW warming for March/April/(or month of most interest) looks compared to some lower elevations, or compared to the 'background' change of 'non-mountain' regions shown from the CMA05, if the CMA05 and CTMD are indeed comparable (see general

comment). These are the two criteria for EDW and need to be more convincingly demonstrated and discussed.

-Answer: Thanks a lot for the comments. After carefully reviewing the literatures again, we have to admit that our previous definition of EDW were a bit arbitrary. To be precise, regional warming amplification and altitude warming amplification are the two basic characteristics (or "fundamental questions" from Rangwala and Miller, 2012) of EDW. In the previous literatures, although there are many discussions on altitude warming amplification in high mountains, no literature clearly states that regional warming amplification is one of criteria for EDW. We revised this part in the revision.

The reviewer provides a very good suggestion. We added more analysis on the comparison of warming trends in high altitudes and lower elevations based on CAM05 and revised the Table 3 and 4 in the revision. For example:

**Table 3. Annual and seasonal temperature trends ($°C\ 10a^{-1}$) in the CTM (based on CTMD) and the whole continental China (WCC) and low-altitude areas (LCC) by excluding the CTM and the QTP from the WCC (both based on CMA05) from 1979–2016.**

| | CTM | | | WCC | | | LCC | | |
|---|---|---|---|---|---|---|---|---|---|
| | Tmin | Tmean | Tmax | Tmin | Tmean | Tmax | Tmin | Tmean | Tmax |
| Spring | 0.633 *** | 0.522 *** | 0.640 *** | 0.557 *** | 0.513 *** | 0.518 *** | 0.543 *** | 0.498 *** | 0.505 *** |
| Summer | 0.441 *** | 0.342 *** | 0.266 ** | 0.472 *** | 0.388 *** | 0.378 *** | 0.404 *** | 0.336 *** | 0.348 *** |
| Autumn | 0.302 | 0.200 * | 0.270 | 0.551 *** | 0.458 *** | 0.420 *** | 0.506 *** | 0.411 *** | 0.371 *** |
| Winter | 0.014 | -0.085 | 0.115 | 0.432 *** | 0.361 *** | 0.327 *** | 0.333 ** | 0.257 | 0.211 |
| Annual | 0.347 *** | 0.245 *** | 0.323 *** | 0.503 *** | 0.430 *** | 0.411 *** | 0.446 *** | 0.376 *** | 0.359 *** |

Note: the bold and underlined value indicates a greater warming trend in the CTM than WCC and LCC. * denotes the significance level $p<0.1$, ** denotes the significance level $p<0.05$, and *** denotes the significance level $p<0.01$.

**Table 4. Monthly temperature trends ($°C\ 10a^{-1}$) in the CTM (based on CTMD) and the whole continental China (WCC) and low-altitude areas (LCC) by excluding the CTM and the QTP from the WCC (both based on CMA05) from 1979–2016.**

| | CTMD | | | WCC | | | LCC | | |
|---|---|---|---|---|---|---|---|---|---|
| | Tmin | Tmean | Tmax | Tmin | Tmean | Tmax | Tmin | Tmean | Tmax |
| **January** | -0.133 | -0.269 | -0.235 | 0.343 ** | 0.256 | 0.212 | 0.225 | 0.143 | 0.102 |
| **February** | 0.313 | 0.177 | 0.605 ** | 0.558 *** | 0.523 *** | 0.549 ** | 0.486 ** | 0.456 * | 0.475 * |
| **March** | 0.835 ** | 0.818 *** | 1.339 *** | 0.651 *** | 0.672 *** | 0.752 *** | 0.661 *** | 0.673 *** | 0.738 *** |

| | | | | | | | | | |
|---|---|---|---|---|---|---|---|---|---|
| **April** | 0.441 | **0.537** *** | **0.664** * | 0.547 *** | 0.522 *** | 0.516 *** | 0.520 *** | 0.503 *** | 0.508 *** |
| **May** | **0.624** ** | 0.211 | -0.082 | 0.475 *** | 0.345 *** | 0.284 *** | 0.447 *** | 0.317 *** | 0.270 *** |
| **June** | **0.752** *** | **0.476** *** | **0.422** *** | 0.516 *** | 0.390 *** | 0.344 *** | 0.467 *** | 0.348 *** | 0.320 *** |
| **July** | 0.227 | 0.331 *** | 0.28 | 0.472 *** | 0.411 *** | 0.416 *** | 0.402 *** | 0.343 *** | 0.359 *** |
| **August** | 0.342 | 0.217 * | 0.095 | 0.429 *** | 0.363 *** | 0.375 *** | 0.343 *** | 0.318 *** | 0.363 *** |
| **September** | 0.246 | 0.237 | 0.33 | 0.559 *** | 0.486 *** | 0.495 *** | 0.517 *** | 0.445 *** | 0.456 *** |
| **October** | 0.273 | 0.18 | 0.227 | 0.524 *** | 0.434 *** | 0.398 *** | 0.496 *** | 0.407 *** | 0.372 ** |
| **November** | 0.386 | 0.183 | 0.252 | 0.569 *** | 0.455 *** | 0.368 ** | 0.503 *** | 0.381 ** | 0.285 |
| **December** | -0.137 | -0.164 | -0.025 | 0.394 *** | 0.303 ** | 0.219 | 0.287 * | 0.171 | 0.055 |

Note: the bold and underlined value indicates a greater warming trend in the CTM than WCC and LCC.

* denotes the significance level $p<0.1$, ** denotes the significance level $p<0.05$, and *** denotes the significance level $p<0.01$.

---

## Author Response (AR3)

**Response to Editor**

We would like to thank Editor Dr. Francesca Pellicciotti for reviewing our revised manuscript as well as the responses. In the following, we address all comments point-by-point according to editor's comments.

Here are some suggestions of additional papers to consider for your Introduction and discussion:

Li et al., ERL, 2020, Does elevation dependent warming exist in high mountain Asia; You et al., Earth-Science Reviews, 2020, Elevation dependent warming over the Tibetan Plateau: patterns, mechanisms and perspectives; Guo et al., 2021, Sci Bull, Local changes in snow depth dominate the evolving pattern of elevation-dependent warming on the Tibetan Plateau.

Reviewer 1 suggests to cite the reference "Mountain Research Initiative EDW Working Group: Elevation-dependent warming in mountain regions of the world, Nature Climate Change, 5, 424-430, 2015." using "Pepin et al., 2015" rather than the present "Mountain Research Initiative EDW Working Group, 2015".

-Answer: Thanks a lot for the references. We added these publications in the context and also revised the Pepin et al. (2015) citation.

**Response to Referee 2**

We would like to thank anonymous Referee 2 for reviewing our revised manuscript as well as the responses. In the following, we address all comments point-by-point according to referee's comments.

The paper is improved by the addition of more detail of the datasets used, and I commend the authors on their careful and considered conclusions, which rightly highlight the complexity of this phenomena. While I still have some reservations about the dataset due to the fundamental importance of the lapse rates used to create the dataset in determining EDW, I understand the difficulties in validating such a dataset, and the limitations are well discussed in the paper now.

Some comments are discussed below.

I am clearer now about the purpose of looking at regional warming amplification, however I think the distinction between regional warming amplification and altitude warming amplification could still be made clearer. Unless the terminology of 'regional warming amplification' and 'altitude warming amplification' are used elsewhere in the literature, it might be clearer to stick to the terminology used in Rangwala and Miller (2012), and only use 'elevation dependent warming' to describe altitude warming amplification. It could be made clearer that section 3.1 is considering whether the Chinese Tianshan Mountains are warming faster than the surrounding lowland areas as a whole, and then that 3.2 and 3.3 are looking at elevation dependence within these mountains.

-Answer: Thanks a lot for the comment. This is a very important issue. In Rangwala and Miller (2012), they clearly pointed out that "*…we have explored available literature to address two important questions related to climate change in the mountain regions: (1) are mountain regions warming faster than*

*low lying regions, and (2) is there an elevation-dependent climate response within mountain regions? From the available studies, it remains difficult to sufficiently assess whether mountains have warmed at a higher rate than the rest of the global land surface primarily because we lack adequate observations to resolve it conclusively.*" According to their conclusion, we summarized the "fundamental questions" of EDW into two aspects: regional warming amplification and altitude warming amplification. The former one focuses on the climate warming comparison with external regions (also include global), and the latter one focuses on the warming comparison within the mountain. Sometimes the altitude warming amplification is treated EDW in a narrow sense. In our understanding, these two "fundamental questions" must be answered separately.

As the Referee 2 pointed out, we investigated the "regional warming amplification" in Section 3.1 and detected the "altitude warming amplification" in Section 3.2. How does the temperature warming and what are the spatial characteristics within the CTM? To answer this question, we analyzed the spatial patterns of the CTM in Section 3.3. Meanwhile, we discussed the potential EDW mechanism in Section 4. At present, the structure of our paper is clear and complete, and it is consistent with the understanding of Referee 2.

Table 3 and 4: There still seem to be some instances where the warming trends are larger in both Tmin and Tmax than in Tmean. This may be what the data show, but I think it needs some discussion. It suggests either a fundamental change to the diurnal cycle, or that the results may be overly dependent on the hours chosen.

-Answer: Thanks a lot for the comment. We added more discussion with respect to Table 3 and 4 in this section. This phenomenon exists regardless of CTMD or CMA05 in the CTM, WCC and LCC. Therefore, it cannot be simply judged whether it is the reason from the changed diurnal cycle or the data

source. We believe both have impacts on the warming trend. In the up to date publication, You et al (2020) also found this phenomenon over the Tibetan Plateau based on multiple data products.

Table 5: It is very useful to have all these put numbers in one table and makes a good addition to the paper. However, the method used to determine the trends is suggesting startling differences between the trends, which are being exasperated by elevation bands used to determine the trend.

For example, April in table 5, there is a suggestion of increased warming with elevation in Tmin and Tmax, but decreased in Tmean. This discrepancy seems to be due to the authors taking the gradient of the slope for minimum and mean temperature from all the elevation bands, but the gradient of the slope for the mean temperature only from 2500 m upward (Fig S4). Could you explain why you chose a different method for Tmax and Tmean? I think the values in table 5 should compare similar slopes, otherwise they are somewhat confusing. Fig S6 is also somewhat surprising, in that in the highest elevation band, the trends for minimum and maximum daily temperature are both smaller than the trend for mean daily temperature.

-Answer: Thanks a lot for the comment. That is true we used different elevation bands in Table 5 and Figures S1 to S12 in supplementary material. Our purpose is to discover the EDW at different altitudes. For example, if we take the all elevation bands for maximum temperature in April (Fig. S4), there is not any warming trend; even there is a cooling trend below 2500 m. However, it is obvious that there is an obvious warming trend above 2500m altitude. Due to length limitation, the results presented by the figures are only the final findings we found. In fact, in our analysis process, we did a lot of data analysis. Especially we attempted to fit the warming trend with different altitudes. Most of the fits did not reach statistical significance. Thus, we did not show these fitting results. However, this is still an open topic. For example, Li et al (2020)

found a significant EDW in the altitude of 2500–5000m from 1980-2012 in the high mountain Asia. You et al (2020) concluded a clear EDW would be found above 2000 m in the Tibetan Plateau in 1961-1990. We share all the dataset released at https://doi.org/10.1594/PANGAEA.887700. We welcome other scholars to further explore EDW at different altitudes in the CTM. In the cation of Table 5, we also emphasize the bold and underlined value indicates a warming trend at higher altitudes, rather than the whole elevation band. We added more discussion in the revision.

Figure 5: While the subplots are added are striking, I am not wholly convinced that they are representative of the whole subregion being examined. For example, figure 5 b, in zone 2, if you took a similar transect at the very northern region of zone 2, would you see the opposite results? These subplots would be better based on average temperatures with elevation within each zone, rather than unique transects.

-Answer: Thanks a lot for the comment. We agree that the presented transect is just an example. However, if we plot the subplots based on the average temperatures and elevations for each zone, the consistent trend between temperature and trend would be nonexistent, because the altitude differences are averaged. As below Figure 1 shown that, the relationship between December minimum temperature and trend in Zone 2 is very complicated. We cannot arbitrarily judge that there is EDW or not. However, we could found that there are some very good liner relationships between temperature and elevation, which represent the EDW (within the red rectangle). Thus, the subplot (the example transect) we showed in the presented Figure 5 is just one of the "good relationships". We believe there are some other transects with significant EDW.

[Figure]

Figure 1: Scatter of December minimum temperature and elevation in Zone 2.

Line 75: please provide some references relating to the Alps, Andes and Rockies.

-Answer: Thanks a lot for the comment. We added the references in the revision.

Line 80: is this trend in minimum and maximum temperature differences a worldwide phenomena?

-Answer: Thanks a lot for the comment. We made it clear in the revision.

Line 137: some words missing in this sentence 'for example, the lapse rates of ERA_Interim are greater than those from September to December'.

-Answer: Thanks a lot for the comment. We revised this language mistakes in the revision.

**Response to Referee 3**

We would like to thank anonymous Referee 3 for reviewing our revised manuscript as well as the responses. In the following, we address all comments point-by-point according to referee's comments.

The manuscript revision by Lu Gao and co-authors has improved significantly based upon the detailed comments of the editor and all reviewers. The authors have done a lot of additional work to incorporate the reviewer's points and this reflects in a more robust article that better explains the methodology and limitations of EDW exploration in the Tianshan mountain range. While I am generally happy with the changes made to the manuscript, a few comments remain, as well as some small minor text changes. With these changes made, I would recommend the manuscript be accepted for publication.

General comments:

1) The results section is nicely divided into a regional, altitudinal and sub-domain focus. However, each section is a little too descriptive and the authors should attempt to shorten each section, focusing upon the main features and utilizing the tables and figures to explain all of the individual values of warming and significance etc.

-Answer: Thanks a lot for the comment. At present, the length of the article is indeed longer than the previous version, since we answered all referees' comments point-by-point and added some more analysis. We revised some parts in the revision.

2) I believe that the figures have improved slightly, though I think small changes could still help the reader to navigate the information more easily. See specific comments on figures below.

-Answer: Thanks a lot for the comment. We revised the figures according to referee's comments in the revision.

3) The authors have incorporated information regarding snow following my initial review. I think this is a good additional to the manuscript, though a few small pieces of information are still missing in my opinion. For example, the snow cover rate (or rather fraction) and average (?) depth is provided for the whole CTM for a given month in each year and compared to the mean (all pixel) warming rate for each month? I think the authors could show the elevation of snow cover in those years vs. the EDW without too much additional effort, but adding extra value to the study. I think the authors should in fact add this as an additional (but succinct) results section rather than just in the discussion. Especially as the data are presented earlier in the manuscript.

-Answer: Thanks a lot for the comment. The referee is right that "snow cover fraction" is more appropriate. We changed the "snow cover rate" to "snow cover fraction" in the revision. This fraction is for the entire CTM. We added more information about the snow data set in the revision. The snow cover fraction is at annual scale. We have no elevation information of snow cover, only two values (maximum fraction and minimum fraction) per year. Here, we added two more tables to show the relationship between maximum/minimum snow cover fraction and monthly temperatures (Table 1 and 2). The monthly snow depth calculated from daily depth was applied for the relationship of snow depth and temperature.

**Table 1. Relationship ($R^2$) of maximum snow cover fraction (%) and monthly Tmin, Tmean and Tmax from 2002 to 2013.**

|          | Tmin     | Tmean | Tmax  |
|----------|----------|-------|-------|
| January  | 0.086    | 0.024 | 0.117 |
| February | 0.302 *  | 0.038 | 0.009 |
| March    | 0.005    | 0.073 | 0.102 |
| April    | 0.075    | 0.089 | 0.060 |
| May      | 0.162    | 0.000 | 0.012 |
| June     | 0.025    | 0.096 | 0.012 |
| July     | 0.144    | 0.158 | 0.161 |

| | | | |
|---|---|---|---|
| August | 0.033 | 0.036 | 0.001 |
| September | 0.019 | 0.186 | 0.003 |
| October | 0.003 | 0.001 | 0.001 |
| November | 0.060 | 0.097 | 0.017 |
| December | 0.002 | 0.017 | 0.003 |

Note: $^*$ denotes the significance level $p < 0.1$.

**Table 2. Relationship ($R^2$) of minimum snow cover fraction (%) and monthly Tmin, Tmean and Tmax from 2002 to 2013.**

| | Tmin | Tmean | Tmax |
|---|---|---|---|
| January | 0.181 | 0.092 | 0.093 |
| February | 0.198 | 0.320 | 0.073 |
| March | 0.171 | 0.153 | 0.068 |
| April | 0.106 | 0.118 | 0.006 |
| May | 0.031 | 0.296 $^*$ | 0.043 |
| June | 0.085 | 0.244 | 0.020 |
| July | 0.246 | 0.006 | 0.019 |
| August | 0.000 | 0.156 | 0.256 $^*$ |
| September | 0.004 | 0.081 | 0.043 |
| October | 0.056 | 0.026 | 0.022 |
| November | 0.001 | 0.024 | 0.009 |
| December | 0.001 | 0.011 | 0.003 |

Note: $^*$ denotes the significance level $p < 0.1$.

Specific comments:

L24: "..typical high mountain regions…"

-Answer: Thanks a lot for the comment. We revised it in the revision.

L40, add semi-colon after "characteristics".

-Answer: Thanks a lot for the comment. We revised it in the revision.

L43: "..Outside of these mountain ranges."

-Answer: Thanks a lot for the comment. We revised it in the revision.

L45: as L24

-Answer: Thanks a lot for the comment. We revised it in the revision.

L45: add "..BOTH" before " observations and models"

-Answer: Thanks a lot for the comment. We revised it in the revision.

L50: I think a more recent reference regarding water towers (Immerzeel et al. 2020) would be suitable here. Immerzeel, W. W. et al. (2020) 'Importance and vulnerability of the world's water towers', Nature, 577(7790), pp. 364–369. doi: 10.1038/s41586-019-1822-y.

-Answer: Thanks a lot for the comment. We added this reference in the revision.

L60-65: There are a lot of short sentences that could be merged and improved for flow.

-Answer: Thanks a lot for the comment. We revised it in the revision.

L97: The Immerzeel reference would also be appropriate here.

-Answer: Thanks a lot for the comment. We added this reference in the revision.

L99: How do the authors quantify "water resources" here? Is this a water equivalent of ice volume? It is not clear to me and should be revised.

-Answer: Thanks a lot for the comment. Yes, it means ice volume. We revised it in the revision.

L101: warming at what elevation? A mean of the entire CTM? Perhaps clarify that here.

-Answer: Thanks a lot for the comment. We revised it in the revision.

L115: change "system" to "systematic"

-Answer: Thanks a lot for the comment. We revised it in the revision.

-Answer: Thanks a lot for the comment. We revised it in the revision.

L160: Reference needed here to support the 'reliability' of CMA05.

-Answer: Thanks a lot for the comment. We added some references in the revision.

L162: first step? I do not understand what the authors refer to here. Please revise this to clarify.

-Answer: Thanks a lot for the comment. The terrain of China can be roughly divided into three steps according to altitude range. The Qinghai-Tibet Plateau has the highest average altitude. Thus, it is called the first step. To avoid ambiguity, we revised it in the revision.

L167: snow cover rate? The authors describe a snow cover fraction here. This terminology should be ideally used throughout the manuscript. It is not explicit whether this fraction is just for the entire CTM or another area. The authors need to briefly clarify this. As mentioned, I think the authors could better leverage this information, if only simply, in order to show the snow cover fraction by elevation bands. I believe that this would more appropriately indicate the relationship to altitudinally resolved EDW.

-Answer: Thanks a lot for the comment. Yes, "snow cover fraction" is more appropriate, and we revised it in the revision. This fraction is for the entire CTM. The snow cover fraction is at annual scale. It means that we have no elevation information of snow cover, only two values (maximum fraction and minimum fraction) per year. Here, we added two more tables to show the relationship between maximum/minimum snow cover fraction and monthly temperatures (Table 1 and 2). The monthly snow depth calculated from daily depth was applied for the relationship of snow depth and temperature.

**Table 1. Relationship ($R^2$) of maximum snow cover fraction (%) and monthly Tmin, Tmean and Tmax from 2002 to 2013.**

|           | Tmin     | Tmean | Tmax  |
|-----------|----------|-------|-------|
| January   | 0.086    | 0.024 | 0.117 |
| February  | 0.302 *  | 0.038 | 0.009 |
| March     | 0.005    | 0.073 | 0.102 |
| April     | 0.075    | 0.089 | 0.060 |
| May       | 0.162    | 0.000 | 0.012 |
| June      | 0.025    | 0.096 | 0.012 |
| July      | 0.144    | 0.158 | 0.161 |
| August    | 0.033    | 0.036 | 0.001 |
| September | 0.019    | 0.186 | 0.003 |
| October   | 0.003    | 0.001 | 0.001 |
| November  | 0.060    | 0.097 | 0.017 |
| December  | 0.002    | 0.017 | 0.003 |

Note: * denotes the significance level $p < 0.1$.

**Table 2. Relationship ($R^2$) of minimum snow cover fraction (%) and monthly Tmin, Tmean and Tmax from 2002 to 2013.**

|           | Tmin  | Tmean   | Tmax    |
|-----------|-------|---------|---------|
| January   | 0.181 | 0.092   | 0.093   |
| February  | 0.198 | 0.320   | 0.073   |
| March     | 0.171 | 0.153   | 0.068   |
| April     | 0.106 | 0.118   | 0.006   |
| May       | 0.031 | 0.296 * | 0.043   |
| June      | 0.085 | 0.244   | 0.020   |
| July      | 0.246 | 0.006   | 0.019   |
| August    | 0.000 | 0.156   | 0.256 * |
| September | 0.004 | 0.081   | 0.043   |
| October   | 0.056 | 0.026   | 0.022   |
| November  | 0.001 | 0.024   | 0.009   |
| December  | 0.001 | 0.011   | 0.003   |

Note: * denotes the significance level $p < 0.1$.

L189: but y was just given as variable estimate from equation 2

-Answer: Thanks a lot for the comment. We revised it in the revision.

L203-204: Not clear, please re-write this more clearly. The authors mean that although some pixels did not have significant change, all pixels in CTM were averaged and compared to WCC and LCC?

-Answer: Thanks a lot for the comment. Yes, the referee is right. This comparison is to detect the regional warming amplification. Although, the trend in some grids did not reach a statistically significant level, it can still reflect climate warming on a regional scale. Thus, we used the all grids in the CTM for comparison.

L231: better to write as "regional warming"

-Answer: Thanks a lot for the comment. We revised it in the revision.

L297: hilltop? The authors refer to Mountain peaks?

-Answer: Thanks a lot for the comment. We revised it to Mountain peak in the revision.

L324: not types – metrics or indicators (as previously written)

-Answer: Thanks a lot for the comment. We revised it in the revision.

L326: "terrain" not "terrains"

-Answer: Thanks a lot for the comment. We revised it in the revision.

L335: "for" Tmin

-Answer: Thanks a lot for the comment. We revised it in the revision.

L358: A reference is required for this statement.

-Answer: Thanks a lot for the comment. We revised it in the revision.

L370: These are very small snow depth values and likely within the uncertainty of the microwave measurements? Perhaps the changes in depth are therefore not significant? This is another example where elevation bands of depth could be more informative than the average for the whole CTM.

-Answer: Thanks a lot for the comment. We admit that there may be certain errors in the snow depth data, especially the snow depth information was extracted from different remote sensing data. However, the data provider claimed that the data accuracy is above 90% via validation. Thus, this is the best data we could obtain. We believe that the referee's suggestion on the elevation bands of depth is fantastic. We found that Li et al (2021) just published a similar work on the Tibetan Plateau. Currently, the elevation information of snow depth needs more time to collect and process. Thus, this suggestion is a topic of on-going and future research in the CTM.

L377: the reported value is the significance (p) value, not the "remarkable" correlation value.

-Answer: Thanks a lot for the comment. We revised it in the revision.

L396: How can the authors state a higher accuracy of monthly EDW here? There is no evidence that the monthly values are more accurate, rather that they allow the exploration of sub-seasonal trends that are obscured when averaging over several months/the whole year.

-Answer: Thanks a lot for the comment. Yes, the referee is right. This statement is indeed not very rigorous. Previous studies on EDW were mostly on a seasonal scale, and it tends to overlook the potential EDW. According to our experience, the detection of EDW on a monthly scale is more effective and reasonable. We revised it in the revision.

Figures:

The figures have improved a little, though I still find figures 2-4 could be improved. I think that each subplot could have a title that specifies the month, rather than having to look to the caption, especially as the group of months changes for each figure. I think a righthand axes with a shaded area or bar

could be used to indicate the percentage of the pixels in each elevation band as a product of the total area (total pixels). As this does not change between each panel, it could also be added to figure 1.

-Answer: Thanks a lot for the comment. The referee is right. We added the month for all subplots for Figure 2 to 4 for a better readability.

[Figure]

Figure 2: Box plots of monthly minimum temperature trends at different elevations from 1979 to 2016. (a) January, (b) February, (c) April, and (d) December. Thick horizontal lines in boxes show the median values. Boxes indicate the inner-quantile range (25% to 75%) and the whiskers show the full range of the values. The red dashed lines represent the significance of EDW.

About the percentage of the pixels in each elevation band, actually we have provided this information in Table 1. We added one more column to show the percentage information in the revision.

**Table 1. Grid number and percentage for each altitude group over the CTMD.**

|    | Altitude range (m) | Grid number | Percentage (%) |
|----|--------------------|-------------|----------------|
| 1  | <500               | 3139        | 0.881          |
| 2  | 500–1000           | 30810       | 8.651          |
| 3  | 1000-1500          | 83018       | 23.311         |
| 4  | 1500-2000          | 70229       | 19.720         |
| 5  | 2000-2500          | 46545       | 13.069         |
| 6  | 2500-3000          | 43400       | 12.186         |
| 7  | 3000-3500          | 39579       | 11.114         |
| 8  | 3500-4000          | 28256       | 7.934          |
| 9  | 4000-4500          | 8789        | 2.468          |
| 10 | 4500-5000          | 1666        | 0.468          |
| 11 | 5000-5500          | 496         | 0.139          |
| 12 | 5500-6000          | 150         | 0.042          |
| 13 | 6000-6500          | 52          | 0.015          |
| 14 | >6500              | 4           | 0.001          |

Figure 1 should also be referred to in the text, as it is not currently.

-Answer: Thanks a lot for the comment. We revised it in the revision.

Figures 5-7: I understand the reviewers point regarding the colour bar scaling being different in each figure. However, I think the authors should still consider setting 0°C 10a-1 to yellow in all figures, so the divergent colour scale (blue negative, red positive) is always equal and the intensity of blues and reds can still be compared for different figures, even though the scale limits are different.

-Answer: Thanks a lot for the comment. We revised all the figures S14 to S30 in the Supplementary material using the same colour bar for a better comparison. As we expected, sometimes the spatial pattern is not significant. The following is an example:

[Figure]

**Fig. S14 Monthly temperature trends in January for the entire CTM from 1979–2016, minimum temperature (up), maximum temperature (middle) and mean temperature (down).**

Thus, we keep different colour bars in figure 5 to 7. Actually, in the last revision, we have set the yellow (RGB: 255, 255, 0) for zero, blue for negative trend and red for positive trend according to referee's comment. The gradient divergent color (blue and red) represents the changed value. In the software (ArcMap 10.0©), the value of zero is not labeled. We could compare the old version and the revised version for maximum temperature in March (Figure 3).

[Figure]

Figure 3: Monthly maximum temperature trends in March in old version (up) and revised version (down).

References:

Guo, D., Pepin, N., Yang, K., Sun, J., and Li, D.: Local changes in snow depth dominate the evolving pattern of elevation-dependent warming on the Tibetan Plateau, Science Bulletin, 66(11), 1146-1150, 2021.

Immerzeel, W.W., Lutz, A.F., Andrade, M., Bahl, A., Biemans, H., Bolch, T., Hyde, S., Brumby, S., Davies, B.J., Elmore, A.C., Emmer, A., Feng, M., Fernández, A., Haritashya, U., Kargel, J.S., Koppes, M., Kraaijenbrink, P.D.A., Kulkarni, A.V., Mayewski, P.A., Nepal, S., Pacheco, P., Painter, T.H., Pellicciotti, F., Rajaram, H., Rupper, S., Sinisalo, A., Shrestha, A.B., Viviroli, D., Wada, Y., Xiao, C., Yao, T., and Baillie, J.E.M.: Importance and vulnerability of the world's water towers, Nature, 577(7790), 364-369, 2020.

Li, B., Chen, Y., and Shi, X.: Does elevation dependent warming exist in high mountain Asia? Environmental Research Letters, 15, 024012, 2020.

Pepin, N., Bradley, R.S., Diaz, H.F., Baraer, M., Caceres, E.B., Forsythe, N., Fowler, H., Greenwood, G., Hashmi, M.Z., Liu, X.D., Miller, J.R., Ning, L., Ohmura, A., Palazzi, E., Rangwala, I., Schöner, W., Severskiy, I., Shahgedanova, M., Wang, M.B., Williamson, S.N., and Yang, D.Q.: Elevation-dependent warming in mountain regions of the world, Nature Climate Change, 5, 424-430, 2015.

Sun, Q., Miao, C., Duan, Q., and Wang, Y.: Temperature and precipitation changes over the Loess Plateau between 1961 and 2011, based on high-density gauge observations, Global and Planetary Change, 132, 1-10, 2015.

Vuille, M., Franquist, E., Garreaud, R., Lavado Casimiro, W.S., and Cáceres, B.: Impact of the global warming hiatus on Andean temperature, Journal of Geophysical Research: Atmospheres, 120(9), 3745-3757, 2015.

Wu, X., Wang, Z., Zhou, X., Lai, C., and Chen, X.: Trends in temperature extremes over nine integrated agricultural regions in China, 1961–2011, Theoretical And Applied Climatology, 129, 1279-1294, 2017.

You, Q., Chen, D., Wu, F., Pepin, N., Cai, Z., Ahrens, B., Jiang, Z., Wu, Z., Kang, S., and AghaKouchak, A.: Elevation dependent warming over the Tibetan Plateau: Patterns, mechanisms and perspectives, Earth-Science Reviews, 210, 103349, 2020.

---

## Author Response (AR5)

**Response to Referee 2**

We would like to thank anonymous Referee 2 for reviewing our revised manuscript as well as the responses. In the following, we address all comments point-by-point according to referee's comments.

The manuscript has been improved with additional explanations of the methods and limitations since last reviewed. I believe that the conclusions are correct, however I still have some comments about the analysis used, particularly for table 5.

Major comments

Table 5, lines 269-276: The authors have clearly addressed the different altitudes over which warming amplification was detected. However I am still concerned about the use of different elevations to determine significant trends. The authors point to studies where EDW was found at high elevations only, and I would suggest that they conduct similar analysis to form a stronger and more meaningful conclusion. If there is a suggestion that altitude dependent warming occurs above a certain threshold (in the discussion, you suggest 4500 m), that threshold can be chosen and all trends analysed on the same grid points above the threshold. Even if this results in fewer significant trends, I believe it will result in more meaningful conclusions. Where many trends are fitted and significant ones are chosen, it increases the likelihood that those that appear significant have occurred by chance. I think the authors have made this clear and evident in their conclusions, but that the analysis could be more focussed to support these conclusions.

-Answer: This is a very important issue. Thanks a lot for referee's constructive comments. The EDW performances may be different using different altitude thresholds. We recalculated the EDW using different altitude thresholds according to referee's suggestion. We replaced Table 5 by other three new

tables. Meanwhile, we re-draw the Fig. 2 to Fig. 4. We revised the whole Section 3.2.

[revised manuscript text omitted]

Figure 5: It is not clear to me the advantage of showing one transect where elevation and warming trends are both increasing in the subplots. The transects shown here do not give an indication of whether the elevation is the determining factor, just that (e.g. in figure 5), both elevation and temperature trends increase from west to east. The figure 1 included in the authors response suggests to me that there may be many transects where the temperature trend is decreasing with altitude, indicating there may be another control on warming rates, rather than elevation. I would suggest these subplots are removed, or changed to plots of elevation versus temperature trend over the entire sub-region.

-Answer: Thanks a lot for the comment. We fully understand the referee's doubts. As shown in Figure 1 in the last pee-review, there are indeed typical EDW phenomenon on some transects, not just shown in the subplots in Figure 5. As the referee pointed out, such a subplot could lead to misunderstandings, and it cannot represent the entire zone. Therefore, we removed the subplots

from Figure 5-7 in the revised manuscript according to the referee's suggestion.

At the same time, the referee pointed out an important issue. There are indeed many EDW phenomena in the west-east transect. However, there are also some elevation-dependent cooling phenomena (please see blue rectangle in the Figure 1). Thus, on the other hand, it reflects the complexity of temperature changes in high mountains. For sure, the altitude is not the only factor for the temperature variations. The slope and aspect are other important factors responsible for the temperature changes due to the widespread valleys in the CTM. The local micro-terrains directly affect the absorption of solar radiation which would change the land surface processes such as latent heat, sensible heat and evapotranspiration. In order to explain this issue more clearly, we added discussion in the revised manuscript in the Section 3.3, and also added the Figure 1 in the Supplementary Material (Fig. S31) for readers' better understanding.

[Figure]

*Figure S31: Scatter of December minimum temperature trend and elevation in Zone 2. The grids in the red rectangle show the elevation-dependent warming while the grids in the blue rectangle show the elevation-dependent cooling.*

*Figures 5 to 7 show the general features of EDW in four typical areas. Taking Zone 2*

*as an example, Fig. S31 showed the warming rate of minimum temperature in December was amplified with elevation in some certain transects (the grids in the red rectangles). However, there are also some elevation-dependent cooling phenomena in Zone 2 (the grids in the blue rectangle). It reveals that altitude is not the only factor that affects temperature changes. The slope and aspect are other important factors responsible for the temperature changes due to the widespread valleys in the Zone 2. The local micro-terrains directly affect the absorption of solar radiation which would change the land surface processes such as latent heat, sensible heat and evapotranspiration. Thus, the EDW should be further detected on a finer spatial scale in some specific areas.*

[Figure]

*Figure 5: Monthly minimum temperature trends (a) January and (b) December for the*

*entire CTM from 1979 to 2016.*

[Figure]

*Figure 6: Monthly mean temperature trends (a) January and (b) February for the entire CTM from 1979 to 2016.*

[Figure]

*Figure 7: Monthly maximum temperature trends (a) March and (b) September for the entire CTM from 1979 to 2016.*

Table 2: I can't quite work out what this is showing, what is the significance of, and where are the different significance levels? Is this the percentage of grid points in the entire CTM that show significant warming trends? If so, which significance level is being used?

-Answer: Thanks a lot for the comment. The Table 2 was added in the last review round according to a referee's comment. The referee wanted to know how many grids reach the significance levels. Thus, we counted the percentage of grid points in the entire CTM that show significant warming

trends. However, here we counted the sum of grids reach three significance levels (p < 0.1, p < 0.05, and p < 0.01) for Tmin, Tmean and Tmax. For better understanding, we provide more statistics in the revision.

**Table 2. Ratio (%) of grids at different significance levels ($p < 0.1$, $p < 0.05$, and $p < 0.01$) to total grids (356133).**

| | Tmin | | | Tmean | | | Tmax | | |
|---|---|---|---|---|---|---|---|---|---|
| | $p<0.1$ | $p<0.05$ | $p<0.01$ | $p<0.1$ | $p<0.05$ | $p<0.01$ | $p<0.1$ | $p<0.05$ | $p<0.01$ |
| January | 1.57 | 1.47 | 0.24 | 2.65 | 1.00 | 0.00 | 4.38 | 2.10 | 0.00 |
| February | 5.18 | 4.48 | 0.01 | 0.55 | 0.00 | 0.00 | 23.50 | 27.28 | 5.87 |
| March | 17.26 | 21.26 | 13.50 | 4.30 | 19.61 | 75.44 | 0.54 | 7.57 | 91.88 |
| April | 3.09 | 0.66 | 0.00 | 8.93 | 14.26 | 45.97 | 8.13 | 19.05 | 19.18 |
| May | 11.78 | 15.99 | 19.19 | 12.47 | 16.31 | 0.44 | 3.74 | 3.76 | 0.13 |
| June | 16.25 | 24.04 | 40.04 | 11.71 | 24.76 | 55.91 | 6.63 | 6.95 | 36.06 |
| July | 9.71 | 15.10 | 22.05 | 4.20 | 4.46 | 43.31 | 4.09 | 9.99 | 24.74 |
| August | 8.21 | 12.96 | 14.40 | 9.86 | 8.06 | 38.45 | 13.70 | 10.67 | 16.47 |
| September | 6.82 | 10.67 | 2.38 | 4.30 | 18.36 | 25.11 | 9.12 | 10.57 | 15.63 |
| October | 6.01 | 5.58 | 0.18 | 12.47 | 13.05 | 0.00 | 5.44 | 5.66 | 0.31 |
| November | 6.00 | 4.98 | 1.02 | 8.64 | 5.43 | 0.00 | 7.55 | 6.57 | 0.00 |
| December | 0.30 | 0.08 | 0.00 | 0.00 | 0.00 | 0.00 | 0.00 | 0.00 | 0.00 |

**Minor comments**

Line 179: please make it clear here that while the 6-hourly data were aggregated to monthly, seasonal and annual time scales, Tmin and Tmax were picked as one of the 4 available UTC times (and which these were). Similarly for line 197.

-Answer: Thanks a lot for pointing this out. We added the specific information in this part. The four times are 00, 06, 12, 18 UTC. The monthly Tmin and Tmax are calculated from the daily Tmin and Tmax, respectively, according to these four UTCs.

Line 209: grammatical error: " WCC and LCC that represented by excluding the CTM and the QTP from the WCC" -> "WCC and LCC which is represented

by excluding the CTM and the QTP from the WCC". Also a general note, that removing some of the acronyms in this paper would greatly improve readability.

-Answer: Thanks a lot for pointing this out. We revised this grammatical error and we also removed some other acronyms in the revision. For example: "*...and LCC that which is represented by excluding the Tianshan Mountains and the Qinghai-Tibetan Plateau from the whole continental China.*"

Figures 5-6: Please indicate either in the figure captions how the place names correspond to the zones (e.g. where is the Ili valley, Tolm mountains?)

-Answer: Thanks a lot for the comment. This is a very important suggestion. We labeled the important places in Figure 1 in the revision.

[Figure]

**Figure 1: Location of the Chinese Tianshan Mountains (CTM).The elevation ranges from 204 m to 7100 m a.s.l., with a DEM resolution of 1 km from SRTM. The grey sub-plot show the extent of the CMA05 at the 0.5 °×0.5 °grid.**